**HESS-2015 -538**, 31 Jan 2017
# Flood risk reduction and flow buffering as ecosystem
# services: I. Theory on flow persistence, flashiness and base
# flow
Meine van Noordwijk[1,2], Lisa Tanika[1], Betha Lusiana[1]
[1]{World Agroforestry Centre (ICRAF), SE Asia program, Bogor, Indonesia}
[2]{Wageningen University and Research, Plant Production Systems, Wageningen, the Netherlands}
Correspondence to: Meine van Noordwijk (m.vannoordwijk@cgiar.org)
**Abstract**
Flood damage reflects insufficient adaptation of human presence and activity to location and
variability of river flow in a given climate. Flood risk increases when landscapes degrade,
counteracted or aggravated by engineering solutions. Efforts to maintain and restore
buffering as ecosystem function may help adaptation to climate change, but require
quantification of effectiveness in their specific social-ecological context. However, the
specific role of forests, trees, soil and drainage pathways in flow buffering, given geology,
land form and climate, remains controversial. Complementing the scarce heavily
instrumented catchments with reliable long-term data, especially in the tropics, there is a
need for metrics for data-sparse conditions. We present and discuss a flow persistence
metric that relates transmission to river flow of peak rainfall events, to the base flow
component of the water balance.  The dimensionless flow persistence parameter $F_p$ is
defined in a recursive flow model and can be estimated from limited time series of observed
daily flow, without requiring knowledge of spatially distributed rainfall upstream. The $F_p$
metric (or its change over time from what appears to be the local norm) matches local
knowledge concepts. Inter-annual variation in the $F_p$ metric in sample watersheds correlates
with variation in the 'flashiness index' used in existing watershed health monitoring
programs, but the relationship between these metrics varies with context. Inter-annual
variation in $F_p$ also correlates with common base-flow indicators, but again in a way that
varies between watersheds. Further exploration of the responsiveness of $F_p$ in watersheds
with different characteristics to the interaction of land cover and the specific realization of
space-time patterns of rainfall in a limited observation period is needed to evaluate
interpretation of $F_p$ as indicator of anthropogenic changes in watershed condition.
**1   Introduction**
Floods can be the direct result of reservoir dams, log jams or protective dykes breaking, with water
derived from unexpected heavy rainfall, rapid snow melt, tsunamis or coastal storm surges. We
focus here on floods that are associated, at least in the public eye, with watershed degradation.
Degradation of watersheds and its consequences for river flow regime and flooding intensity and
frequency are a widespread concern (Brauman et al., 2007; Bishop and Pagiola, 2012; Winsemius et
al., 2013). Engineering measures (dams, reservoirs, canalization, dykes, and flow regulation) can
significantly alter the flow regime of rivers, and reduce the direct relationship with landscape
conditions in the (upper) catchment (Poff et al., 1997). The life expectancy of such structures
depends, however, on the sediment load of incoming rivers and thus on upper watershed conditions
(Graf et al., 2010). Where 'flow regulation' has been included in efforts to assess an economic value
of ecosystem services, it can emerge as a major component of overall value; the economic damage
of floods to cities build on floodplains can be huge and the benefits of avoiding disasters thus large
(Farber et al., 2002; Turner and Daily, 2002; Brauman et al., 2007). The 'counterfactual' part of any
avoided damage argument, however, depends on metrics that are transparent in their basic concept
and relationship with observables. Basic requirements for a metric to be used in managing issues of
public concern in a complex multistakeholder environment are that it i) has a direct relationship with
a problem that needs to be solved ('salience'), ii) is aligned with current science-based
understanding of how the underpinning systems function and can be managed ('credibility') and iii)
can be understood from local and public/policy perspectives ('legitimacy') (Clark et al. 2011). Figure
1 summarizes these requirements, building on van Noordwijk et al. (2016).
⇨  Figure 1
In the popular discussion on floods, especially in the tropics, a direct relationship with deforestation
and reforestation is still commonly perceived to dominate, and forest cover is seen as salient and
legitimate metric of watershed quality (or of urgency of restoration where it is low). A requirement
for 30% forest cover, is for example included in the spatial planning law in Indonesia in this context
(Galudra and Sirait, 2009).  Yet, rivers are probably dominated by the other 70% of the landscape.
There is a problem with the credibility of assumed deforestation-flood relations (van Noordwijk et
al., 2007; Verbist et al., 2010), beyond the local scales (< 10 km$^2$) of paired catchments where ample
direct empirical proof exists, especially in non-tropical climate zones (Bruijnzeel, 1990, 2004).
Current watershed rehabilitation programs that focus on increasing tree cover in upper watersheds
are only partly aligned with current scientific evidence of effects of large-scale tree planting on
streamflow (Ghimire et al., 2014; Malmer et al., 2010; Palmer, 2009; van Noordwijk et al., 2015a).
The relationship between floods and change in forest quality and quantity, and the availability of
evidence for such a relationship at various scales has been widely discussed over the past decades
(Andréassian, 2004; Bruijnzeel, 2004; Bradshaw et al., 2007; van Dijk et al., 2009). Measurements in
Cote d'Ivoire, for example, showed strong scale dependence of runoff from 30-50% of rainfall at 1
m$^2$ point scale, to 4% at 130 ha watershed scale, linked to spatial variability of soil properties plus
variations in rainfall patterns (Van de Giesen et al., 2000). The ratio between peak and average flow
decreases from headwater streams to main rivers in a predictable manner;  while mean annual
discharge scales with (area)$^{1.0}$, maximum river flow was found to scale with (area)$^{0.4}$ to (area)$^{0.7}$ on
average (Rodríguez-Iturbe and Rinaldo, 2001; van Noordwijk et al., 1998; Herschy, 2002), with even
lower powers for area in flash floods that are linked to an extreme rainfall event over a restricted
area (Marchi et al., 2010). The determinants of peak flow are thus scale-dependent, with space-time
correlations in rainfall interacting with subcatchment-level flow buffering at any point along the
river. Whether and where peak flows lead to flooding depends on the capacity of the rivers to pass
on peak flows towards downstream lakes or the sea, assisted by riparian buffer areas with sufficient
storage capacity (Baldasarre et al., 2013). Reducing local flooding risk by increased drainage
increases flooding risk downstream, challenging the nested-scales management of watersheds to
find an optimal spatial distribution, rather than minimization, of flooding probabilities. Well-studied
effects of forest conversion on peak flows in small upper stream catchments (Bruijnzeel, 2004; Alila
et al., 2009) do not necessarily translate to flooding downstream. With most of the published studies
still referring to the temperate zone, the situation in the tropics (generally in the absence of snow) is
contested (Bonell and Bruijnzeel, 2005). As summarized by Beck et al. (2013) meso- to macroscale
catchment studies (>1 and >10 000 km$^2$, respectively) in the tropics, subtropics, and warm
temperate regions have mostly failed to demonstrate a clear relationship between river flow and
change in forest area. Lack of evidence cannot be firmly interpreted as evidence for lack of effect,
however. Detectability of effects depends on their relative size, the accuracy of the measurement
devices, length of observation period, and background variability of the signal.  A recent econometric
study for Peninsular Malaysia by Tan-Soo et al. (2014) concluded that, after appropriate corrections
for space-time correlates in the data-set for 31 meso- and macroscale basins (554-28,643 km$^2$),
conversion of inland rain forest to monocultural plantations of oil palm or rubber increased the
number of flooding days reported, but not the number of flood events, while conversion of wetland
forests to urban areas reduced downstream flood duration. This Malaysian study may be the first
credible empirical evidence at this scale. The difference between results for flood duration and flood
frequency and the result for draining wetland forests warrant further scrutiny. Consistency of these
findings with river flow models based on a water balance and likely pathways of water under the
influence of change in land cover and land use has yet to be shown. Two recent studies for Southern
China confirm the conventional perspective that deforestation increases high flows, but are
contrasting in effects of Reforestation. Zhou et al. (2010) analysed a 50-year data set for Guangdong
Province in China and concluded that forest recovery had not changed the annual water yield (or its
underpinning water balance terms precipitation and evapotranspiration), but had a statistically
significant positive effect on dry season (low) flows.  Liu et al. (2015), however, found for the
Meijiang watershed (6983 km$^2$) in subtropical China that while historical deforestation had
decreased the magnitudes of low flows (daily flows $\leqq$ Q95%) by 30.1%, low flows were not
significantly improved by Reforestation. They concluded that recovery of low flows by Reforestation
may take much longer time than expected probably because of severe soil erosion and resultant loss
of soil infiltration capacity after deforestation. Changes in river flow patterns over a limited period of
time can be the combined and interactive effects of variations in the local rainfall regime, land cover
effects on soil structure and engineering modifications of water flow that can be teased apart with
modelling tools (Ma et al., 2014).
Lacombe et al. (2015) documented that the hydrological effects of natural regeneration differ from
those of plantation forestry, while forest statistics do not normally differentiate between these
different land covers. In a regression study of the high and low flow regimes in the Volta and
Mekong river basins Lacombe and McCartney (2016) found that in the variation among tributaries
various aspects of land cover and land cover change had explanatory power. Between the two
basins, however, these aspects differed. In the Mekong basin variation in forest cover had no direct
effect on flows, but extending paddy areas resulted in a decrease in downstream low flows, probably
by increasing evapotranspiration in the dry season. In the Volta River Basin, the conversion of forests
to crops (or a reduction of tree cover in the existing parkland system) induced greater downstream
flood flows. This observation is aligned with the experimental identification of an optimal,
intermediate tree cover from the perspective of groundwater recharge in parklands in Burkina Faso
(Ilstedt et al., 2016).
The statistical challenges of attribution of cause and effect in such data-sets are considerable with
land use/land cover effects interacting with spatially and temporally variable rainfall, geological
configuration and the fact that land use is not changing in random fashion or following any pre-
randomized design (Alila et al., 2009; Rudel et al., 2005). Hydrological analysis across 12 catchments
in Puerto Rico by Beck et al. (2013) did not find significant relationships between the change in
forest cover or urban area, and change in various flow characteristics, despite indications that
regrowing forests increased evapotranspiration.
These observations imply that percent tree cover (or other forest related indicators) is probably not
a good metric for judging the ecosystem services provided by a watershed (of different levels of
'health'), and that a metric more directly reflecting changes in river flow may be needed. Here we
will explore a simple recursive model of river flow (van Noordwijk et al., 2011) that (i) is focused on
(loss of) flow predictability, (ii) can account for the types of results obtained by the cited recent
Malaysian study (Tan-Soo et al., 2014), and (iii) may constitute a suitable performance indicator to
monitor watershed 'health' through time.
Before discussing the credibility dimension of river flow metrics, the way these relate to the salience
and legitimacy issues around 'flood damage' as policy issue need attention. The salient issue of
'flood damage' is compatible with a common dissection of risk as the product of exposure, hazard
and vulnerability (steps 1, 2 and 3 in Figure 2). Many aspects beyond forests and tree cover play a
role; in fact these factors are multiple steps away (step 7A) from the direct river flow dynamics that
determine floods. Extreme discharge events plus river-level engineering (steps 4 and 5) co-
determine hazard (step 2), while exposure (step 1) depends on topographic position interacting with
human presence, and vulnerability can be modified by engineering at a finer scale and be further
reduced by advice to leave an area in high-risk periods. A recent study (Jongman et al., 2015) found
that human fatalities and material losses between 1980 and 2010 expressed as a share of the
exposed population and gross domestic product were decreasing with rising income. The planning
needed to avoid extensive damage requires quantification of the risk of higher than usual
discharges, especially at the upper tail end of the flow frequency distribution.
⇨  Figure 2
The statistical scarcity, per definition, of 'extreme events' and the challenge of data collection where
they do occur, make it hard to rely on site-specific empirical data as such. Inference of risks needs
some trust in extrapolation methods, as is often provided by use of trusted underlying mechanisms
and/or data obtained in a geographical proximity. Existing data on flood frequency and duration, as
well as human and economic damage are influenced by topography, soils, human population density
and economic activity, responding to engineered infrastructure (step 5 in Figure 2), as well as the
extreme rainfall events that are their proximate cause (step 6). Subsidence due to groundwater
extraction in urban areas of high population density is a specific problem for a number of cities built
on floodplains (such as Jakarta and Bangkok), but subsidence of drained peat areas has also been
found to increase flooding risks elsewhere (Sumarga et al., 2016). Common hydrological analysis of
flood frequency (called 1 in 10-, 1 in 100-, 1 in 1000-year flood events, for example) relies on direct
observations at step 4 in Fig. 2, but typically requires spatial extrapolation beyond points of data
collection through river flow models that combine at least steps 5 and 6. Relatively simple ways of
including the conditions in the watershed (step 7) in such models rely on the runoff curve number
method (Ponce et al., 1996) and the SWAT (Soil water assessment tool) model that was built on its
foundation (Gassman et al. 2007). Applications on tropical soils have had mixed success (Oliveira et
al. 2016). Describing peak flows as a proportion of the rainfall event that triggered them has a long
history, but where the proportionality factors are estimated for ungauged catchments results may
be unreliable (Efstratoiadis et al., 2014). More refined descriptions of the infiltration process (step
7B) are available, using recursive models as filters on empirical data (Grimaldi et al., 2013), but data
for this approach may not be generally available. According to van den Putte et al. (2013) the Green–
Ampt infiltration equation can be fitted to data for dry conditions when soil crusts limit infiltration,
but not in wet winter conditions. These authors argued that simpler models may be better.
Analysis of likely change in flood frequencies in the context of climate change adaptation has been
challenging (Milly et al., 2002; Ma et al., 2014). There is a lack of simple performance indicators for
watershed health at its point of relating precipitation P and river flow Q (step 4 in Figure 2) that align
with local observations of river behaviour and concerns about its change and that can reconcile
local, public/policy and scientific knowledge, thereby helping negotiated change in watershed
management (Leimona et al., 2015). The behaviour of rivers depends on many climatic (step 6 in
Figure 2) and terrain factors (step 7A-D in Figure 2) that make it a challenge to differentiate between
human induced ecosystem structural change and soil degradation (step 7B) on one hand and
intrinsic variability on the other. Step 8 in Figure 2 represents the direct influence of climate on
vegetation, but also a possible reverse influence (van Noordwijk et al., 2015b). Hydrological models
tend to focus on predicting hydrographs at one or more temporal scales, and are usually tested on
data-sets from limited locations. Despite many decades (if not centuries) of hydrological modelling,
current hydrologic theory, models and empirical methods have been found to be largely inadequate
for sound predictions in ungauged basins (Hrachowitz et al., 2013). Efforts to resolve this through
harmonization of modelling strategies have so far failed. Existing models differ in the number of
explanatory variables and parameters they use, but are generally dependent on empirical data of
rainfall that are available for specific measurement points but not at the spatial resolution that is
required for a close match between measured and modelled river flow. Spatially explicit models
have conceptual appeal (Ma et al., 2010) but have too many degrees of freedom and too many
opportunities for getting right answers for wrong reasons if used for empirical calibration (Beven,
2011). Parsimonious, parameter-sparse models are appropriate for the level of evidence available to
constrain them, but these parameters are themselves implicitly influenced by many aspects of
existing and changing features of the watershed, making it hard to use such models for scenario
studies of changing land use and change in climate forcing. Here we present a more direct approach
deriving a metric of flow predictability that can bridge local concerns and concepts to quantified
hydrologic function: the 'flow persistence' parameter as directly observable characteristic (step 4 in
Figure 2), that can be logically linked to the primary points of intervention in watershed
management, interacting with climate and engineering-based change.
In this contribution to the debate we will first define the metric 'flow persistence' in the context of
temporal autocorrelation of river flow and then derive a way to estimate its numerical value. In part
II we will apply the algorithm to river flow data for a number of contrasting meso-scale watersheds.
In the discussion of this paper we will consider the new flow persistence metric in terms of three
groups of criteria for usable knowledge (Fig. 1; Clark et al., 2011; Lusiana et al., 2011; Leimona et al.,
2015) based on salience (I,II), credibility (III, IV) and legitimacy (V-VII):
I.   Does flow persistence relate to important aspects of watershed behaviour, complementing
existing metrics such as the 'flashiness index' and 'base flow separation' techniques?

II.  Does its quantification help to select management actions?
III.    Is there consistency of numerical results?
IV.    How sensitive is it to bias and random error in data sources?
V.    Does it match local knowledge?
VI.    Can it be used to empower local stakeholders of watershed management?
VII.    Can it inform local risk management?
**2 Flow persistence in water balance equations**
**2.1 Recursive model**
One of the easiest-to-observe aspects of a river is its day-to-day fluctuation in water level, related to
the volumetric flow (discharge) via rating curves (Maidment, 1992). Without knowing details of
upstream rainfall and the pathways the rain takes to reach the river, observation of the daily
fluctuations in water level allows important inferences to be made. It is also of direct utility: sudden
rises can lead to floods without sufficient warning, while rapid decline makes water utilization
difficult. Indeed, a common local description of watershed degradation is that rivers become more
'flashy' and less predictable, having lost a buffer or 'sponge' effect (Joshi et al., 2004; Ranieri et al.,
2004; Rahayu et al., 2013). A simple model of river flow at time t, $Q_t$, is that it is similar to that of the
day before ($Q_{t-1}$), multiplied with $F_p$, a dimensionless parameter called 'flow persistence' (van
Noordwijk et al., 2011) plus an additional stochastic term $Q_{a,t}$:
$Q_t = F_p Q_{t-1} + Q_{a,t}$                                                                                    [1].
$Q_t$ is for this analysis expressed in mm d$^{-1}$, which means that measurements in m$^3$ s$^{-1}$ need to be
divided by the relevant catchment area, with appropriate unit conversion. If river flow were
constant, it would be perfectly predictable, i.e. $F_p$ would be 1.0 and $Q_{a,t}$ zero; in contrast, an $F_p$-value
equal to zero and $Q_{a,t}$ directly reflecting erratic rainfall represents the lowest possible level of
predictability.
The $F_p$ parameter is conceptually identical to the 'recession constant' commonly used in hydrological
models, typically assessed during an extended dry period when the $Q_{a,t}$ term is negligible and
streamflow consists of base flow only (Tallaksen, 1995); empirical deviations from a straight line in a
plot of the logarithm of Q against time are common and point to multiple rather than a single
groundwater pool that contributes to base flow. The larger catchment area has a possibility to get
additional flow from multiple independent groundwater contribution.
As we will demonstrate in a next section, it is possible to derive $F_p$ even when $Q_{a,t}$ is not negligible. In
climates without distinct dry season this is essential; elsewhere it allows a comparison of apparent $F_p$
between wet and dry parts of the hydrologic year. A possible interpretation, to be further explored,
is that decrease over the years of $F_p$ indicates 'watershed degradation' (i.e. greater contrast between
high and low flows), and an increase 'improvement' or 'rehabilitation' (i.e. more stable flows).
If we consider the sum of river flow over a period of time (from 1 to T) we obtain
$\Sigma_1^T Q_t = F_p \Sigma_1^T Q_{t-1} + \Sigma_1^T Q_{a,t}$                                                        [2].
If the period is sufficiently long period for $Q_T$ minus $Q_0$ (the values of $Q_t$ for t=T and t=0, respectively)
to be negligibly small relative to the sum over all t's, we may equate $\Sigma_1^T Q_t$ with $\Sigma_1^T Q_{t-1}$ and obtain a
first way of estimating the $F_p$ value:
$F_p = 1 - \Sigma_1^T Q_{a,t} / \Sigma_1^T Q_t$                     [3].
The stochastic $Q_{a,t}$ can be interpreted in terms of what hydrologists call 'effective rainfall' (i.e. rainfall
minus on-site evapotranspiration, assessed over a preceding time period tx since previous rain
event):
$Q_t = F_p Q_{t-1} + (1-F_p)(P_{tx} - E_{tx})$                 [4].
Where $P_{tx}$ is the (spatially weighted) precipitation on day t (or preceding precipitation released as
snowmelt on day t) in mm $d^{-1}$; $E_{tx}$, also in mm $d^{-1}$, is the preceding evapotranspiration that allowed
for infiltration during this rainfall event (*i.e.* evapotranspiration since the previous soil-replenishing
rainfall that induced empty pore space in the soil for infiltration and retention), or replenishment of
a water film on aboveground biomass that will subsequently evaporate. More complex attributions
are possible, aligning with the groundwater replenishing bypass flow and the water isotopic
fractionation involved in evaporation (Evaristo et al., 2015).
The consistency of multiplying effective rainfall with $(1-F_p)$ can be checked by considering the
geometric series $(1-F_p)$, $(1-F_p) F_p$, $(1-F_p) F_p^2$, ..., $(1-F_p) F_p^n$ which adds up to $(1-F_p)(1 - F_p^n)/(1-F_p)$ or $1 -$
$F_p^n$. This approaches 1 for large n, suggesting that all of the water attributed to time t, *i.e.* $P_t - E_{tx}$,
will eventually emerge as river flow. For $F_p = 0$ all of $(P_t - E_{tx})$ emerges on the first day, and river flow
is as unpredictable as precipitation itself. For $F_p = 1$ all of $(P_t - E_{tx})$ contributes to the stable daily flow
rate, and it takes an infinitely long period of time for the last drop of water to get to the river. For
declining $F_p$, $(1 > F_p > 0)$, river flow gradually becomes less predictable, because a greater part of the
stochastic precipitation term contributes to variable rather than evened-out river flow.
Taking long term summations of the right- and left- hand sides of Eq.(4) we obtain:
$\Sigma Q_t = \Sigma(F_p Q_{t-1} + (1-F_p)(P_t - E_{tx})) = F_p \Sigma Q_{t-1} + (1-F_p)(\Sigma P_t - \Sigma E_{tx})$     [5].
Which is consistent with the basic water budget, $\Sigma Q = \Sigma P - \Sigma E$, at time scales long enough for
changes in soil water buffer stocks to be ignored. As such the total annual, and hence the mean daily
river flow are independent of $F_p$. This does not preclude that processes of watershed degradation or
restoration that affect the partitioning of P over Q and E also affect $F_p$.
**2.2 Base flow**
Clarifying the $Q_{a,t}$ contribution is equivalent with one of several ways to separate base flow from
peak flows. Rearranging Eq.(3) we obtain
$\Sigma_1^T Q_{a,t} = (1 - F_p) \Sigma_1^T Q_t$                    [6].
The $\Sigma Q_{a,t}$ term reflects the sum of peak flows in mm. Its complement, $F_p \Sigma Q_t$, reflects the sum of base
flow, also in mm. For $F_p = 1$ (the theoretical maximum) we conclude that all $Q_{a,t}$ must be zero, and all
flow is 'base flow'.

## 2.3 Low flows

The lowest flow expected in an annual cycle is $Q_x F_p^{Nmax}$ where $Q_x$ is flow on the first day without rain and $N_{max}$ the longest series of dry days. Taken at face value, a decrease in $F_p$ has a strong effect on low-flows, with a flow of 10% of $Q_x$ reached after 45, 22, 14, 10, 8 and 6 days for $F_p$ = 0.95, 0.9, 0.85, 0.8, 0.75 and 0.7, respectively. However, the groundwater reservoir that is drained, equalling the cumulative dry season flow if the dry period is sufficiently long, is $Q_x/(1-F_p)$. If $F_p$ decreases to $F_{px}$ but the groundwater reservoir (Res = $Q_x/(1-F_p)$) is not affected, initial flows in the dry period will be higher ($Q_x F_{px}^i (1-F_{px})$ Res > $Q_x F_p^i (1-F_p)$ Res for i < log((1-$F_{px}$)/(1-$F_p$))/log($F_p$/$F_{px}$)). It thus matters how low flows are evaluated: from the perspective of the lowest level reached, or as cumulative flow. The combination of climate, geology and land form are the primary determinants of cumulative low flows, but if land cover reduces the recharge of groundwater there may be impacts on dry season flow, that are not directly reflected in $F_p$.

If a single $F_p$ value would account for both dry and wet season, the effects of changing $F_p$ on low flows may well be more pronounced than those on flood risk. Empirical tests are needed of the dependence of $F_p$ on Q (see below). Analysis of the way an aggregate $F_p$ depends on the dominant flow pathways provides a basis for differentiating $F_p$ within a hydrologic year.

## 2.4 Flow-pathway dependence of flow persistence

The patch-level partitioning of water between infiltration and overland flow is further modified at hillslope level, with a common distinction between three pathways that reach streams: overland flow, interflow and groundwater flow (Band et al., 1993; Weiler and McDonnell, 2004). An additional interpretation of Eq.(1), potentially adding to our understanding of results but not needed for analysis of empirical data, can be that three pathways of water through a landscape contribute to river flow (Barnes, 1939): groundwater release with $F_{p,g}$ values close to 1.0, overland flow with $F_{p,o}$ values close to 0, and interflow with intermediate $F_{p,i}$ values.

$$Q_t = F_{p,g} Q_{t-1,g} + F_{p,i} Q_{t-1,i} + F_{p,o} Q_{t-1,o} + Q_{a,t} \tag{7},$$

$$F_p = (F_{p,g} Q_{t-1,g} + F_{p,i} Q_{t-1,i} + F_{p,o} Q_{t-1,o})/Q_{t-1} \tag{8}.$$

On this basis a decline or increase in overall weighted average $F_p$ can be interpreted as indicator of a shift of dominant runoff pathways through time within the watershed. Dry season flows are dominated by $F_{p,g}$. The effective $F_p$ in the rainy season can be interpreted as indicating the relative importance of the other two flow pathways. $F_p$ reflects the fractions of total river flow that are based on groundwater, overland flow and interflow pathways:

$$F_p = F_{p,g} (\Sigma Q_{t,g} / \Sigma Q_t) + F_{p,o} (\Sigma Q_{t,o} / \Sigma Q_t) + F_{p,i} (\Sigma Q_{t,i} / \Sigma Q_t) \tag{9}.$$

Beyond the type of degradation of the watershed that, mostly through soil compaction, leads to
enhanced infiltration-excess (or Hortonian) overland flow (Delfs et al., 2009), saturated conditions
throughout the soil profile may also induce overland flow, especially near valley bottoms (Bonell,
1993; Bruijnzeel, 2004). Thus, the value of $F_{p,o}$ can be substantially above zero if the rainfall has a
significant temporal autocorrelation, with heavy rainfall on subsequent days being more likely than
would be expected from general rainfall frequencies. If rainfall following a wet day is more likely to
occur than following a dry day, as is commonly observed in Markov chain analysis of rainfall patterns
(Jones and Thornton, 1997; Bardossy and Plate, 1991), the overland flow component of total flow
will also have a partial temporal autocorrelation, adding to the overall predictability of river flow. In
a hypothetical climate with evenly distributed rainfall, we can expect $F_p$ to be 1.0 even if there is no
infiltration and the only pathway available is overland flow. Even with rainfall that is variable at any
point of observation but has low spatial correlation it is possible to obtain $F_p$ values of (close to) 1.0
in a situation with (mostly) overland flow (Ranieri et al., 2004).
**2.5 Relationship between flow persistence and flashiness index**
The Richards-Baker 'R-B Flashiness index' (Baker et al. 2004) is defined as
$$FI = \sum_t |\Delta Q_t| / \sum_t Q_t = \sum_{ti} (Q_t - Q_{t-1}) + \sum_{td} (Q_{t-1} - Q_t) \qquad [10]$$
with *ti* indicating all times t that $Q_t > Q_{t-1}$ and *td* indicating all times t that $Q_t =< Q_{t-1}$. Over a
timeframe that flow has no net trend, the sum of increments ($\sum_{ti} (Q_t - Q_{t-1})$) is equal to the sum of
declines ($\sum_{td} (Q_{t-1} - Q_t)$).
Substituting equation [5] in [10] we obtain:
$$FI = 2(1-F_p)( 0.5\ \Delta S + \sum_{ti} (P_t - E_{tx} - Q_t)) / \sum_t Q_t = 2\ (1-F_p)(-0.5\ \Delta S + \sum_{td} (-P_t + E_{tx} + Q_t)) / \sum_t Q_t \qquad [11]$$
With $\Delta S$ representing change in catchment storage; $\Delta S = (1-F_p)\ (-\sum_{ti} (P_t - E_{tx} - Q_t) + \sum_{td} (-P_t + E_{tx} + Q_t))$.
This suggests that $FI = 2\ (1-F_p)$ is a first approximation and becomes zero for $F_p = 1$. These
approximations require that changes in the catchment have no influence on $P_t$ or $E_{tx}$ values. If $E_{tx}$ is
negatively affected (either by a change in vegetation or by insufficient buffering, reducing water
availability on non-rainfall days) flashiness will increase, beyond the main effects on $F_p$.
The rainfall term, counted positive for all days with flow increase and negatively for days with
declining flow, hints at one of the major reasons why the flashiness index tends to get smaller when
larger catchment areas are involved: rainfall will tend to get more evenly distributed over time,
unless the spatial correlation of rainfall is (close to) 1 and all rainfall derives from fronts passing over
the area uniformly. Where (part of) precipitation occurs as snow, the timing of snow melt defines $P_t$
as used here. Where vegetation influences timing and synchrony of snowmelt, this will be reflected
in the flashiness index. It may not directly influence flow persistence, but will be accounted for in the
flow description that uses flow persistence as key parameter.

## 3. Methods

### 3.1 River flow data for four tropical watersheds

To test the applicability of the $F_p$ metric and explore its properties, data from four Southeast Asian watersheds were used, that will be described and further analysed in part II. The first watershed data set is the Way Besai (414.4 km$^2$) in Lampung province, Sumatra, Indonesia (Verbist et al., 2010). With an elevation between 720-1831 m a.s.l., the Way Besai is dominated by various coffee production systems (64%), with remaining forest (18%), horticulture and crops (12%) and other land uses (6%). Daily rainfall data from 1976 – 2007, was generated by interpolation of eight rainfall stations using Thiessen polygons; data were obtained from BMKG (*Agency on Meteorology, Climatology and Geophysics*), PU (Public Work Agency) and PLN (*National Electricity Company*). The average of annual rainfall was 2474 mm, with observed values in the range 1216 – 3277 mm. River flow data at the outflow of the Way Besai was also obtained from PU and PUSAIR (*Centre for Research and Development on Water Resources),* with an average of river flow of 16.7 m$^3$/s.

Data from three other watersheds were used to explore the variation of $F_p$ across multiple years and its relationship with the Flashiness Index: Bialo (111.7 km$^2$) in South Sulawesi, Indonesia with Agroforestry as the dominant land cover type, Cidanau (241.6 km$^2$) in West Java, Indonesia, dominated by mixed Agroforestry land uses but with a peat swamp before the final outlet and Mae Chaem (3892 km$^2$) in Northern Thailand, part of the upper Ping Basin, and dominated by evergreen, deciduous and pine forest. Detailed information on these watersheds and the data sources is provided in Paper II.

### 3.2 Numerical examples

For visualizing the effects of stochastic rainfall on river flow according to equation [1] a spreadsheet model that is available from the authors on request was used in 'Monte Carlo' simulations. Fixed values for $F_p$ were used in combination with a stochastic $Q_{a,t}$ value. The latter was obtained from a random generator (rand) with two settings for a (truncated) sinus-based daily rainfall probability: A) one for situations that have approximately 120 rainy days, and an annual Q of around 1600 mm, and B) one that leads to around 45 rainy days and an annual total around 600 mm. Maximum daily $Q_{a,t}$ was chosen as 60 mm in both cases. For the figures, realizations for various $F_p$ values were retained that were within 10% of this number of rainy days and annual flow total, to focus on the effects of $F_p$ as such.

### 3.3 Flow persistence as a simple flood risk indicator

For numerical examples (implemented in a spreadsheet model) flow on each day can be derived as:

$$Q_t = \Sigma_j^t \, F_p^{t-j} \, (1-F_p) \, p_j P_j \qquad\qquad [12].$$

Where $p_j$ reflects the occurrence of rain on day j (reflecting a truncated sine distribution for seasonal trends) and $P_j$ is the rain depth (drawn from a uniform distribution). From this model the effects of $F_p$ (and hence of changes in $F_p$) on maximum daily flow rates, plus maximum flow totals assessed over a 2-5 days period, was obtained in a Monte Carlo process (without Markov autocorrelation of rainfall in the default case – see below). Relative flood protection was calculated as the difference between

peak flows (assessed for 1-5 days duration after a 1 year 'warm-up' period) for a given $F_p$ versus
those for $F_p = 0$, relative to those at $F_p = 0$.

**3.4 An algorithm for deriving $F_p$ from a time series of stream flow data**

Equation (3) provides a first method to derive $F_p$ from empirical data if these cover a full hydrologic
year. In situations where there is no complete hydrograph and/or in situations where we want to
quantify $F_p$ for shorter time periods (e.g. to characterise intraseasonal flow patterns) and the change
in the storage term of the water budget equation cannot be ignored, we need an algorithm for
estimating $F_p$ from a series of daily $Q_t$ observations.
Where rainfall has clear seasonality, it is attractive and indeed common practice to derive a
groundwater recession rate from a semi-logarithmic plot of Q against time (Tallaksen, 1995). As we
can assume for such periods that $Q_{a,t} = 0$, we obtain $F_p = Q_t /Q_{t-1}$, under these circumstances. We
cannot be sure, however, that this $F_{p,g}$ estimate also applies in the rainy season, because overall wet-
season $F_p$ will include contributions by $F_{p,o}$ and $F_{p,i}$ as well (compare Eq. 9). In locations without a
distinct dry season, we need an alternative method.
A biplot of $Q_t$ against $Q_{t-1}$ will lead to a scatter of points above a line with slope $F_p$, with points above
the line reflecting the contributions of $Q_{a,t} > 0$, while the points that plot on the $F_p$ line itself
represent $Q_{a,t} = 0$ mm d$^{-1}$. There is no independent source of information on the frequency at which
$Q_{a,t} = 0$, nor what the statistical distribution of $Q_{a,t}$ values is if it is non-zero. Calculating back from the
$Q_t$ series we can obtain an estimate ($Q_{a,Fptry}$) of $Q_{a,t}$ for any given estimate ($F_{p,try}$) of $F_p$, and select the
most plausible $F_p$ value. For high $F_{p,try}$ estimates there will be many negative $Q_{a,t,Fptry}$ values, for low
$F_{p,try}$ estimates all $Q_{a,t,Fptry}$ values will be larger. An algorithm to derive a plausible $F_p$ estimate can
thus make use of the corresponding distribution of 'apparent $Q_a$' values as estimates of $F_{p,try}$,
calculated as $Q_{a,t,Fptry} = Q_t - F_{p,try} Q_{t-1}$. While $Q_{a,t}$ cannot be negative in theory, small negative $Q_a$
estimates are likely when using real-world data with their inherent errors. The FlowPer $F_p$ algorithm
(van Noordwijk et al., 2011) derives the distribution of $Q_{a,t,Fptry}$ estimates for a range of $F_{p,try}$ values
(Figure 3B) and selects the value $F_{p,try}$ that minimizes the variance $Var(Q_{a,t,Fptry})$ (or its standard
deviation) (Figure 3C). It is implemented in a spreadsheet workbook that can be downloaded from
the ICRAF website  ([http://www.worldAgroforestry.org/output/flowper-flow-persistence-model](http://www.worldAgroforestry.org/output/flowper-flow-persistence-model))
➔Figure 3
A consistency test is needed that the high-end $Q_t$ values relate to $Q_{t+1}$ in the same was as do low or
medium $Q_t$ values. Visual inspection of $Q_{t+1}$ versus $Q_t$, with the derived $F_p$ value, provides a
qualitative view of the validity of this assumption. The $F_p$ algorithm can be applied to any population
of ($Q_{t-1}, Q_t$) pairs, e.g. selected from a multiyear data set on the basis of 3-month periods within the
hydrological year.

**3.5 Flashiness and flow separation**

Hydrographs analysed for $F_p$ were also used for calculating the Richards-Baker or R-B Flashiness
index (Baker et al. 2004) by summing the absolute values of all daily changes in flow. Two common
flow separation algorithms (fixed and sliding interval methods, Furey and Gupta, 2001) were used to
estimate the base flow fraction at an annual basis. The average of the two was compared to $F_p$.

# 4 Results

## 4.1 Numerical examples


Figure 4 provides two examples, for annual river flows of around 1600 and 600 mm y$^{-1}$, of the way a
change in $F_p$ values (based on Eq. 1) influences the pattern of river flow for a unimodal rainfall
regime with a well-developed dry season. The increasing 'spikiness' of the graph as $F_p$ is lowered,
regardless of annual flow, indicates reduced predictability of flow on any given day during the wet
season on the basis of the flow on the preceding day.
⇨ Figure 4
A bi-plot of river flow on subsequent days for the same simulations (Figure 5) shows two main
effects of reducing the $F_p$ value: the scatter increases, and the slope of the lower envelope
containing the swarm of points is lowered (as it equals $F_p$). Both of these changes can provide entry
points for an algorithm to estimate $F_p$ from empirical time series, provided the basic assumptions of
the simple model apply and the data are of acceptable quality.
⇨ Figure 5
For the numerical examples shown in Figure 4, the relative increase of the maximum daily flow when
the $F_p$ value decreased from a value close to 1 (0.98) to nearly 0 depended on the rainfall regime;
with lower annual rainfall but the same maximum daily rainfall, the response of peak flows to
decrease in $F_p$ became stronger.

## 4.2 Flood intensity and duration


Figure 6 shows the effect of $F_p$ values in the range 0 to 1 on the maximum flows obtained with a
random time series of 'effective rainfall', compared to results for $F_p = 0$. Maximum flows were
considered at time scales of 1 to 5 days, in a moving average routine. This way a relative flood
protection, expressed as reduction of peak flow, could be related to $F_p$ (Figure 6A).
⇨ Figure 6
Relative flood protection rapidly decreased from its theoretical value of 100% at $F_p = 1$ (when there
was no variation in river flow), to less than 10% at $F_p$ values of around 0.5. Relative flood protection
was slightly lower when the assessment period was increased from 1 to 5 days (between 1 and 3
days it decreased by 6.2%, from 3 to 5 days by a further 1.3%). Two counteracting effects are at play
here: a lower $F_p$ means that a larger fraction $(1-F_p)$ of the effective rainfall contributes to river flow,
but the increased flow is less persistent. In the example the flood protection in situations where the
rainfall during 1 or 2 days causes the peak is slightly stronger than where the cumulative rainfall over
3-5 days causes floods, as typically occurs downstream.
As we expect from equation 5 that peak flow is to $(1-F_p)$ times peak rainfall amounts, the effect of a
change in $F_p$ not only depends on the change in $F_p$ that we are considering, but also on its initial
value. Higher initial $F_p$ values will lead to more rapid increases in high flows for the same reduction in
$F_p$ (Figure 6B). However, flood duration rather responds to changes in $F_p$ in a curvilinear manner, as
flow persistence implies flood persistence (once flooding occurs), but the greater the flow
persistence the less likely such a flooding threshold is passed (Figure 6C). The combined effect may
be restricted to about 3 days of increase in flood duration for the parameter values used in the
default example, but for different parametrization of the stochastic ε other results might be
obtained.

## 4.3 Algorithm for $F_p$ estimates from river flow time series

The algorithm has so far returned non-ambiguous $F_p$ estimates on any modelled time series data of
river flow, as well as for all empirical data set we tested (including all examples tested in part II),
although there probably are data sets on which it can breakdown. Visual inspection of $Q_{t-1}/Q_t$ biplots
(as in Figure 4) can provide clues to non-homogenous data sets, to potential situations where
effective $F_p$ depends on flow level $Q_t$ and where data are not consistent with a straight-line lower
envelope. Where river flow estimates were derived from a model with random elements, however,
variation in $F_p$ estimates was observed, that suggests that specific aspects of actual rainfall, beyond
the basic characteristics of a watershed and its vegetation, do have at least some effect. Such effects
deserve to be further explored for a set of case studies, as their strength probably depends on
context.

## 4.4 Flow persistence compared to base flow and flashiness index

Figure 7 compares results for a hydrograph of a single year for the Way Besai catchment, described
in more detail in paper II. While there is agreement on most of what is indicated as baseflow, the
short term response to peaks in the flow differ, with baseflow in the $F_p$ method more rapidly
increasing after peak events.

⇨  Figure 7

When compared across multiple years for four Southeast Asian catchments (figure 8A), there is
partial agreement in the way interannual variation is described in each catchment, while numerical
values are similar. However, the ratio of what is indicated as baseflow according to the $F_p$ method
and according to standard hydrograph separation varies from 1.05 to 0.86.

⇨  Figure 8

Figure 8 compares numerical results for the R-B Flashiness Index with $F_p$ for the four test catchments
and for a number of hydrographs constructed as in Fig. 3A. The two concepts are inversely related,
as expected from equation [11], but where $F_p$ is constrained to the 0-1 interval, the R-B Flashiness
Index can attain values up to 2.0, with the value for $F_p = 0$ depending on properties of the local
rainfall regime. Where hydrographs were generated with a simple flow model with $F_p$ parameter as
key variable, the flashiness index is more tightly related to, especially for higher $F_p$ values, than
where both flashiness index and $F_p$ were derived from existing flow data (Figure 8C versus 8B). The
difference in slope between the four watersheds in Fig. 8B appears to be primarily related to aspects
of the local rainfall pattern that deserve further analysis in larger data sets of this nature.

# 5 Discussion

We will discuss the flow persistence metric based on the seven questions raised from the
perspectives of salience, credibility and legitimacy and refer back to figure 2 that clarified how
ecosystem structure, ecosystem function and human land use interact in causal loops that can lead
to flood damage, its control and/or prevention.

**5.1 Salience**

Key *salience* aspects are "Does flow persistence relate to important aspects of watershed
behaviour?" and "Does it help to select management actions?". A major finding in the derivation of
$F_p$ was that the flow persistence measured at daily time scale can be logically linked to the long-term
water balance under the assumption that the watershed is defined on the basis of actual
groundwater flows, and that the proportion of peak rainfall that translates to peak river flow equals
the complement of flow persistence. This feature links effects on floods of changes in watershed
quality, as commonly expressed in curve numbers and flashiness indices, to effects on low flows, as
commonly expressed in base flow metrics. The $F_p$ parameter as such does not predict when and
where flooding will occur, but it does help to assess to what extent another condition of the
watershed, with either higher or lower $F_p$ would translate the same rainfall into larger or small peak
water flows. This is salient, especially if the relative contributions of (anthropogenic) land cover and
the (exogenous, probabilistic) specifics of the rainfall pattern can be further teased apart (see part
II). Where $F_p$ may describe the descending branch of hydrographs at a relevant time scale, details of
the ascending branch beyond the maximum daily flow reached may be relevant for reducing flood
damage, and may require more detailed study at higher temporal resolution.
Figures 3 and 6 show that most of the effects of a decreasing $F_p$ value on peak discharge (which is
the basis for downstream flooding) occur between $F_p$ values of 1 and 0.7, with the relative flood
protection value reduced to 10% when $F_p$ reaches 0.5. As indicated in Figure 2, peak discharge is only
one of the factors contributing to flood risk in terms of human casualties and physical damage. Flood
risks are themselves nonlinearly and in strongly topography-specific ways related to the volume of
river flow after extreme rainfall events. While the expected fraction of rainfall that contributes to
direct flow is linearly related to rainfall via (1-Fp), flooding risk as such will have a non-linear
relationship with rainfall, that depends on topography and antecedent rainfall. Catchment changes,
such as increases or decreases in percentage tree cover, will generally have a non-linear relationship
with $F_p$ as well as with flooding risks. The $F_p$ value has an inverse effect on the fraction of recent
rainfall that becomes river flow, but the effect on peak flows is less, as higher $F_p$ values imply higher
base flow. The way these counteracting effects balance out depends on details of the local rainfall
pattern (including its Markov chain temporal autocorrelation), as well as the downstream
topography and risk of people being at the wrong time at a given place, but the $F_p$ value is an
efficient way of summarizing complex land use mosaics and upstream topography in its effect on
river flow. The difference between wet-season and dry-season $F_p$ deserves further analysis. In
climates with a real rainless dry-season, dry season $F_p$ is dominated by the groundwater release
fraction of the watershed, regardless of land cover, while in wet season it depends on the mix
(weighted average) of flow pathways. The degree to which $F_p$ can be influenced by land cover needs
to be assessed for each landscape and land cover combination, including the locally relevant forest
and forest derived land classes, with their effects on interception, soil infiltration and time pattern of
transpiration. The $F_p$ value can summarize results of models that explore land use change scenarios
in local context. To select the specific management actions that will maintain or increase $F_p$ a locally
calibrated land use/hydrology model is needed, such as GenRiver (part II), DHV (Bergström, 1995) or
SWAT (Yen et al., 2015).
The 'health' wording has been used as a comprehensive concept of the way a) climate forcing, b)
watershed vegetation and soil conditions and c) engineering interventions interact on functional
aspects of river flow. Ma et al (2014) described a method to separate these three influences on river
flow. In the four catchments we used as example there have been no major dams or reservoirs
installed upstream of the points of measurement. Where these do exist the specific operating rules
of reservoirs need to be included in any model and these can have a major influence on downstream
flow, depending on the primary use for power generation, dry season irrigation or stabilizing river
flow for riverine transport. Although a higher $F_p$ value will in most cases be desirable (and a decrease
in $F_p$ undesirable), we may expect that In an ecological perspective on watershed health, the change
in low flows that can occur in the flow regime of degrading and intensively managed watersheds
alike, depending on the management rules for reservoirs, is at least as relevant as changes in flood
risks, as many aquatic organisms thrive during floods (Pahl-Wostl et al., 2013; Poff et al., 2010).
Downstream biota can be expected to have adapted to the pre-human flow conditions, inherent $F_p$
and variability. Decreased variability of flow achieved by engineering interventions (e.g. a reservoir
with constant release of water to generate hydropower) may have negative consequences for fish
and other biota (Richter et al., 2003; McCluney et al., 2014). In an extensive literature review Poff
and Zimmerman (2010) found no general, transferable quantitative relationships between flow
alteration and ecological response, but the risk of ecological change increases with increasing
magnitude of flow alteration.
Various geographically defined watershed health concepts are in use (see for example
https://www.epa.gov/hwp/healthy-watersheds-projects-region-5; City of Fort Collins, 2015,
employing a range of specific indicators, including the 'R-B flashiness index' (Baker et al. 2004). The
definition of watershed health, like that of human health has evolved over time. Human health was
seen as a state of normal function that could be disrupted from time to time by disease. In 1948 the
World Health Organization (1958) proposed a definition that aimed higher, linking health to well-
being, in terms of physical, mental, and social aspects, and not merely the absence of disease and
infirmity. Health became seen as the ability to maintain homeostasis and recover from injury, but
remained embedded in the environment in which humans function.
**5.2 Credibility**
Key *credibility* questions are "Consistency of numerical results?" and "How sensitive are results to
bias and random error in data sources?". A key strength of our flow persistence parameter, that it
can be derived from a limited number of observations of river flow at a single point along the river,
without knowledge of rainfall events and catchment conditions, is also its major weakness. If rainfall
data exist, and especially rainfall data that apply to each subcatchment, the $Q_a$ term doesn't have to
be treated as a random variable and event-specific information on the flow pathways may be
inferred for a more precise account of the hydrograph. But for the vast majority of rivers in the
tropics, advances in remotely sensed rainfall data are needed to achieve that situation and $F_p$ may be
all that is available to inform public debates on the location-specific relation between forests and
floods.
The main conclusions from the numerical examples analysed so far are that intra-annual variability
of $F_p$ values between wet and dry seasons was around 0.2, interannual variability in either annual or
seasonal $F_p$ was generally in the 0.1 range, while the difference between observed and simulated
flow data as basis for $F_p$ calculations was mostly less than 0.1. With current methods, it seems that
effects of land cover change on flow persistence that shift the $F_p$ value by about 0.1 are the limit of
what can be  asserted from empirical data (with shifts of that order in a single year a warning sign
rather than a firmly established change). When derived from observed river flow data $F_p$ is suitable
for monitoring change (degradation, restoration) and can be a serious candidate for monitoring
performance in outcome-based ecosystem service management contracts. In interpreting changes in
$F_p$ as caused by changes in the condition in the watershed, however, changes in specific properties of
the rainfall regime must be excluded. At the scale of paired catchment studies this assumption may
be reasonable, but in temporal change (or using specific events as starting point for analysis), it is
not easy to disentangle interacting effects (Ma et al., 2014). Recent evidence that vegetation not
only responds to, but also influences rainfall (arrow 10 in Figure 2; van Noordwijk et al., 2015b)
further complicates the analysis across scales.
As indicated, the $F_p$ method is related to earlier methods used in streamflow hydrograph separation
of base flow and quick flow. While textbooks (Ward and Robinson, 2000; Hornberger et al 2014)
tend to be critical of the lack of objectivity of graphical methods, algorithms are used for deriving the
minimum flow in a fixed or sliding period of reference as base flow (Sloto and Crouse, 1996; Furey
and Gupta, 2001). The time interval used for deriving the minimum flow depends on catchment size.
Recursive models that describe flow in a next time interval on the basis of a fraction of that in the
preceding time interval with a term for additional flow due to additional rainfall have been used in
analysis of peak flow event before, with time intervals as short as 1 minute rather than the 1 day we
use here (Rose, 2004). Through reference to an overall mass balance a relationship similar to what
we found here ($F_p$ times preceding flow plus $1 - F_p$ times recent inputs) was also used in such
models. To our knowledge, the method we describe here at daily timescales has not been used
before.
The idea that the form of the storage-discharge function can be estimated from analysis of
streamflow fluctuations has been explored before for a class of catchments in which discharge is
determined by the volume of water in storage (Kirchner, 2009). Such catchments behave as simple
first-order nonlinear dynamical systems and can be characterized in a single-equation rainfall-runoff
model that predicted streamflow, in a test catchment in Wales, as accurately as other models that
are much more highly parameterized. This model of the dQ/dt versus Q relationship can also be
analytically inverted; thus, it can, according to Kirchner (2009), be used to "do hydrology backward,"
that is, to infer time series of whole-catchment precipitation directly from fluctuations in
streamflow. The slope of the log-log relationship between flow recession (dQ/dt) and Q that
Kirchner (2009) used is conceptually similar to the $F_p$ metric we derived here, but the specific
algorithm to derive the parameter from empirical data differs. Further exploration of the underlying
assumptions is needed. Estimates of dQ/dt are sensitive to noise in the measurement of Q and the
possibly frequent and small increases in Q can be separated from the expected flow recession in the
algorithm we presented here.
Table 1 compares a number of properties (Salience and Legitimacy in properties 1-4, Credibility
dimensions in 5-10) for the R-B Flashiness Index (Baker et al. 2004) and flow persistence. The main
advantage of continuing with the flashiness index is that there is an empirical basis for comparisons
and the index has been included in existing 'watershed health' monitoring programs, especially in
the USA. The main advantage of including $F_p$ is that it can be estimated from incomplete flow
records, has a clear link to peak flow events and has a more direct relationship with underlying flow
pathways, changes in rainfall (or snowmelt) and evapotranspiration, reflecting land cover change.
➜ Table 1
Seibert and Beven (2009) discussed the increase in predictive skill of models depending on the
amount of location-specific data that can be used to constrain them. They found that the ensemble
prediction of multiple models for a single location clearly outperformed the predictions using single
parameter sets and that surprisingly little runoff data was necessary to identify model
parameterizations that provided good results for 'ungauged' test periods in cases where actual
measurements were available. Their results indicated that a few runoff measurements can contain
much of the information content of continuous runoff time series. The way these conclusions might
be modified if continuous measurements for limited time periods, rather than separated single data
points on river flow could be used, remains to be explored. Their study indicated that results may
differ significantly between catchments and critical tests of $F_p$ across multiple situations are
obviously needed, as paper II will provide.
In discussions and models of temperate zone hydrology (Bergström, 1995; Seibert, 1999) snowmelt
is a major component of river flow and effects of forest cover on spring temperatures are important
to the buffering of the annual peaks in flow that tend to occur in this season. Application of the $F_p$
method to data describing such events has yet to be done.
**5.3 Legitimacy**
*Legitimacy* aspects are "Does it match local knowledge?" and "Can it be used to empower local
stakeholders of watershed management?" and "Can it inform risk management?". As the $F_p$
parameter captures the predictability of river flow that is a key aspect of degradation according to
local knowledge systems, its results are much easier to convey than full hydrographs or exceedance
probabilities of flood levels. By focusing on observable effects at river level, rather than prescriptive
recipes for land cover ("Reforestation"), the $F_p$ parameter can be used to more effectively compare
the combined effects of land cover change, changes in the riparian wetlands and engineered water
storage reservoirs, in their effect on flow buffering. It is a candidate for shifting environmental
service reward contracts from input to outcome based monitoring (van Noordwijk et al., 2012).  As
such it can be used as part of a negotiation support approach to natural resources management in
which  levelling off on knowledge and joint fact finding in blame attribution are key steps to
negotiated solutions that are legitimate and seen to be so (van Noordwijk et al., 2013; Leimona et
al., 2015). Quantification of $F_p$ can help assess tactical management options (Burt et al., 2014) as in a
recent suggestion to minimize negative downstream impacts of forestry operations on stream flow
by avoiding land clearing and planting operations in locally wet La Niña years. But the most
challenging aspect of the management of flood, as any other environmental risk, is that the
frequency of disasters is too low to intuitively influence human behaviour where short-term risk
taking benefits are attractive. Wider social pressure is needed for investment in watershed health
(as a type of insurance premium) to be mainstreamed, as individuals waiting to see evidence of
necessity are too late to respond. In terms of flooding risk, actions to restore or retain watershed
health can be similarly justified as insurance premium. It remains to be seen whether or not the
transparency of the $F_p$ metric and its intuitive appeal are sufficient to make the case in public debate
when opportunity costs of foregoing reductions in flow buffering by profitable land use are to be
compensated and shared (Burt et al., 2014).

**6 Conclusions**

In conclusion, the $F_p$ metric appears to allow an efficient way of summarizing complex landscape
processes into a single parameter that reflects the effects of landscape management within the
context of the local climate. If rainfall patterns change but the landscape does not, the resultant flow
patterns may reflect a change in watershed health (van Noordwijk et al., 2016). Flow persistence is
the result of rainfall persistence and the temporal delay provided by the pathway water takes
through the soil and the river system. High flow persistence indicates a reliable water supply, while
minimizing peak flow events. Wider tests of the $F_p$ metric as boundary object in science-practice-
policy boundary chains (Kirchhoff et al., 2015; Leimona et al., 2015) are needed. Further tests for
specific case studies can clarify how changes in tree cover (deforestation, reforestation and
agroforestation) in different contexts influence river flow dynamics and $F_p$ values. Sensitivity to
specific realizations of underlying time-space rainfall patterns needs to be quantified, before
changes in $F_p$ can be attributed to changed 'watershed health', rather than chance events.

# Author contributions

Meine van Noordwijk designed method and paper, Lisa Tanika refined the empirical algorithm and
handled the case study data and modelling for part II, and Betha Lusiana contributed statistical
analysis; all contributed and approved the final manuscript

# Acknowledgements

This research is part of the Forests, Trees and Agroforestry research program of the CGIAR. Several
colleagues contributed to the development and early tests of the $F_p$ method. Thanks are due to Eike
Luedeling, Sonya Dewi, Sampurno Bruijnzeel and three anonymous reviewers for comments on an
earlier version of the manuscript.

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

**Figures:**

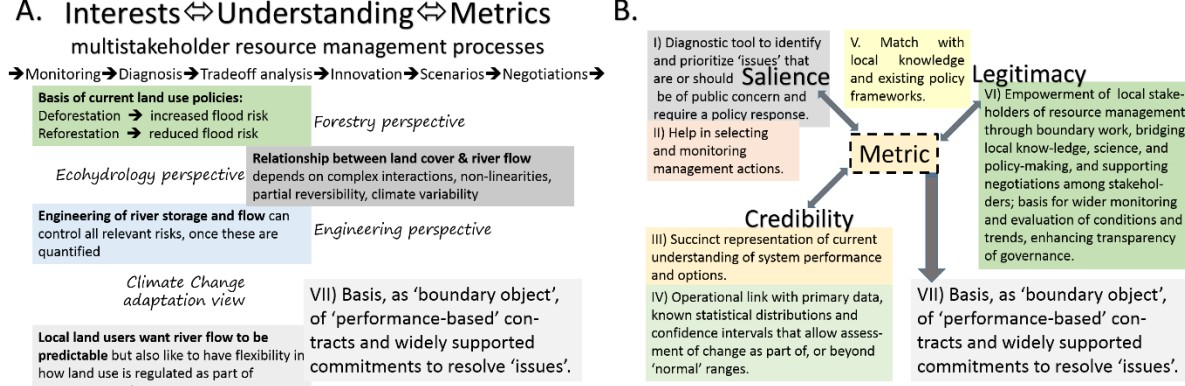


Figure 1. A. Multiple perspectives on the way flood risk is to be understood, monitored and handled
according to different knowledge systems; B. Basic requirements for a 'metric' to be used in public
discussions of natural resource management issues that deserve to be resolved and acted upon
(modified from van Noordwijk et al., 2016)

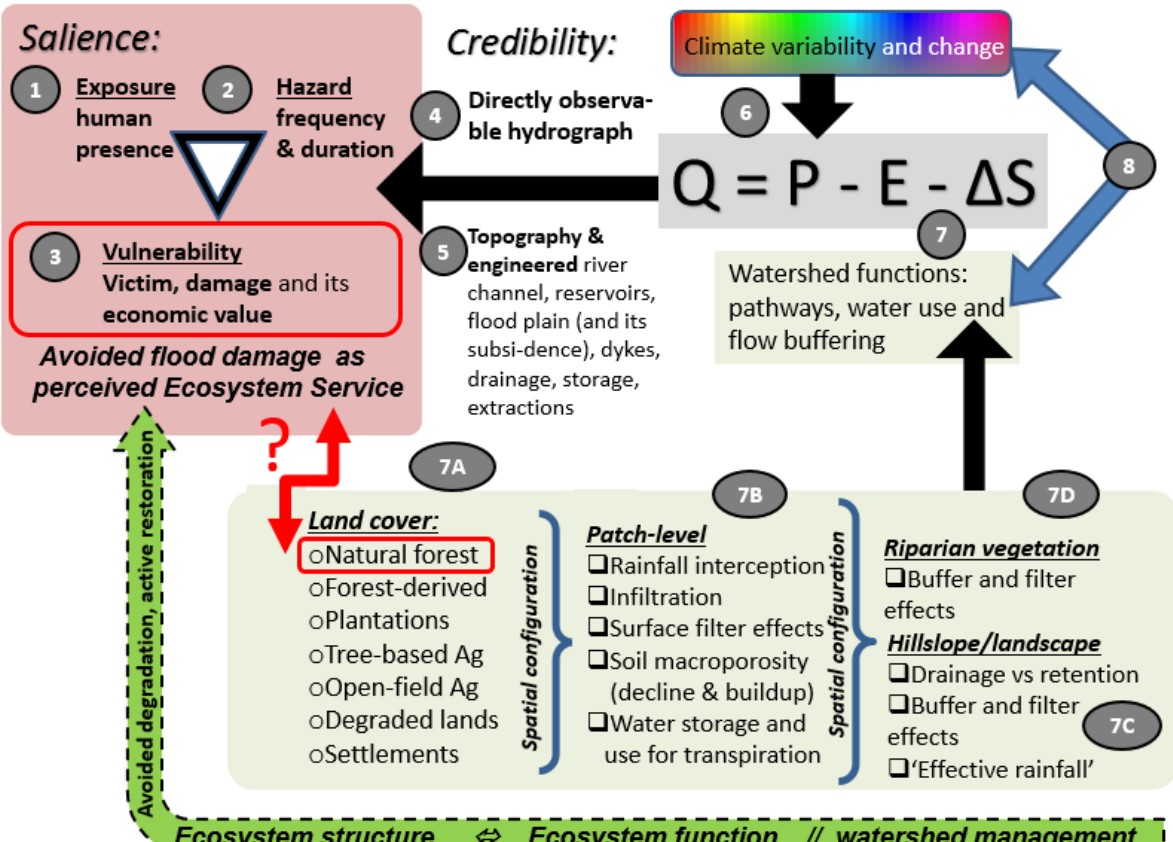


Figure 2. Steps in a causal pathway that relates the salience of 'avoided flood damage as
ecosystem service' to the interaction of exposure (1; being in the wrong place at critical
times), hazard (2; spatially explicit flood frequency and duration) and human determinants
of vulnerability (3); the hazard component depends, in common scientific analysis, on the
pattern of river flow described in a hydrograph (4), which in turn is understood to be
influenced by conditions along the river channel (5), precipitation and potential
evapotranspiration ($E_{pot}$ as climatic factors (6) and the condition in the watershed (7)
determining evapotranspiration ($E_{act}$), temporary water storage ($\Delta S$) and water partitioning
over overland flow and infiltration; these watershed functions in turn depend on the
interaction of terrain (topography, soils, geology), vegetation and human land use; current
understanding of a two-way interaction between vegetation and rainfall adds further
complexity (8)

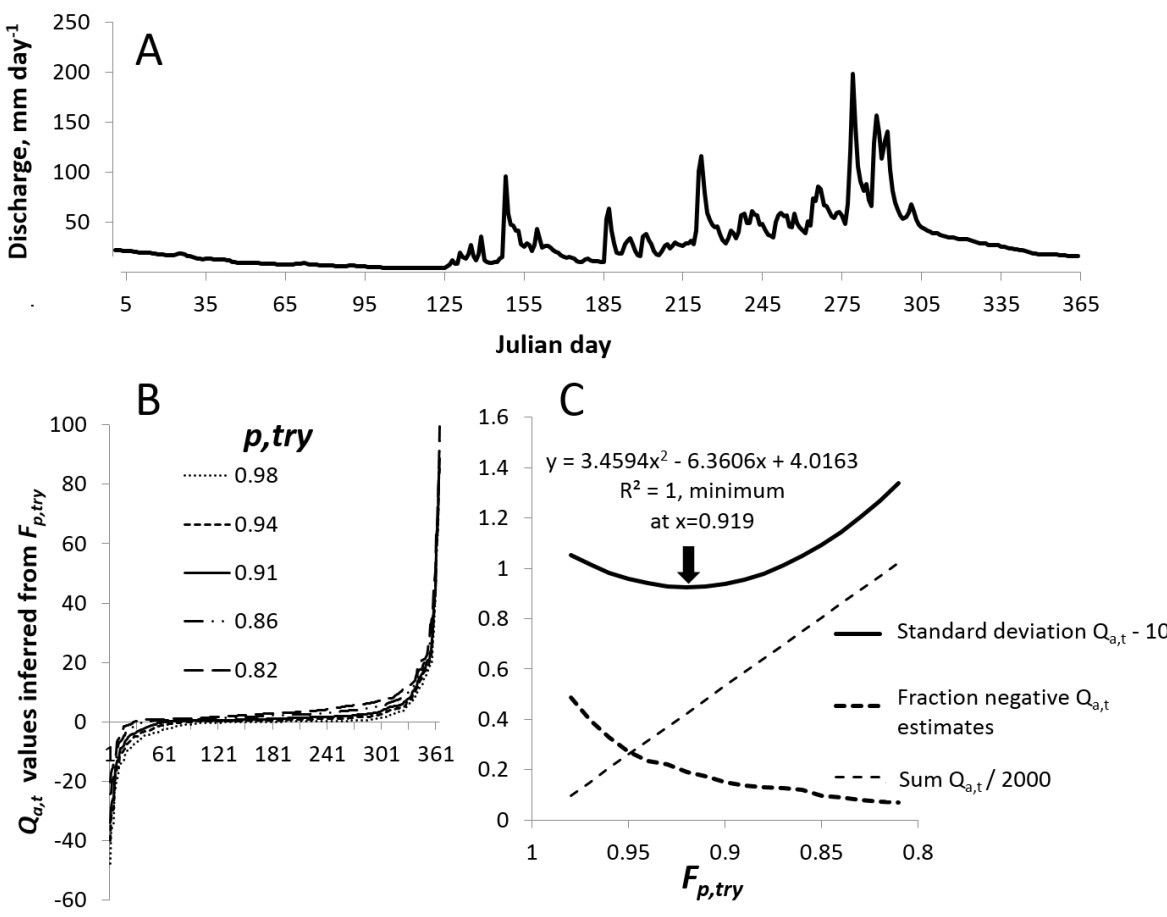


Figure 3. Example of the derivation of best fitting $F_{p,try}$ value for an example hydrograph (A) on the

basis of the inferred $Q_a$ distribution (cumulative frequency in B), and three properties of this

distribution (C): its sum, frequency of negative values and standard deviation; the $F_{p,try}$ minimum

of the latter is derived from the parameters of a fitted quadratic equation


**A.** 120 rainy days, Discharge ~ 1600 mm year$^{-1}$

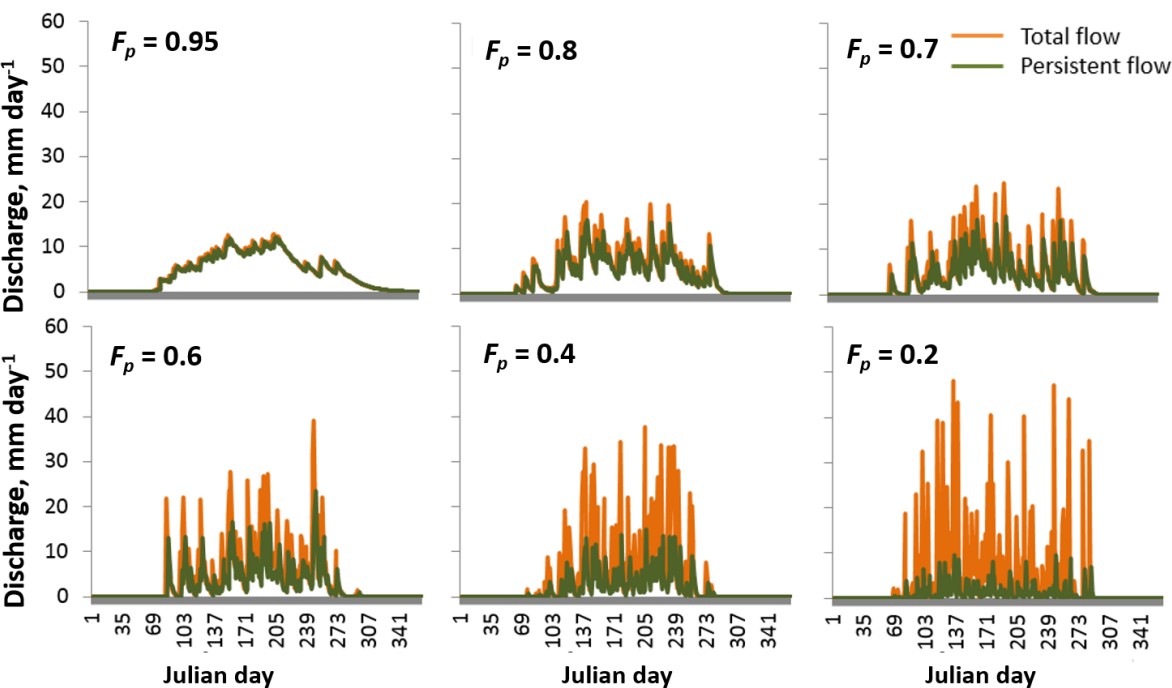

**B.** 45 rainy days, Discharge ~ 600 mm year$^{-1}$

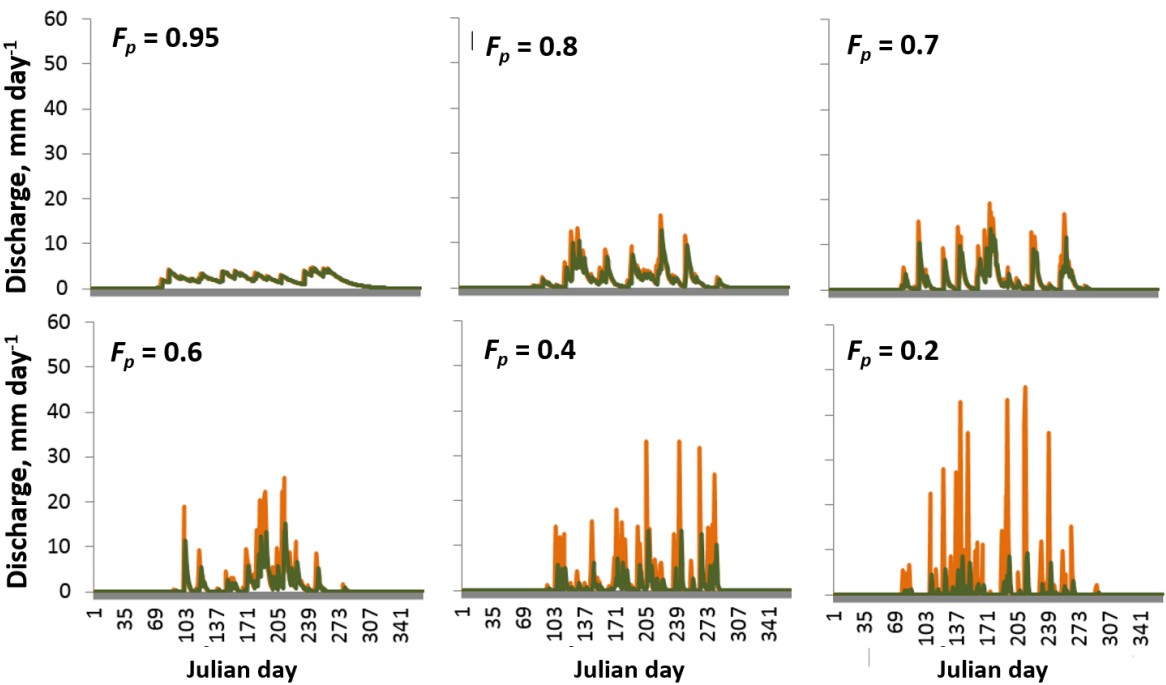


Figure 4. Effects of the $F_p$ parameter on hydrographs of daily river flow generated by a random
rainfall generator, with persistent and additional flow components indicated, for two settings
with total rainfall of approximately 1600 and 600 mm/yr (NB river flow is here expressed as mm
d$^{-1}$ rather than as m$^3$ s$^{-1}$ as in figure 3)


**A.** 120 rainy days, Discharge ~ 1600 mm year$^{-1}$

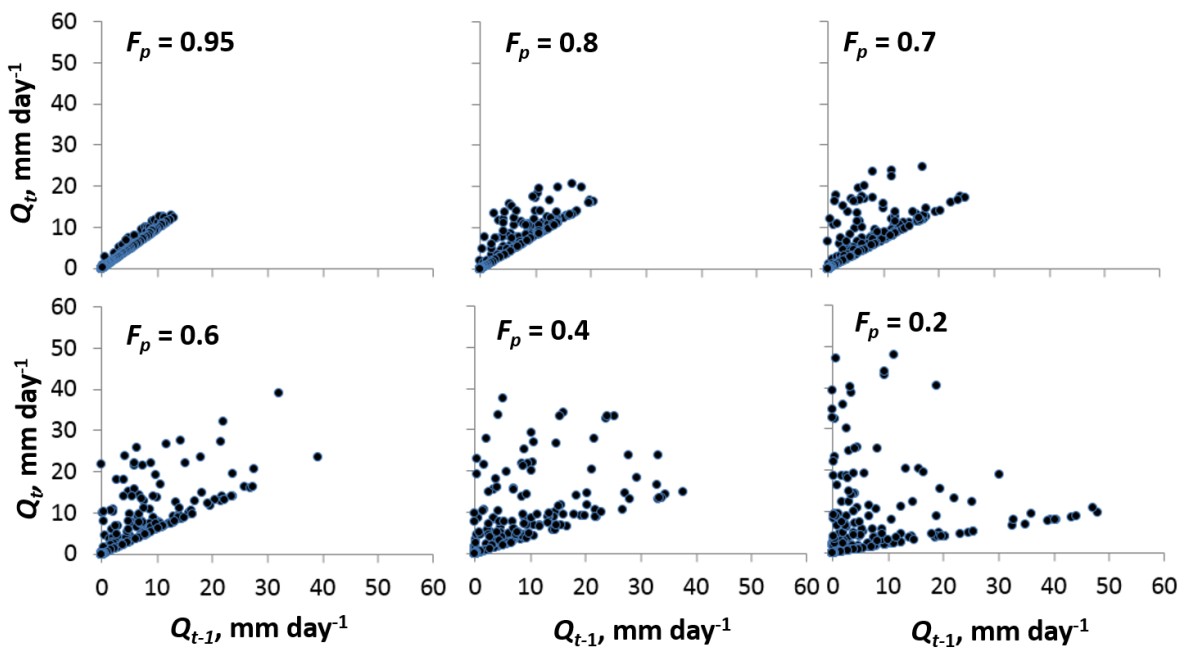

**B.** 45 rainy days, Discharge ~ 600 mm year$^{-1}$

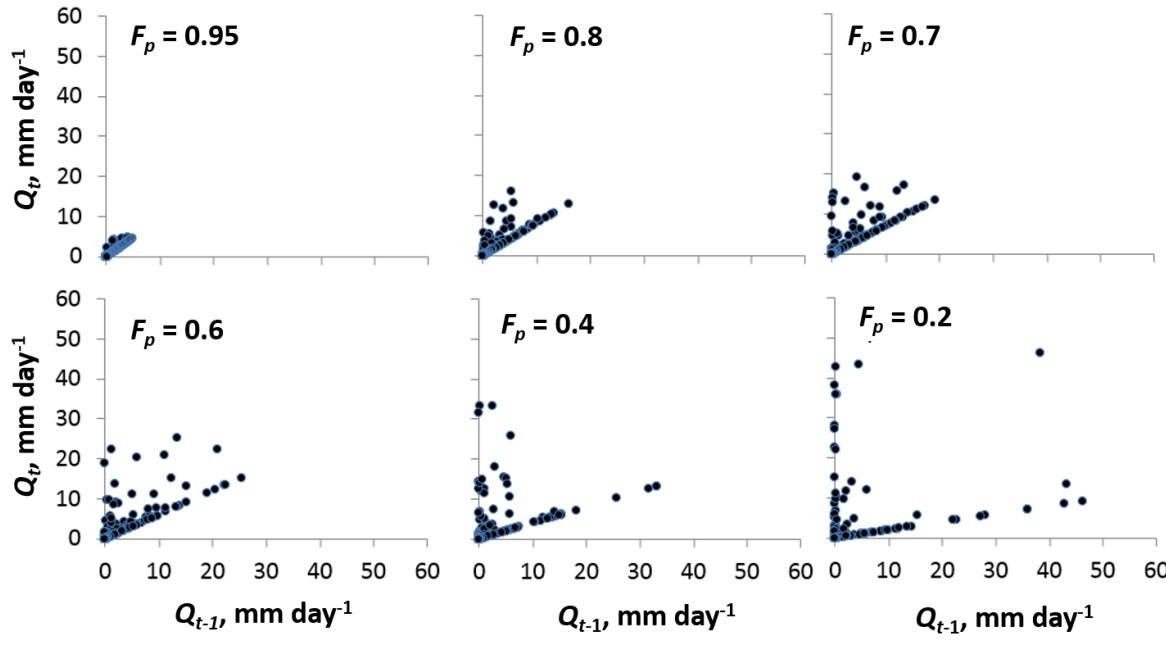


Figure 5 A and B Temporal autocorrelation of river flow for the same simulations as Figure 4; the
lower envelope of the points indicated slope $F_p$, the points above this line the effect of fresh
additions to river flow


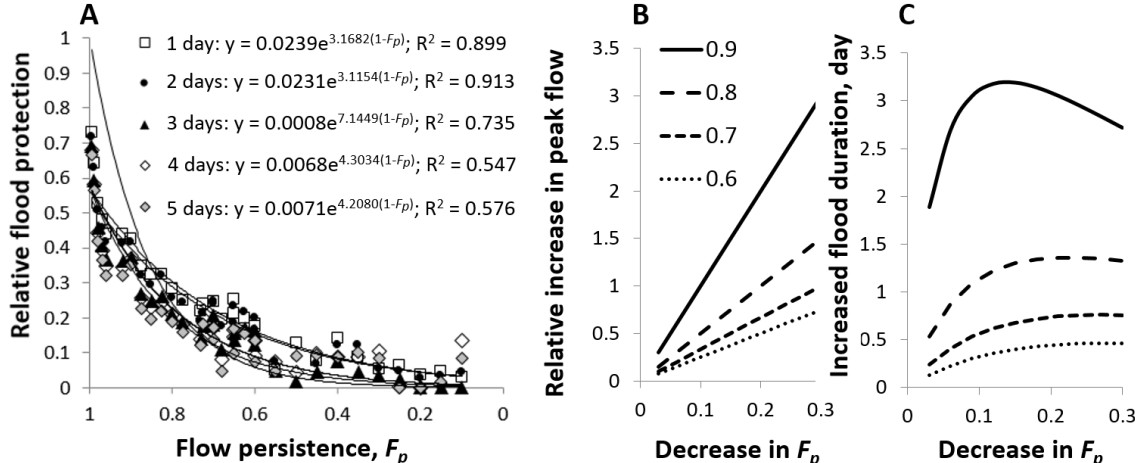

Figure 6. A. Effects of flow persistence on the relative flood protection (decrease in
maximum flow measured over a 1 – 5 d period relative to a case with $F_p$ = 0 (a few small
negative points were replaced by small positive values to allow the exponential fit); B and
C. effects of a decrease in flow persistence on the volume of water involved in peak flows
(B; relative to the volume at $F_p$ is 0.6 – 0.9) and in the duration (in d) of floods (C)


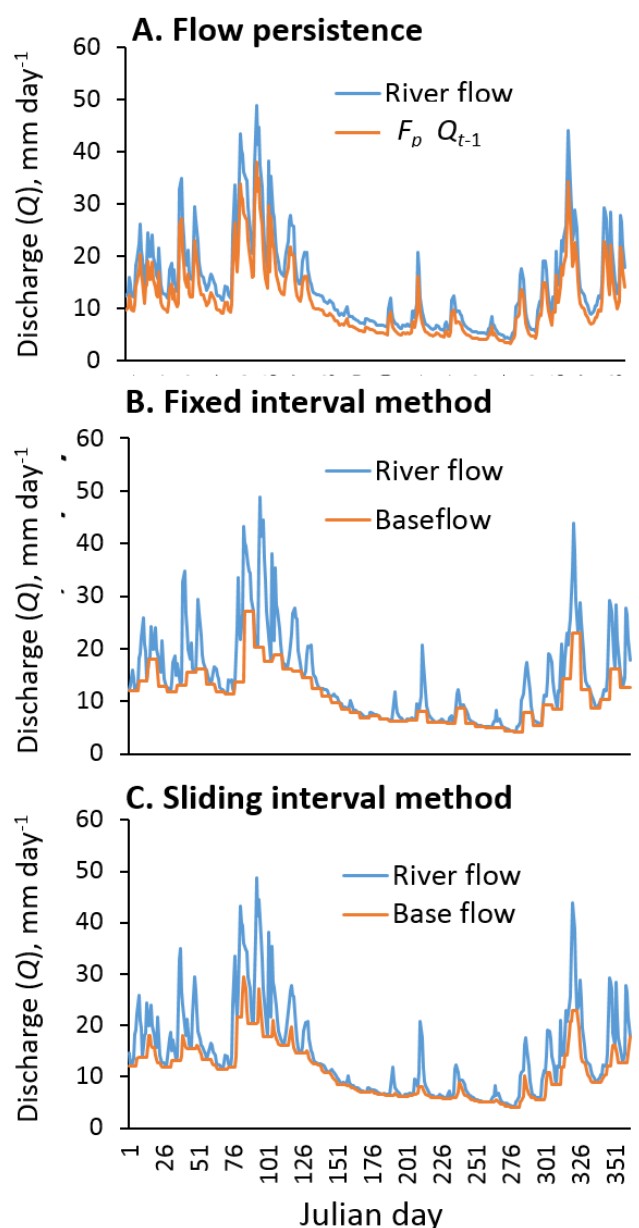

Figure 7. Comparison of base flow separation of a hydrograph according to the flow
persistence method (A) and two common flow separation methods, respectively with
fixed (B) and sliding intervals (C)

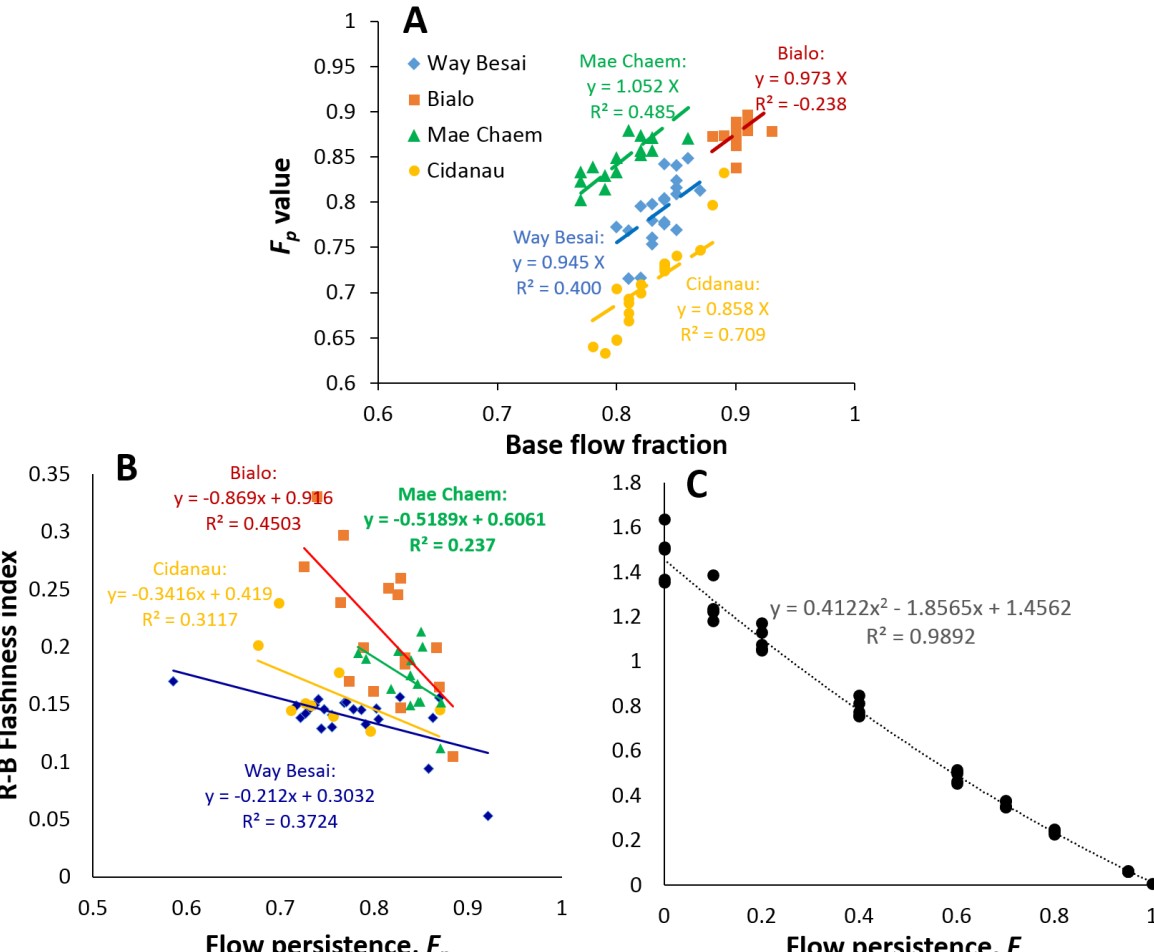


Figure 8. A) Comparison of yearly data for four Southeast Asian watersheds analysed with
common flow separation methods (average of results in Fig. 7) and the flow persistence
method and comparison of the Richards-Baker Flashiness Index (Baker et al., 2004) and
the flow persistence metric $F_p$ for B) four Southeast Asian watersheds, C) a series of
hydrographs as in Fig. 4A, with 5 replicates per $F_p$ value


Table 1. Comparison of properties of the Flashiness Index and Flow persistence $F_p$

| | Flashiness Index (Baker et al. 2004) | Flow persistence (as defined here) |
|---|---|---|
| 1. | Has direct appeal to non-technical audiences | Potentially similar |
| 2. | Where reservoir management rules imply major changes in ΔS, flashiness still describes implications for flow regimes | Is focused on the effects of changes in (upper) catchment land cover, not where reservoir management determines flow |
| 3. | Values depend on the scale of evaluating river flow; no absolute criteria for what is 'healthy' | Similar |
| 4. | Increase generally not desirable | Decrease generally not desirable |
| 5. | Varies in range [0-2], may need normalizing by division by 2 | Varies in range [0-1] |

| | | |
|---|---|---|
| 6. | Requires full year flow record to be calculated | Can be estimated from any set of sequential flow observations |
| 7. | Empirical metric, no direct link to underlying process understanding | Overall $F_p$ can be understood as weighted average of the $F_p$'s of contributing flow pathways (overland, subsurface and groundwater-based) |
| 8. | No directly visible relationship between peak and low flow characteristics | The $F_p$ term low flows and the $(1 - F_p)$ term for peak flows show the water balance logic of a link between peak and low flows |
| 9. | Aggregates changes in flow regime; no directly visible link between the performance metric, rainfall (or snow melt) and (vegetation dependent) evapotranspiration | The main water balance terms are directly reflected in the flow descriptions based on $F_p$ |
| 10. | Substantial empirical data bases available for comparison and meta studies | Not yet |


# Flood risk reduction and flow buffering as ecosystem services: II. Land use and rainfall intensity effects in Southeast Asia

Meine van Noordwijk[1,2], Lisa Tanika[1], Betha Lusiana[1]

[1]{World Agroforestry Centre (ICRAF), SE Asia program, Bogor, Indonesia}

[2]{Wageningen University, Plant Production Systems, Wageningen, the Netherlands}

Correspondence to: Meine van Noordwijk (m.vannoordwijk@cgiar.org)

**Abstract**

Watersheds buffer the temporal pattern of river flow relative to the temporal pattern of rainfall. This 'ecosystem service' is inherent to geology and climate, but buffering also responds to human use and misuse of the landscape. Buffering can be part of management feedback loops if salient, credible and legitimate indicators are used. The flow persistence parameter $F_p$ in a parsimonious recursive model of river flow (Part I) couples the transmission of extreme rainfall events (1- $F_p$), to the annual base flow fraction of a watershed ($F_p$). Here we compare $F_p$ estimates from four meso-scale watersheds in Indonesia (Cidanau, Way Besai, and Bialo) and Thailand (Mae Chaem), with varying climate, geology and land cover history, at a decadal time scale. The likely response in each of these four to variation in rainfall properties (incl. the maximum hourly rainfall intensity) and land cover (comparing scenarios with either more or less forest and tree cover than the current situation) was explored through a basic daily water balance model, GenRiver. This model was calibrated for each site on existing data, before being used for alternative land cover and rainfall parameter settings. In both data and model runs, the wet-season (3-monthly) $F_p$ values were consistently lower than dry-season values for all four sites. Across the four catchments $F_p$ values decreased with increasing annual rainfall, but specific aspects of watersheds, such as the riparian swamp (peat soils) in Cidanau reduced effects of land use change in the upper watershed. Increasing the mean rainfall intensity (at constant monthly totals for rainfall) around the values considered typical for each landscape was predicted to cause a decrease in $F_p$ values by between 0.047 (Bialo) and 0.261 (Mae Chaem). Sensitivity of $F_p$ to changes in land use change plus changes in rainfall intensity depends on other characteristics of the watersheds, and generalizations made on the basis of one or two case studies may not hold, even within the same climatic zone. A wet-season $F_p$ value above 0.7 was achievable in forest-Agroforestry mosaic case studies. Interannual variability in $F_p$ is large relative to effects of land cover change. Multiple (5-10) years of paired-plot data would generally be needed to reject no-change null-hypotheses on the effects of land use change (degradation and restoration). $F_p$ trends over time serve as a holistic scale-dependent performance indicator of degrading/recovering watershed health and can be tested for acceptability and acceptance in a wider social-ecological context.

**Introduction**

Inherent properties (geology, geomorphology) interact with climate and human modification of
vegetation, soils, drainage and riparian wetlands in effectuating the degree of buffering that
watersheds provide (Andréassian 2004; Bruijnzeel, 2004). Buffering of river flow relative to the
space-time dynamics of rainfall is an ecosystem service, reducing the exposure of people living on
geomorphological floodplains to high-flow events, and increasing predictability and river flow in dry
periods (Joshi et al., 2004; Leimona et al., 2015; Part I). In the absence of any vegetation and with a
sealed surface, river flow will directly respond to the spatial distribution of rainfall, with only the
travel time to any point of specific interest influencing the temporal pattern of river flow. Any
persistence or predictability of river flow in such a situation will reflect temporal autocorrelation of
rainfall, beyond statistical predictability in seasonal rainfall patterns. On the other side of the
spectrum, river flow can be constant every day, beyond the theoretical condition of constant rainfall,
in a watershed that provides perfect buffering, by passing all water through groundwater pools that
have sufficient storage capacity at any time during the year. Both infiltration-limited (Hortonian) and
saturation-induced use of more rapid flow pathways (inter and overland flows) will reduce the flow
persistence and make it, at least in part, dependent on rainfall events. Separating the effects of land
cover (land use), engineering and rainfall on the actual flow patterns of rivers remains a considerable
challenge (Ma et al., 2014; Verbist et al., 2019). It requires data, models and concepts that can serve
as effective boundary object in communication with stakeholders (Leimona et al. 2015; van
Noordwijk et al. 2012, 2016). There is a long tradition in using forest cover as such a boundary
object, but there is only a small amount of evidence supporting this (Tan-Soo et al., 2014; van Dijk et
al., 2009; van Noordwijk et al. 2015a; part I).
In part I, we introduced a flow persistence parameter ($F_p$) that links the two, asymmetrical aspects of
flow dynamics: translating rainfall excess into river flow, and gradually releasing water stored in the
landscape. The direct link between these two aspects can be seen from equation [4] in part I:
$Q_t = F_p Q_{t-1} + (1-F_p)(P_t − E_{tx})$
Where $Q_t$ and $Q_{t-1}$ represent river flow on subsequent days, $P_{tx}$ the precipitation on day t (or
preceding precipitation released as snowmelt on day t) and $E_{tx}$ the preceding evapotranspiration
since the previous precipitation event, creating storage space in the soils of the watershed. The first
term on the right-hand side of the equation represents the gradual release of stored water, causing
a slow decline of flow as the pools feeding this flow are gradually depleted. The second term reflects
the part of fresh additions of water are partitioned over immediate river flow and the increase of
stocks from which water can be gradually released.  The derivation of the link depended on the long
term water balance, and thus assumed that all out- and inflows are accounted for in the watershed.
Commonly used rainfall-runoff models (including the curve number approach and SWAT models)
only focus on the second term of the above equation (Ponce et al., 1996; Gassman et al., 2007),
without link to the first. Various empirical methods for deriving 'base flow' are in use, but details of
the calculation procedure matter. Results in part I for a number of contrasting meso-scale
watersheds in Southeast Asia suggested that interannual variation in $F_p$ within a given watershed
correlates with both the R-B Flashiness Index (Baker et al., 2004) and the base-flow fraction of
annual river flow. However, the slope of these relationships varied between watersheds. Here, in
part II we will further analyse the $F_p$ results for these watersheds that were selected to represent
variation in rainfall and land cover, and test the internal consistency of results based on historical
data: two located in the humid and one in the subhumid tropics of Indonesia, and one in the
unimodal subhumid tropics of northern Thailand.
After exploring the patterns of variation in $F_p$ estimates derived from actual river flow records, we
will quantify the sensitivity of the $F_p$ metric to variations in rainfall intensity and its response, on a
longer timescale to land cover change. To do so, we will use a model that uses basic water balance
concepts: rainfall interception, infiltration, water use by vegetation, overland flow, interflow and
groundwater release, to a spatially structured watershed where travel time from sub watersheds to
any point of interest modifies the predicted river flow. In the specific model used land cover effects
on soil conditions, interception and seasonal water use have been included. After testing whether $F_p$
values derived from model outputs match those based on empirical data where these exist, we rely
on the basic logic of the model to make inference on the relative importance of modifying rainfall
and land cover inputs. With the resulting temporal variation in calculated $F_p$ values, we consider the
time frame at which observed shifts in $F_p$ can be attributed to factors other than chance (that means:
null-hypotheses of random effects can be rejected with accepted chance of Type I errors).
**2. Methods**
**2.1 GenRiver model for effects of land cover on river flow**
The GenRiver model (van Noordwijk et al., 2011) is based on a simple water balance concept with a
daily time step and a flexible spatial subdivision of a watershed that influences the routing of water
and employs spatially explicit rainfall. At patch level, vegetation influences interception, retention
for subsequent evaporation and delayed transfer to the soil surface, as well as the seasonal demand
for water. Vegetation (land cover) also influences soil porosity and infiltration, modifying the
inherent soil properties. Water in the root zone is modelled separately for each land cover within a
subcatchment, the groundwater stock is modelled at subcatchment level. The spatial structure of a
watershed and the routing of surface flows influences the time delays to any specified point of
interest, which normally includes the outflow of the catchment. Land cover change scenarios are
interpolated annually between time-series (measured or modelled) data. The model may use
measured rainfall data, or use a rainfall generator that involves Markov chain temporal
autocorrelation (rain persistence). As our data sources are mostly restricted to daily rainfall
measurements and the infiltration model compares instantaneous rainfall to infiltration capacity, a
stochastic rainfall intensity was applied at subcatchment level, driven by the mean as parameter and
a standard deviation for a normal distribution (truncated at 3 standard deviations from the mean)
proportional to it via a coefficient of variation as parameter. For the Mae Chaem site in N Thailand
data by Dairaku et al. (2004) suggested a mean of less than 3 mm/hr. For the three sites in Indonesia
we used 30 mm/hr, based on Kusumastuti et al. (2016). Appendix 1 provides further detail on the
GenRiver model. The model itself, a manual and application case studies are freely available
(http://www.worldAgroforestry.org/output/genriver-genetic-river-model-river-flow;van Noordwijk
et al., 2011).
**2.2 Empirical data-sets, model calibration**
Table 1 and Figure 1 provide summary characteristics and the location of river flow data  used in four
meso-scale watersheds for testing the $F_p$ algorithm and application of the GenRiver model. Figure 1
includes a water tower category in the agro-ecological zones; this is defined on the basis of a ratio of
precipitation and potential evapotranspiration of more than 0.65, and a product of that ratio and
relative elevation exceeding 0.277.
⇨ Table 1
⇨ Figure 1
As major parameters for the GenRiver model were not independently measured for the respective
watersheds, we tuned (calibrated) the model by modifying parameters within a predetermined
plausible range, and used correspondence with measured hydrograph as test criterion (Kobolt et al.
2008). We used the Nash-Sutcliff Efficiency (NSE) parameter (target above 0.5) and bias (less than
25%) as test criteria and targets. Meeting these performance targets (Moriasi et al., 2007), we
accepted the adjusted models as basis for describing current conditions and exploring model
sensitivity. The main site-specific parameter values are listed in Table 2 and (generic) land cover
specific default parameters in Table 3.
⇨ Table 2
⇨ Table 3
Table 4 describes the six scenarios of land use change that were evaluated in terms of their
hydrological impacts. Further description on the associated land cover distribution for each scenario
in the four different watersheds is depicted in Appendix 2.
⇨ Table 4
**2.3 Bootstrapping to estimate the minimum observation**
The bootstrap methods (Efron and Tibshirani, 1986) is a resampling methods that is commonly used
to generate 'surrogate population' for the purpose of approximating the sampling distribution of a
statistic. In this study, the bootstrap approach was used to estimate the minimum number of
observation (or yearly data) required for a pair-wise comparison test between two time-series of
stream flow or discharge data (representing two scenarios of land use distributions) to be
distinguishable from a null-hypothesis of no effect. The pair-wise comparison test used was
Kolmogorov-Smirnov test that is commonly used to test the distribution of discharge data (Zhang eta
al, 2006). We built a simple macro in R (R Core Team, 2015) that entails the following steps:
(i)   Bootstrap or resample with replacement 1000 times from both time-series discharge data
with sample size $n$;
(ii)  Apply the Kolmogorov-Smirnov test to each of the 1000 generated pair-wise discharge data,
and record the P-value;
(iii) Perform (i) and (ii) for different size of $n$, ranging from 5 to 50.
(iv)  Tabulate the p-value from the different sample size $n$, and determine the value of $n$ when the
p-value reached equal to or less than 0.025 (or equal to the significance level of 5%). The
associated $n$ represents the minimum number of observations required.
Appendix 3 provides an example of the macro in R used for this analysis.

## 3. Results
**3.1 Empirical data of flow persistence as basis for model parameterization**
Inter-annual variability of $F_p$ estimates derived for the four catchments (Figure 2) was of the order of
0.1 units, while the intra-annual variability between dry and rainy seasons was 0.1-0.2. For all years
and locations, rainy season $F_p$ values, with mixed flow pathways, were consistently below dry-season
values, dominated by groundwater flows. If we can expect $F_{p,i}$ and $F_{p,o}$ (see equation 8 in part I) to be
approximately 0.5 and 0, this difference between wet and dry periods implies a 40% contribution of
interflow in the wet season, a 20% contribution of overland flow or any combination of the two
effects.
Overall the estimates from modelled and observed data are related with 16% deviating more than
0.1 and 3% more than 0.15 (Figure 3). As the Moriasi et al. (2007) performance criteria for the
hydrographs were met by the calibrated models for each site, we tentatively accept the model to be
a basis for sensitivity study of Fp to modifications to land cover and/or rainfall
⇨ Figure 2
⇨ Figure 3

## 3.2 Comparing $F_p$ effects of rainfall intensity and land cover change

A direct comparison of model sensitivity to changes in mean rainfall intensity and land use change
scenarios is provided in Figure 4. Varying the mean rainfall intensity over a factor 7 shifted the $F_p$
value by only 0.047 and 0.059 in the case of Bialo and Cidanau, respectively, but by 0.128 in Way
Besai and 0.261 in Mae Chaem (Figure 4A). The impact of the land use change scenarios on $F_p$ was
smallest in Cidanau (0.026), intermediate in Way Besai (0.048) and relatively large in Bialo and Mae
Chaem, at 0.080 and 0.084, respectively (Figure 4B). The order of $F_p$ across the land use change
scenarios was mostly consistent between the watersheds, but the contrast between the
Reforestation and Natural Forest scenario was largest in Mae Chaem and smallest in Way Besai. In
Cidanau, Way Besai and Mae Chaem, variations in rainfall were 2.2 to 3.1 times more effective than
land use change in shifting $F_p$, in Bialo its relative effect was only 58%. Apparently, the sensitivity to
changes in land use change plus changes in rainfall intensity depends on other characteristics of the
watersheds, and generalizations made on the basis of one or two case studies may not hold, even
within the same climatic zone.
⇨ Figure 4

## 3.3 Further analysis of $F_p$ effects for scenarios of land cover change

Among the four watersheds there is consistency in that the 'forest' scenario has the highest, and the
'degraded lands' the lowest $F_p$ value (Figure 5), but there are remarkable differences as well: in
Cidanau the interannual variation in $F_p$ is clearly larger than land cover effects, while in the Way
Besai the spread in land use scenarios is larger than interannual variability. In Cidanau a peat swamp
between most of the catchment and the measuring point buffers most of land cover related
variation in flow, but not the interannual variability. Considering the frequency distributions of $F_p$
values over a 20 year period, we see one watershed (Way Besai) where the forest stands out from all
others, and one (Bialo) where the degraded lands are separate from the others. Given the degree of
overlap of the frequency distributions, it is clear that multiple years of empirical observations will be
needed before a change can be affirmed.
Figure 5 shows the frequency distributions of expected effect sizes on $F_p$ of a comparison of any land
cover with either forest or degraded lands. Table 5 translates this information to the number of
years that a paired plot (in the absence of measurement error) would have to be maintained to
reject a null-hypothesis of no effect, at p=0.05. As the frequency distributions of $F_p$ differences of
paired catchments do not match a normal distribution, a Kolmorov-Smirnov test can be used to
assess the probability that a no-difference null hypothesis can yield the difference found. By
bootstrapping within the years where simulations supported by observed rainfall data exist, we
found for the Way Besai catchment, for example, that 20 years of data would be needed to assert (at
P = 0.05) that the Reforestation scenario differs from Agroforestation, and 16 years that it differs
from Actual and 11 years that it differs from Degrade. In practice, that means that empirical
evidence that survives statistical tests will not emerge, even though effects on watershed health are
real.
⇨ Figure 5
⇨ Table 5
At process-level the increase in 'overland flow' in response to soil compaction due to land cover
change has a clear and statistically significant relationship with decreasing $F_p$ values in all catchments
(Figure 6), but both year-to-year variation within a catchment and differences between catchments
influence the results as well, leading to considerable spread in the biplot. Contrary to expectations,
the disappearance of 'interflow' by soil compaction is not reflected in measurable change in $F_p$ value.
The temporal difference between overland and interflow (one or a few days) gets easily blurred in
the river response that integrates over multiple streams with variation in delivery times; the
difference between overland- or interflow and baseflow is much more pronounced. Apparently,
according to our model, the high macroporosity of forest soils that allows interflow and may be the
'sponge' effect attributed to forest, delays delivery to rivers by one or a few days, with little effect on
the flow volumes at locations downstream where flow of multiple days accumulates.  The difference
between overland- or interflow and baseflow in time-to-river of rainfall peaks is much more
pronounced.
⇨ Figure 6
Tree cover has two contradicting effects on baseflow:  it reduces the surplus of rainfall over
evapotranspiration (annual water yield) by increased evapotranspiration (especially where
evergreen trees or trees with a large canopy interception are involved), but it potentially increases
soil macroporosity that supports infiltration and interflow, with relatively little effect on water
holding capacity measured as 'field capacity' (after runoff and interflow have removed excess
water). Figure 7 shows that the total volume of baseflow differs more between sites and their
rainfall pattern than it varies with tree cover. Between years total evapotranspiration and baseflow
totals are positively correlated, but for a given rainfall there is a trade-off. Overall these results
support the conclusion that generic effects of deforestation on decreased flow persistence, and of
(agro)/(re)-forestation on increased flow persistence are small relative to interannual variability due
to specific rainfall patterns, and that it will be hard for any empirical data process to pick-up such
effects, even if they are qualitatively aligned with valid process-based models.
⇨ Figure 7
**4. Discussion**
In the discussion of Part I the credibility questions on replicability of the $F_p$ metric and its sensitivity
to details of rainfall pattern versus land cover as potential causes of variation were seen as requiring
case studies in a range of contexts. Although the four case studies in Southeast Asia presented here
cannot be claimed to represent the global variation in catchment behaviour (with absence of a
snowpack and its dynamics as an obvious element of flow buffering not included), the diversity of
responses among these four already point to challenges for any generic interpretation of the degree
of flow persistence that can be achieved under natural forest cover, as well as its response to land
cover change.
The empirical data summarized here for (sub)humid tropical sites in Indonesia and Thailand show
that values of $F_p$ above 0.9 are scarce in the case studies provided, but values above 0.8 were found,
or inferred by the model, for forested landscapes. Agroforestry landscapes generally presented $F_p$
values above 0.7, while open-field agriculture or degraded soils led to $F_p$ values of 0.5 or lower. Due
to differences in local context, it may not be feasible to relate typical $F_p$ values to the overall
condition of a watershed, but temporal change in $F_p$ can indicate degradation or restoration if a
location-specific reference can be found. The difference between wet and dry season $F_p$ can be
further explored in this context. The dry season $F_p$ value primarily reflects the underlying geology,
with potential modification by engineering and operating rules of reservoirs, the wet season $F_p$ is
generally lower due to partial shifts to overland and interflow pathways.  Where further uncertainty
is introduced by the use of modelled rather than measured river flow, the lack of fit of models
similar to the ones we used here would mean that scenario results are indicative of directions of
change rather than a precision tool for fine-tuning combinations of engineering and land cover
change as part of integrated watershed management.
The differences in relative response of the watersheds to changes in mean rainfall intensity and land
cover change, suggest that generalizations derived from one or a few case studies are to be
interpreted cautiously. If land cover change would influence details of the rainfall generation process
(arrow 10 in Figure 1 of part I; e.g. through release of ice-nucleating bacteria Morris et al., 2014; van
Noordwijk et al., 2015b) this can easily dominate over effects via interception, transpiration and soil
changes.
Our results indicate an intra-annual variability of $F_p$ values between wet and dry seasons of around
0.2 in the case studies, while interannual variability in either annual or seasonal $F_p$ was generally in
the 0.1 range. The difference between observed and simulated flow data as basis for $F_p$ calculations
was mostly less than 0.1. With current methods, it seems that effects of land cover change on flow
persistence that shift the $F_p$ value by about 0.1 are the limit of what can be  asserted from empirical
data (with shifts of that order in a single year a warning sign rather than a firmly established change).
When derived from observed river flow data $F_p$ is suitable for monitoring change (degradation,
restoration) and can be a serious candidate for monitoring performance in outcome-based
ecosystem service management contracts. Choice of the part of the year for which $F_p$ changes are
used as indicator may have to depend on the seasonal patterns of rainfall.
In view of our results the lack of robust evidence in the literature of effects of change in forest and
tree cover on flood occurrence may not be a surprise; effects are subtle and most data sets contain
considerable variability. Yet, such effects are consistent with current process and scaling knowledge
of watersheds.
In summarizing findings on the $F_p$ metric, we can compare it with existing ones across the seven
questions raised in Fig. 1 of part I. Comparator metrics can derive from various data sources,
including the amount (and/or quality) of forest cover upstream, the fraction of flows that is
technically controlled, direct records of river flow (over a short or longer time period), records of
rainfall and/or models that combine landscape properties, climate and land cover. Tentative scoring
for these metrics (Table 6) suggest that the $F_p$ metric is an efficient tool for data-scarce
environments, as it indicates aspects of hydrographs that so far required multi-annual records of
river flow.
➜Table 6

## Conclusion

Overall, our analysis suggests that the level of flow buffering achieved depends on both land cover
(including its spatial configuration and effects on soil properties) and space-time patterns of rainfall
(including maximum rainfall intensity as determinant of overland flow). Generalizations on dominant
influence of either, derived from one or a few case studies are to be interpreted cautiously. If land
cover change would influence details of the rainfall generation process this can easily dominate over
effects via interception, transpiration and soil changes. Multi-year data will generally be needed to
attribute observed changes in flow buffering to degradation/restoration of watersheds, rather than
specific rainfall events. With current methods, it seems that effects of land cover change on flow
persistence that shift the $F_p$ value by about 0.1 are the limit of what can be  asserted from empirical
data, with shifts of that order in a single year a warning sign rather than a firmly established change.
When derived from observed river flow data $F_p$ is suitable for monitoring change (degradation,
restoration) and can be a serious candidate for monitoring performance in outcome-based
ecosystem service management contracts. Watershed health is here characterized through the flow
pattern it generates, leaving the attribution to land cover, rainfall pattern and engineering of that
pattern and of changes in pattern to further location-specific analysis, just as a symptom of a high
body temperature can indicate health, but not diagnose the specific illness causing it.
The data sets analysed so far did not indicate that the flow persistence at high flows differed from
that at lower flows within the same season, but in other circumstances this may not be the case and
further care may be needed to use $F_p$ values beyond the measurement period in which they were
derived. While a major strength of the $F_p$ method over existing procedures for parameterizing curve
number estimates, for example, is that the latter depend on scarce observations during extreme
events and $F_p$ can be estimated for any part of the flow record, the reliability of $F_p$ estimates will still
increase with the length of the observation period.
Further tests on the performance of the $F_p$ metric and its standard incorporation into the output
modules of river flow and watershed management models will broaden the basis for interpreting the
value ranges that can be expected for well-functioning watersheds in various conditions of climate,
topography, soils, vegetation and engineering interventions. Such a broader empirical base could
test the possible use of $F_p$ as performance metric for watershed rehabilitation efforts.

## Data availability

Table 7 specifies the rainfall and river flow data we used for the four basins and specifies the links to
detailed descriptions.
⇨  Table 7

## Acknowledgements

This research is part of the Forests, Trees and Agroforestry research program of the CGIAR. Several colleagues contributed to the development and early tests of the $F_p$ method. Thanks are due to Thoha Zulkarnain for assistance with Figure 1 and to Eike Luedeling, Sonya Dewi, Sampurno Bruijnzeel and two anonymous reviewers for comments on an earlier version of the manuscript.

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

Table 1. Basic physiographic characteristics of the four study watersheds

| Parameter | Bialo | Cidanau | Mae Chaem | Way Besai |
|---|---|---|---|---|
| Location | South Sulawesi, Indonesia | West Java, Indonesia | Northern Thailand | Lampung, Sumatera, Indonesia |
| Coordinates | 5.43 S, 120.01 E | 6.21 S, 105.97 E | 18.57 N, 98.35 E | 5.01 S, 104.43 E |
| Area (km$^2$) | 111.7 | 241.6 | 3892 | 414.4 |
| Elevation (m a.s.l.) | 0 – 2874 | 30 – 1778 | 475-2560 | 720-1831 |
| Flow pattern | Parallel | Parallel (with two main river flow that meet in the downstream area) | Parallel | Radial |
| Land cover type | Forest (13%) Agroforest (59%) Crops (22%) Others (6%) | Forest (20%) Agroforest (32%) Crops (33%) Others (11%) Swamp(4%) | Forest (evergreen, deciduous and pine) (84%) Crops (15%) Others (1%) | Forest (18%) Coffee (monoculture and multistrata) (64%) Crop and Horticulture (12%) Others (6%) |
| Mean annual rainfall, mm | 1695 | 2573 | 1027 | 2474 |
| Wet season | April – June | January - March | July - September | January - March |
| Dry season | July - September | July - September | January - March | July - September |
| Mean annual runoff, mm | 947 | 917 | 259 | 1673 |
| Major soils | Inceptisols | Inceptisols | Ultisols, Entisols | Andisols |


Table 2. Parameters of the GenRiver model used for the four site specific simulations (van Noordwijk
et al., 2011 for definitions of terms; sequence of parameters follows the pathway of water)

| Parameter | Definition | Unit | Bialo | Cidanau | Mae Chaem | Way Besai |
|---|---|---|---|---|---|---|
| RainIntensMean | Average rainfall intensity | mm hr$^{-1}$ | 30 | 30 | 3 | 30 |
| RainIntensCoefVar | Coefficient of variation of rainfall intensity | mm hr$^{-1}$ | 0.8 | 0.3 | 0.5 | 0.3 |

| | | | | | | |
|---|---|---|---|---|---|---|
| RainInterceptDripRt | Maximum drip rate of intercepted rain | mm hr$^{-1}$ | 80 | 10 | 10 | 10 |
| RainMaxIntDripDur | Maximum dripping duration of intercepted rain | hr | 0.8 | 0.5 | 0.5 | 0.5 |
| InterceptEffectontrans | Rain interception effect on transpiration | - | 0.35 | 0.8 | 0.3 | 0.8 |
| MaxInfRate | Maximum infiltration capacity | mm d$^{-1}$ | 580 | 800 | 150 | 720 |
| MaxInfSubsoil | Maximum infiltration capacity of the sub soil | mm d$^{-1}$ | 80 | 120 | 150 | 120 |
| PerFracMultiplier | Daily soil water drainage as fraction of groundwater release fraction | - | 0.35 | 0.13 | 0.1 | 0.1 |
| MaxDynGrWatStore | Dynamic groundwater storage capacity | mm | 100 | 100 | 300 | 300 |
| GWReleaseFracVar | Groundwater release fraction, applied to all subcatchments | - | 0.15 | 0.03 | 0.05 | 0.1 |
| Tortuosity | Stream shape factor | - | 0.4 | 0.4 | 0.6 | 0.45 |
| Dispersal Factor | Drainage density | - | 0.3 | 0.4 | 0.3 | 0.45 |
| River Velocity | River flow velocity | m s$^{-1}$ | 0.4 | 0.7 | 0.35 | 0.5 |


Table 3. GenRiver defaults for land use specific parameter values, used for all four watersheds
(BD/BDref indicates the bulk density relative to that for an agricultural soil pedotransfer function;
see van Noordwijk et al., 2011)

| Land cover Type | Potential interception (mm/d) | Relative drought threshold | BD/BDref |
|---|---|---|---|
| Forest[1] | 3.0 - 4.0 | 0.4 - 0.5 | 0.8 - 1.1 |
| Agroforestry[2] | 2.0 - 3.0 | 0.5 - 0.6 | 0.95 - 1.05 |
| Monoculture tree[3] | 1.0 | 0.55 | 1.08 |
| Annual crops | 1.0 - 3.0 | 0.6 - 0.7 | 1.1 - 1.5 |
| Horticulture | 1.0 | 0.7 | 1.07 |
| Rice field[4] | 1.0 - 3.0 | 0.9 | 1.1 - 1.2 |
| Settlement | 0.05 | 0.01 | 1.3 |
| Shrub and grass | 2.0 - 3.0 | 0.6 | 1.0 - 1.07 |
| Cleared land | 1.0 - 1.5 | 0.3 - 0.4 | 1.1 - 1.2 |

Note:    1. Forest: primary forest, secondary forest, swamp forest, evergreen forest, deciduous forest

2. Agroforestry: mixed garden, coffee, cocoa, clove

3. Monoculture : coffee

4. Rice field: irrigation and rainfed


Table 4. Land use scenarios explored for four watersheds

| Scenario | Description |
| --- | --- |
| Natural Forest | Full natural forest, hypothetical reference scenario |
| Reforestation | Reforestation, replanting shrub, cleared land, grass land and some agricultural area with forest |
| Agroforestation | Agroforestry scenario, maintaining Agroforestry areas and converting shrub, cleared land, grass land and some of agricultural area into Agroforestry |
| Actual | Baseline scenario, based on the actual condition of land cover change during the modelled time period |
| Agriculture | Agriculture scenario, converting some of tree based plantations, cleared land, shrub and grass land into rice fields or dry land agriculture, while maintain existing forest |
| Degrading | No change in already degraded areas, while converting most of forest and Agroforestry area into rice fields and dry land agriculture |



Table 5. Number of years of observations required to estimate flow persistence to reject the null-hypothesis of 'no land use effect', at p-value = 0.05 using Kolmogorov-Smirnov test. The probability of the test statistic in the first significant number is provided between brackets and where the number of observations exceeds the time series available, results are given in *italics*

A. Natural Forest as reference

| Way Besai (N=32) | Refores-tation | Agrofo-restation | Actual | Agricultural |
|---|---|---|---|---|
| Reforestation | | 20 (0.035) | 16 (0.037) | 13 (0.046) |
| Agroforestation | | | n.s. | n.s. |
| Actual | | | | n.s. |
| Agricultural | | | | |
| Degrading | | | | |

| Bialo (N=18) | | | | |
|---|---|---|---|---|
| Reforestation | | n.s. | n.s. | *37 (0.04)* |
| Agroforestation | | | n.s. | n.s. |
| Actual | | | | n.s. |
| Agricultural | | | | |
| Degrading | | | | |

| Cidanau (N=20) | | | | |
|---|---|---|---|---|
| Reforestation | | n.s. | n.s. | *32 (0.037)* |
| Agroforestation | | | n.s. | n.s. |
| Actual | | | | n.s. |
| Agricultural | | | | |
| Degrading | | | | |

| Mae Chaem (N=15) | | | | |
|---|---|---|---|---|
| Reforestation | | n.s. | *23 (0.049)* | *18 (0.050)* |
| Agroforestation | | | *45 (0.037)* | *33 (0.041)* |

| Actual | | | 33 (0.041) | |
| Agricultural | | | | |

### B. Degrading scenario as reference

| Way Besai (N=32) | Natural forest | Reforestation | Agrofo-restation | Actual | Agriculture |
|---|---|---|---|---|---|
| Natural forest | | n.s. | 17 (0.042) | 13 (0.046) | 7 (0.023) |
| Reforestation | | | 21 (0.037) | 19 (0.026) | 7 (0.023) |
| Agroforestation | | | | n.s. | 28 (0.046) |
| Actual | | | | | 30 (0.029) |
| Agriculture | | | | | |

| Bialo (N=18) | | | | | |
|---|---|---|---|---|---|
| Natural forest | | n.s. | n.s. | *41 (0.047)* | *19 (0.026)* |
| Reforestation | | | n.s. | n.s. | *32 (0.037)* |
| Agroforestation | | | | n.s. | n.s. |
| Actual | | | | | n.s. |
| Agricultural | | | | | |

| Cidanau (N=20) | | | | | |
|---|---|---|---|---|---|
| Natural forest | | n.s. | n.s. | *33 (0.041)* | 8 (0.034) |
| Reforestation | | | n.s. | n.s. | 15 (0.028) |
| Agroforestation | | | | n.s. | n.s. |
| Actual | | | | | *25 (0.031)* |
| Agricultural | | | | | |

| Mae Chaem (N=15) | Natural forest | Reforestation | Actual | Agriculture |
|---|---|---|---|---|
| Natural forest | | n.s. | 25 (0.031) | 12 (0.037) |
| Reforestation | | | n.s. | *18 (0.050)* |

| | |
|---|---|
| Agroforestation | *18 (0.050)* |
| Actual | |


Table 6. Comparison of metrics at various points in the causal network (Fig. 2 of Paper I) that can
support watershed management and prevention of flood damage on the list of seven issues (I – VII)
introduced in Fig. 1 Paper I[*].

| | Terrain-based (7A and 5 in Fig. 2 of part I) | | Based on river flow characteristics (4 in Fig. 2 of part I) | | | | | | Integrated (5-7) terrain + climate + land use + river flow models | |
|---|---|---|---|---|---|---|---|---|---|---|
| Is-sues[*] | Forest cover | Fraction of flow technically regulated | $Q_{max}$ / $Q_{min}$ | Flashi-ness index | Flow fre-quency analysis | Curve-number (rainfall-runoff) | Base-flow | Flow persis-tence, $F_p$ | Spatial analysis | Spatial water flow model |
| Range | 0-100% | 0–100% | $1 - \omega$ | 0 - 2 | | 1 - 100 | 0-100% | 0 - 1 | | |
| IA | No | Yes | No | Yes | Yes | Yes | No | Yes | Partially | Yes |
| IB | No | Yes | No | No | Yes | No | Yes | Yes | Partially | Yes |
| IIA | Not | Partially | Not | Not | Yes | Partially | Partially | Partially | Partially | Partially |
| IIB | Partially | Yes | Not | Not | Not | Partially | Partially | Partially | Partially | Yes |
| IIC | Not | Partially | Not | Partially | Partially | Not | Partially | Partially | Partially | Yes |
| III | Partially | Partially | Not | Partially | Yes | Partially | Partially | Partially | Partially | Yes |
| IVA | Single | - | Single | Single | Multi | Multi | Single | Single | Single | Single |
| IVB | Robust | Robust | Sensitive | Sensitive | Sensitive | Sensitive | Robust | Robust | Robust | Robust |
| V | Partially | Not | Not | Yes | No | No | Partially | Yes | Partially | Partially |
| VI | Not | Not | Not | Partially | Not | Not | Not | Yes | Partially | Partially |
| VII | Not | Neutral | Not | Yes | Yes | Neutral | Neutral | Yes | Yes | Yes |

I.   Does the indicator relate to important aspects of watershed behaviour (A. Flood damage
prevention; B. Low flow water availability)?

II.   Does its quantification help to select management actions? (A. Risk assessment, insurance
design; B. Spatial planning, engineering interventions; C. Fine-tuning land use)

III.   Is it consistent with current understanding of key processes
IV.   Are data requirements feasible (A. Lowest temporal resolution for estimates (years); B.
Consistency of numerical results and sensitivity to bias and random error in data sources?)

V.   Does it match local knowledge and concerns?
VI.   Can it be used to empower local stakeholders of watershed management through
performance (outcome) based contracts?

VII.   Can it inform local risk management?

Table 7. Data availability

|  | Bialo | Cidanau | Mae Chaem | Way Besai |
|---|---|---|---|---|
| Rainfall data | 1989-2009, Source: BWS Sulawesi[a] and PUSAIR[b]; Average rainfall data from the stations Moti, Bulo-bulo, Seka and Onto | 1998-2008, source: BMKG[c] | 1998-2002, source: WRD55, MTD22, RYP48, GMT13, WRD 52 | 1976-2007, Source: BMKG, PU[d] and PLN[e] (interpolation of 8 rainfall stations using Thiessen polygon) |
| River flow data | 1993-2010, source; BWS Sulawesi and PUSAIR | 2000-2009, source: KTI[f] | 1954-2003, source: ICHARM[g] | 1976-1998, source: PU and PUSAIR |
| Reference of detailed report | http://old.icraf.org/regions/southeast_asia/publications?do=view_pub_detail&pub_no=PP0343-14 | http://worldAgroforestry.org/regions/southeast_asia/publications?do=view_pub_detail&pub_no=PO0292-13 | http://worldAgroforestry.org/regions/southeast_asia/publications?do=view_pub_detail&pub_no=MN0048-11 | http://worldAgroforestry.org/regions/southeast_asia/publications?do=view_pub_detail&pub_no=MN0048-11 |

Note:
[a] BWS: Balai Wilayah Sungai (*Regional River Agency*)
[b] PUSAIR: Pusat Litbang Sumber Daya Air (*Centre for Research and Development on Water Resources)*
[c] BMKG: Badan Meteorologi Klimatologi dan Geofisika (*Agency on Meteorology, Climatology and*
*Geophysics*)

[d] PU: Dinas Pekerjaan Unum *(Public Work  Agency)*
[e] PLN: Perusahaan Listrik Negara (*National Electric Company*)
[f] KTI: Krakatau Tirta Industri, a private steel company
[f] ICHARM: The International Centre for Water Hazard and Risk Management

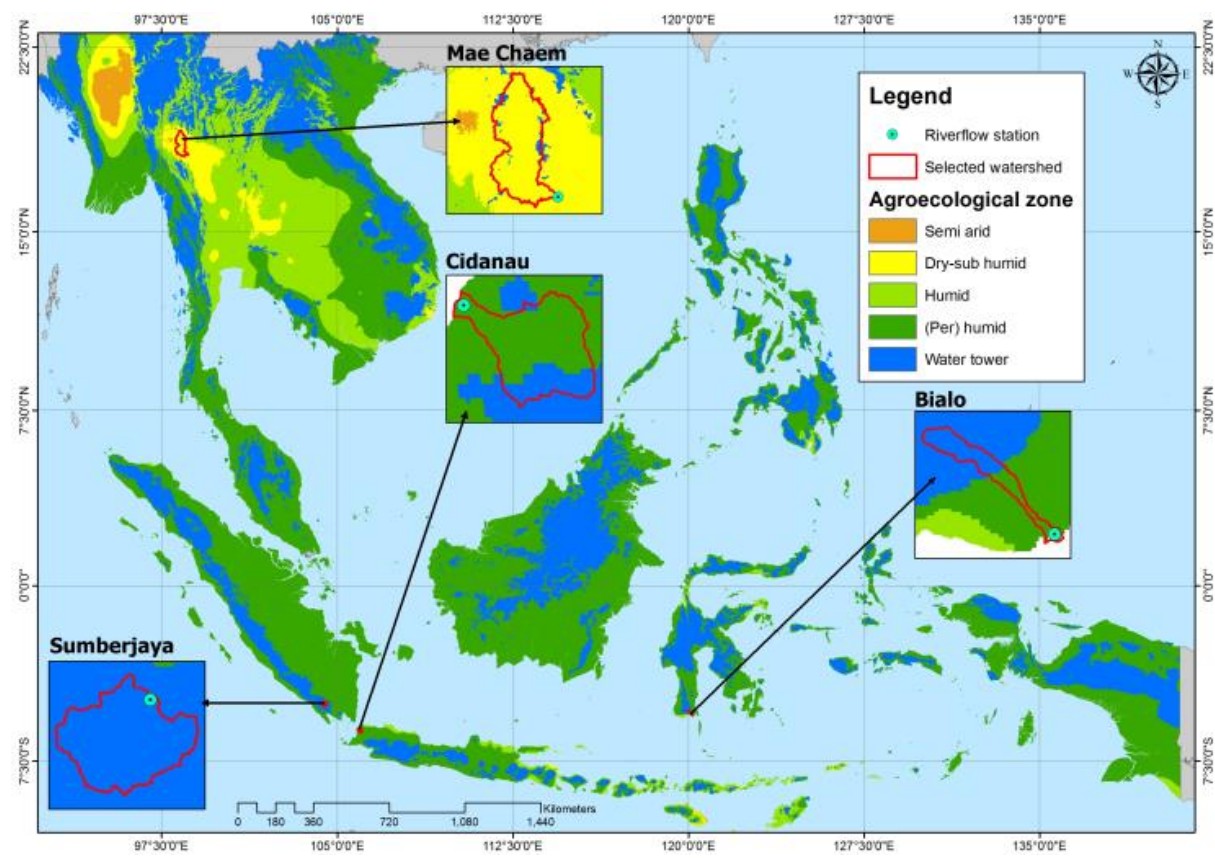


Figure 1. Location of the four watersheds in the agroecological zones of Southeast Asia (water
towers are defined on the basis of ability to generate river flow and being in the upper part of a
watershed)


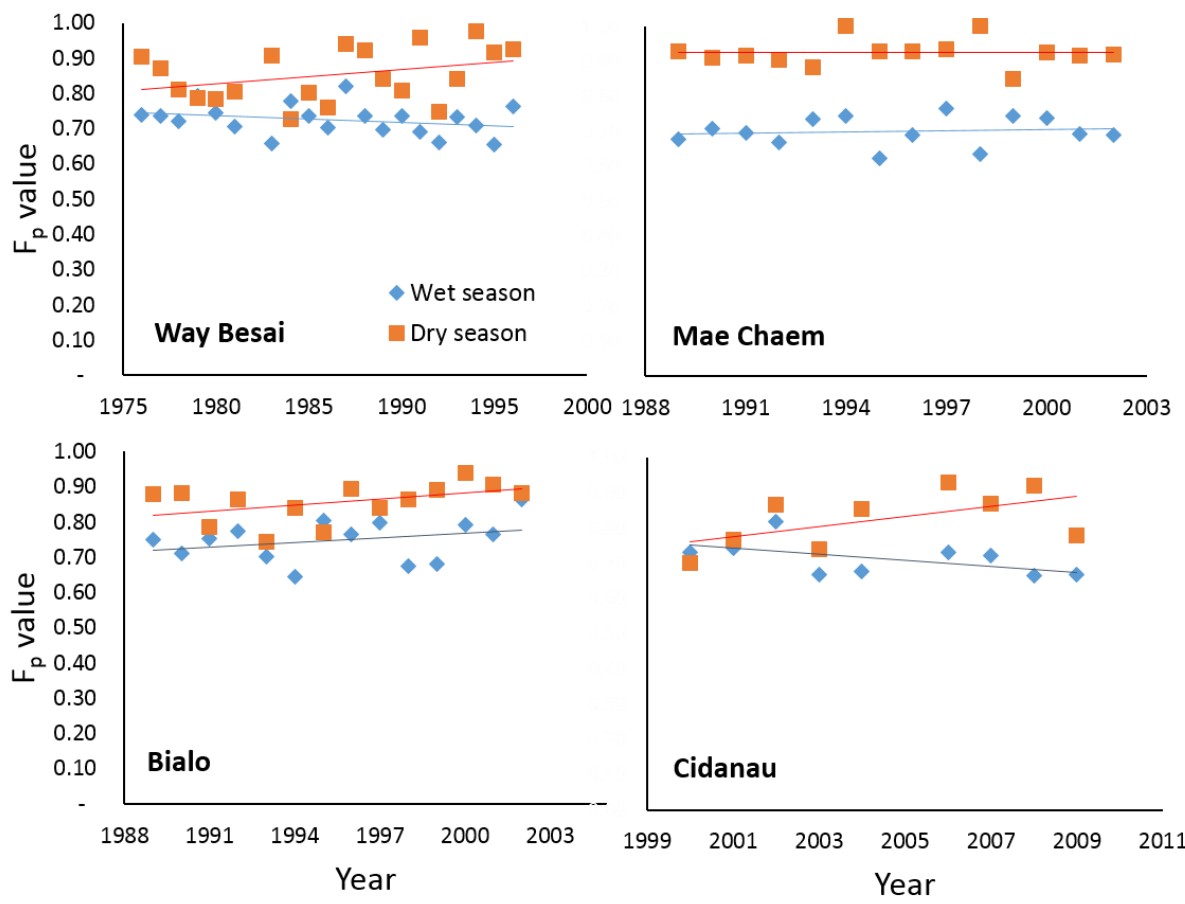


Figure 2. Flow persistence (F$_p$) estimates derived from measurements in four Southeast Asian
watersheds, separately for the wettest and driest 3-month periods of the year



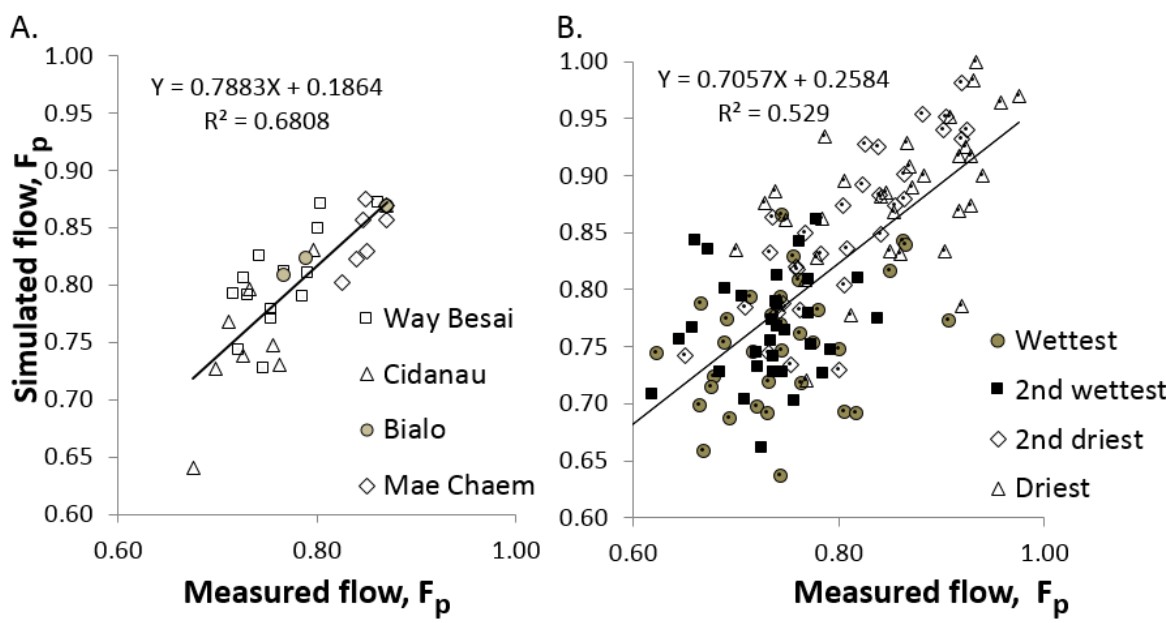


Figure 3. Inter- (A) and intra- (B) annual variation in the $F_p$ parameter derived from empirical versus
modelled flow: for the four test sites on annual basis (A) or three-monthly basis (B)



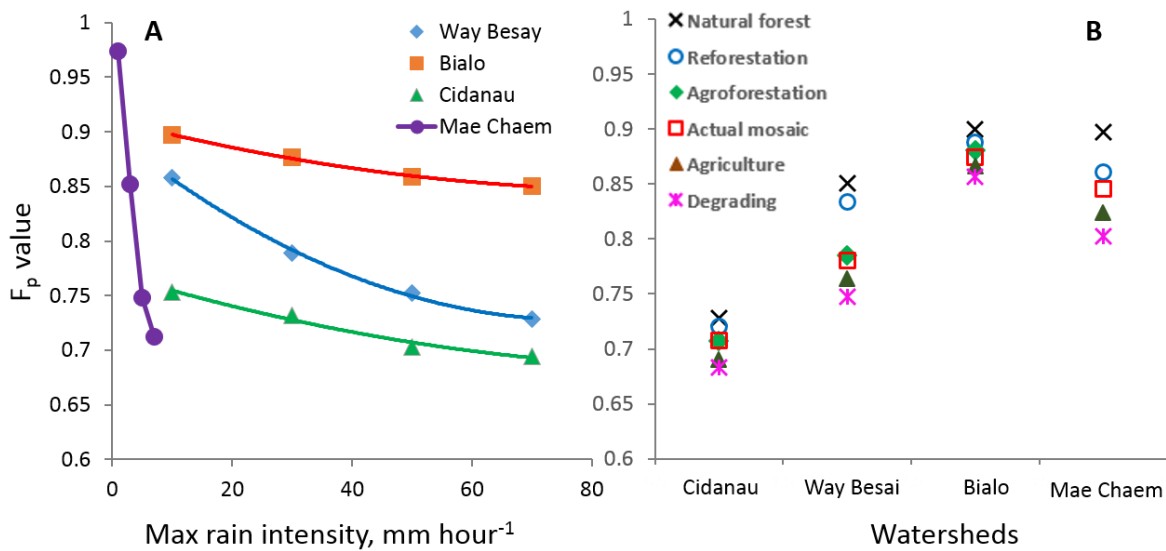


Figure 4 Effects on flow persistence of changes in A) the mean rainfall intensity and B) the land use

change scenarios of Table 4 across the four watersheds


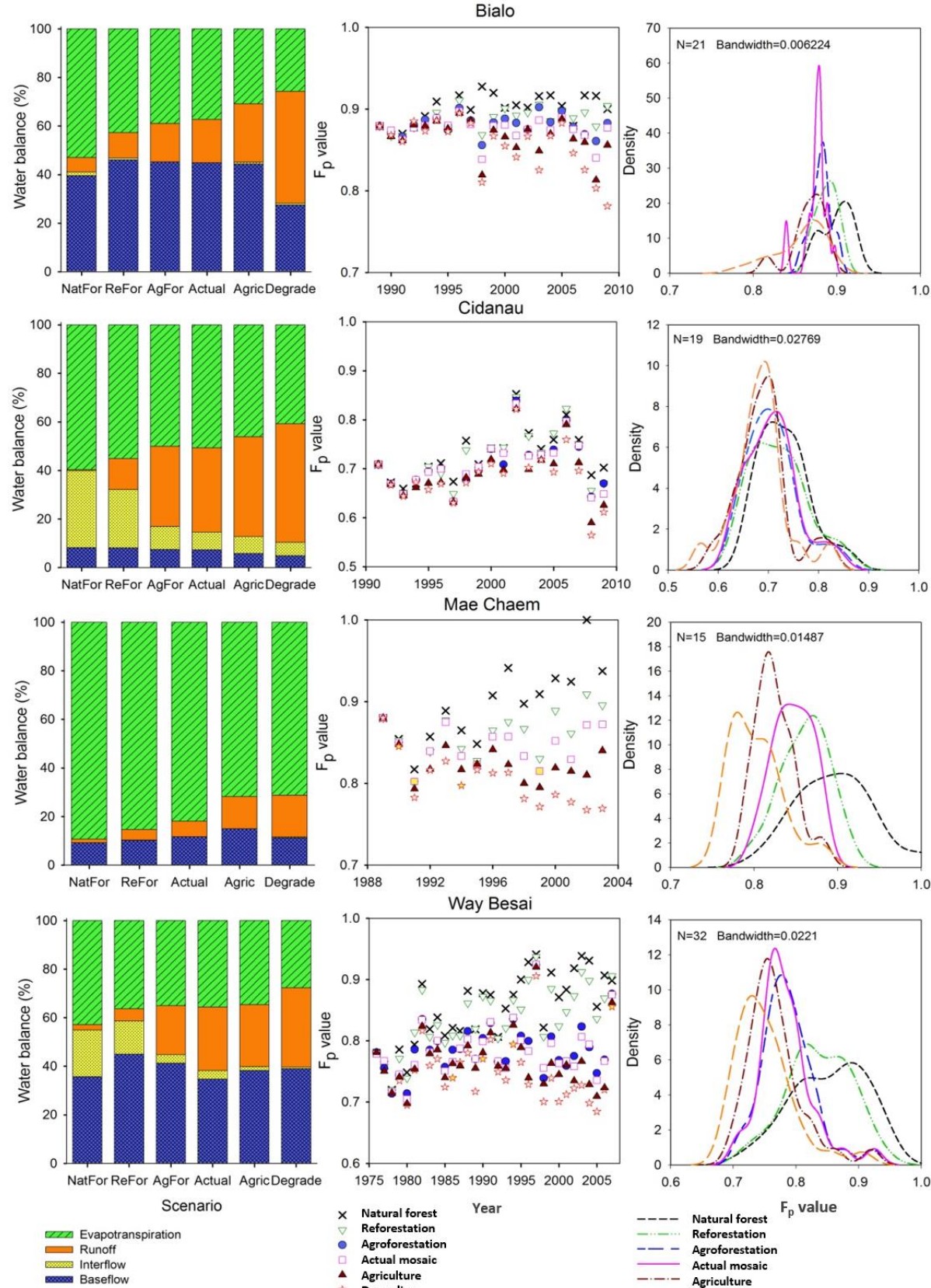


Figure 5. Effects of land cover change scenarios (Table 4) on the flow persistence value in four

watersheds, modelled in GenRiver over a 20-year time-period, based on actual rainfall records;

the left side panels show average water balance for each land cover scenario, the middle panels

the Fp values per year and land use, the right-side panels the derived frequency distributions
(best fitting Weibull distribution)

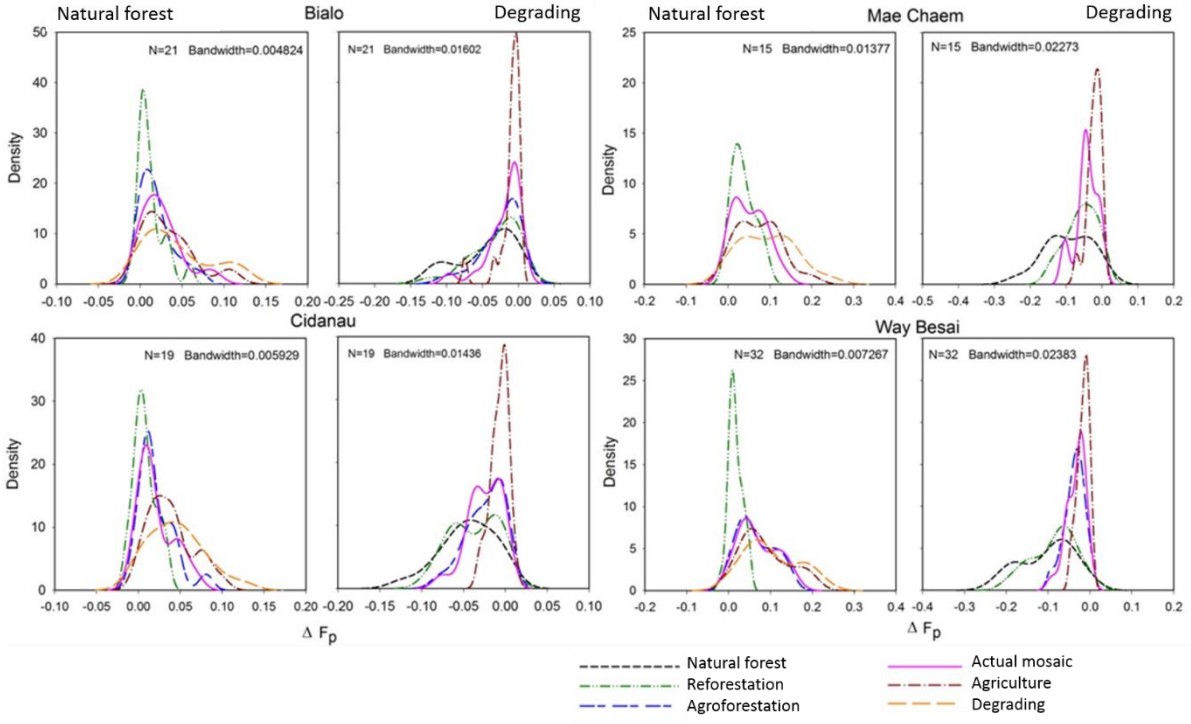


Figure 6. Frequency distribution of expected difference in $F_p$ in 'paired plot' comparisons where land
cover is the only variable; left panels: all scenarios compared to 'Reforestation', right panel: all
scenarios compared to degradation; graphs are based on a kernel density estimation (smoothing)
approach

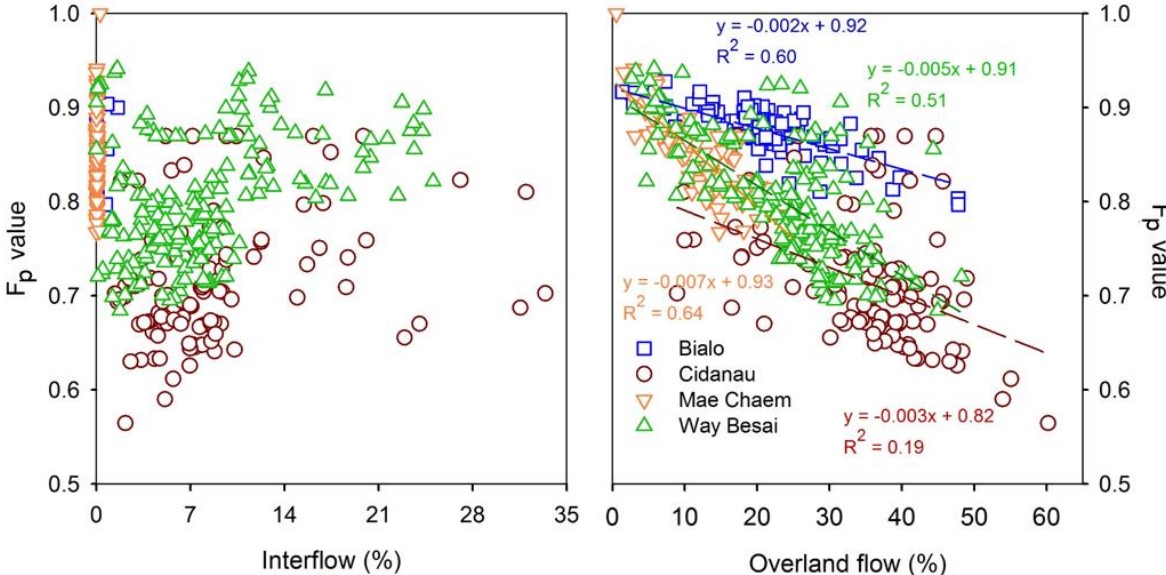


Figure 7. Correlations of $F_p$ with fractions of rainfall that take overland flow and interflow pathways

through the watershed, across all years and land use scenarios of Figure App2


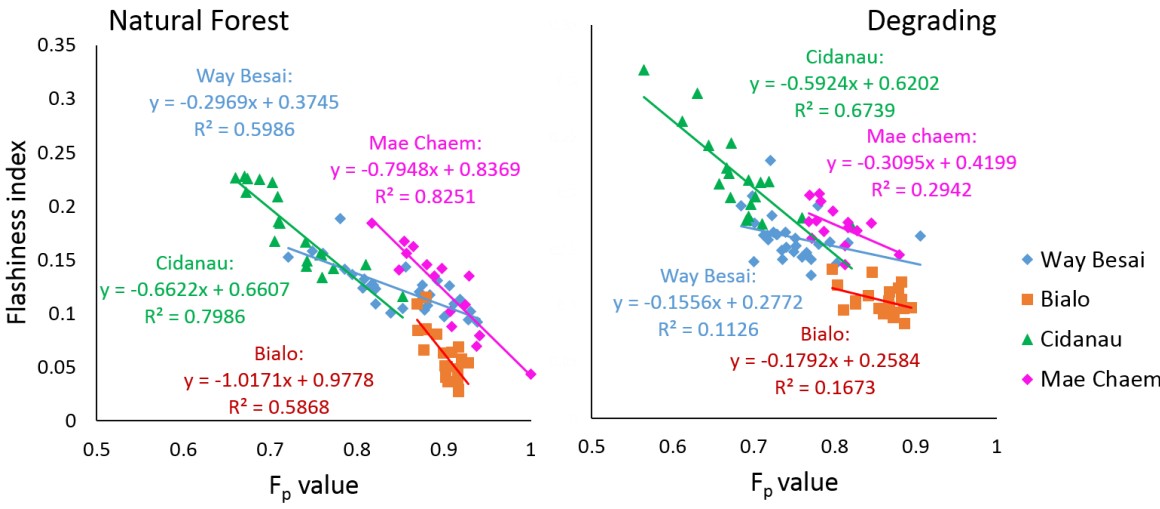


Figure 8. Relationship between $F_p$ value and R-B Flashiness index across years in foru Southeast Asian
watersheds under a 'natural forest' and 'degradation' scenario, simulated with the GenRiver model
Appendix 1. GenRiver model for effects of land cover on river flow
The Generic River flow (GenRiver) model (van Noordwijk et al., 2011) is a simple hydrological model
that simulates river flow based on water balance concept with a daily time step and a flexible spatial
subdivision of a watershed that influences the routing of water. The core of the GenRiver model is a
"patch" level representation of a daily water balance, driven by local rainfall and modified by the
land cover and land cover change and soil properties. The model starts accounting of rainfall or
/precipitation (P) and traces the subsequent flows and storage in the landscape that can lead to
either evapotranspiration (E), river flow (Q) or change in storage (ΔS) (Figure App1):
$P = Q + E + \Delta S$                                                                         [1]

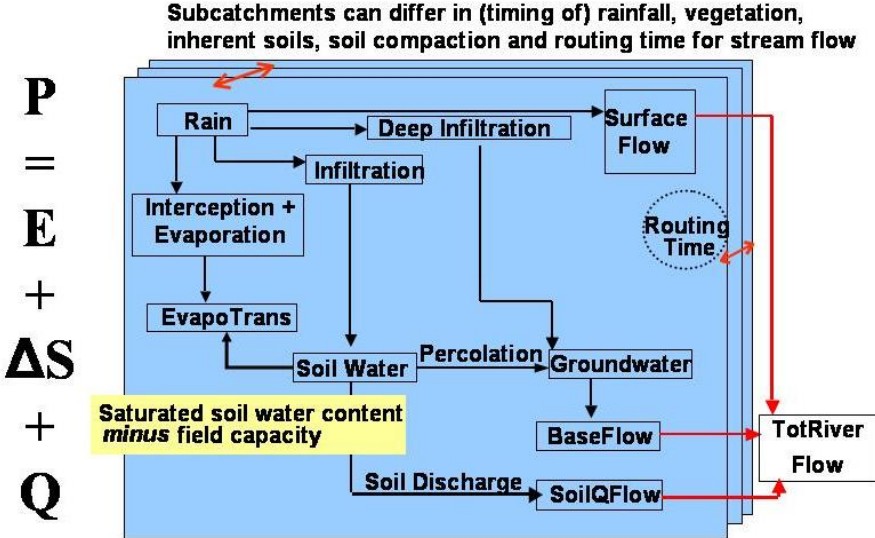

   **Figure** App1.Overview of the GenRiver model

The model may use measured rainfall data, or use a rainfall generator that involves Markov chain
temporal autocorrelation (rain persistence). The model can represent spatially explicit rainfall, with
stochastic rainfall intensity (parameters RainIntensMean, RainIntensCoefVar in Table 2) and partial
spatial correlation of daily rainfall between subcatchments. Canopy interception leads to direct
evaporation of an amount of water controlled by the thickness of waterfilm on the leaf area that
depends on the land cover, and a delay of water reaching the soil surface (parameter
RainMaxIntDripDur in Table 2). The effect of evaporation of intercepted water on other components
of evapotranspiration is controlled by the InterceptEffectontrans parameter that in practice may
depend on the time of day rainfall occurs and local climatic conditions such as windspeed)
At patch level, vegetation influences interception, retention for subsequent evaporation and delayed
transfer to the soil surface, as well as the seasonal demand for water. Vegetation (land cover) also
influences soil porosity and infiltration, modifying the inherent soil properties. Groundwater pool
dynamics are represented at subcatchment rather than patch level, integrating over the landcover
fractions within a subcatchment. The output of the model is river flow which is aggregated from
three types of stream flow: surface flow on the day of the rainfall event; interflow on the next day;
and base flow gradually declining over a period of time. The multiple subcatchments that make up
the catchment as a whole can differ in basic soil properties, land cover fractions that affect
interception, soil structure (infiltration rate) and seasonal pattern of water use by the vegetation.
The subcatchment will also typically differ in "routing time" or in the time it takes the streams and
river to reach any specified observation point (with default focus on the outflow from the
catchment). The model itself (currently implemented in Stella plus Excel), a manual and application
case studies are freely available (http://www.worldAgroforestry.org/output/genriver-generic-river-
model-river-flow ;van Noordwijk et al., 2011).

Appendix 2. Watershed-specific consequences of the land use change scenarios
The generically defined land use change scenarios (Table 4) led to different land cover proportions,
depending on the default land cover data for each watershed, as shown in Figure App2.

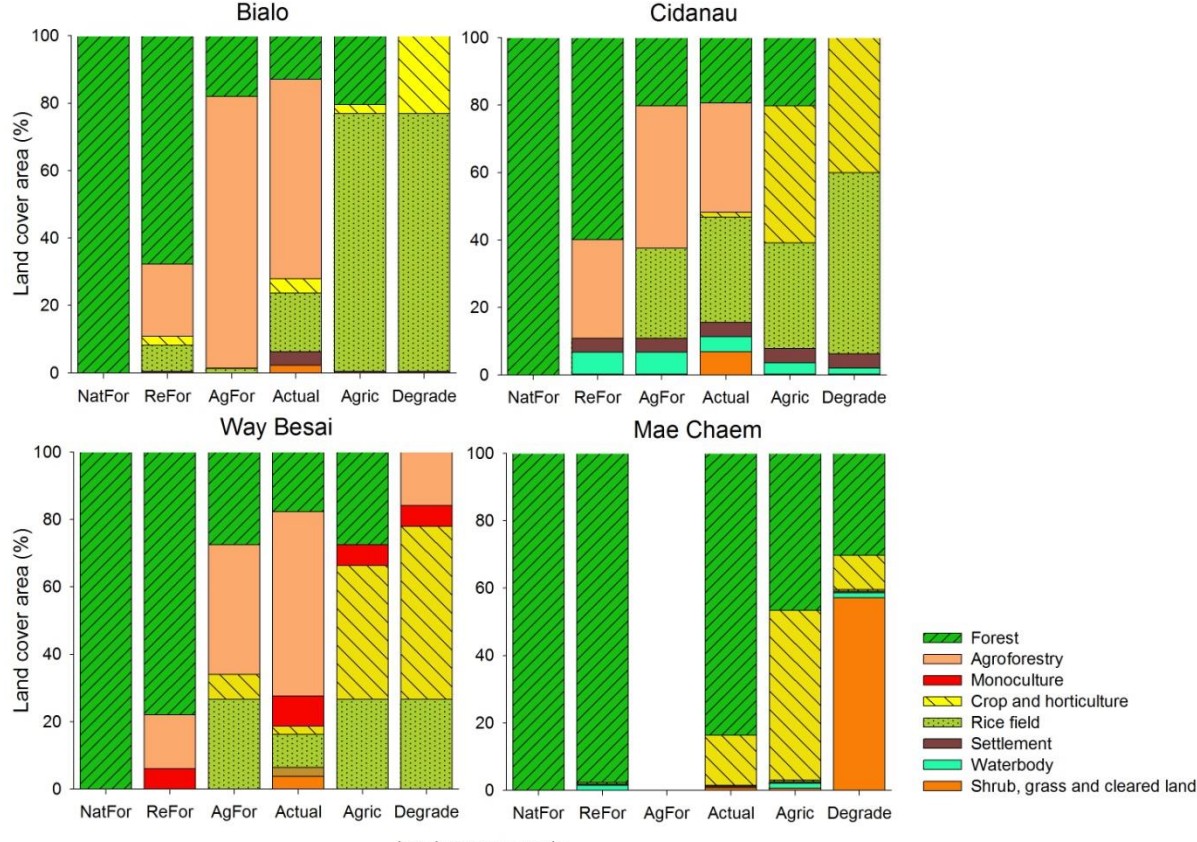

Figure App2. Land use distribution of the various land use scenarios explored for the four
watersheds (see Table 4)

Appendix 3. Example of a macro in R to estimate number of observation required using bootstrap
approach.

#The bootstrap procedure is to calculate the minimum sample size (number of observation) required
#for a significant land use effect on Fp
#bialo1 is a dataset contains delta Fp values for two different from Bialo watershed

#read data
bialo1 <- read.table("bialo1.csv", header=TRUE, sep=",")

#name each parameter
BL1 <- bialo1$ReFor
BL5 <- bialo1$Degrade

N = 1000 #number replication

n <- c(5:50) #the various sample size

J <- 46 #the number of sample size being tested (~ number of actual year observed in the dataset)

P15= matrix(ncol=J, nrow=R) #variable for storing p-value
P15Q3 <- numeric(J) #for storing p-Value at 97.5 quantile

for (j in 1:J) #estimating for different n

#bootstrap sampling
{
for (i in 1:N)
{
#sampling data
S1=sample(BL1, n[j], replace = T)
S5=sample(BL5, n[j], replace = T)

#Kolmogorov-Smirnov test for equal distribution and get the p-Value
KS15 <- ks.test(S1, S5, alt = c("two.sided"), exact = F) P15[i,j] <- KS15$p.value
}

#Confidence interval of CI
P15Q3[j] <- quantile(P15[,j], 0.975)

}

#saving P value data and CI

```
write.table(P15, file = "pValue15.txt") write.table(P15Q3, file = "P15Q3.txt")v
1570    /
```