# Peer review of "HESS-2015 -538, 31 Jan 2017"

_Hydrology and Earth System Sciences, 2015_

## Referee Comment (RC1) · Anonymous Referee #1 · 29 Feb 2016

The paper presents an interesting attempt to develop a simple measure of flow persistence and address the issue of the ability to detect and attribute variations in daily river flows to the effects of land-cover change in large catchments (river basins), a well-known issue in hydrology. The authors then use a model for estimating flow persistence to try to demonstrate how difficult it can be to identify the effects of land-use change in four tropical catchments. Their illustration of the sample sizes needed to identify effects is also interesting but difficult to interpret as too little information is given on the catchments used as examples. I am in full agreement that the issues of detectability and attribution are important for hydrologists to investigate because decision makers need evidence that catchment restoration can reduce flood risks and increase flow persistence by redirecting water from flow paths with rapid responses to rainfall to those with slow paths. This evidence can then be used to demonstrate that those benefits are being realised. The authors also note that these effects are well documented in hillslope and small catchment studies but there is little evidence of such effects at in large catchments (river basins). I have no problem with their argument that some form of measure of hydrological change is important, but (a) their method of deriving the persistence indicator and (b) of applying their model to the four catchments is not adequately explained. Overall the paper needs substantial work and possibly reworking in to two papers.

I have divided my review into two sections based on the two main components of the paper: (a) flow persistence and (b) flow change detection and attribution. There are some typographic errors but I have not gone into these as the paper needs rework.

A) The one part of the paper addresses both flood risk and flow buffering by measuring aspects of the flow responsiveness of a catchment using a simple index (Fp) of the flow persistence. Flow persistence is defined as identical to the 'recession constant' (pg 7 line 7). However, I would argue that flow persistence is only half the picture, what is needed is actually a measure of flow responsiveness to rainfall because flooding can depend on how rapidly the flow increases versus factors that constrain that flow and cause water levels to rise. Flow responsiveness is also directly influenced by antecedent wetness, rainfall event depth and duration, and other factors which they do discuss but do not seem to incorporate in their approach. Maybe I missed it, but I did not find a clear statement that Fp is only being calculated for the descending limb of a hydrograph. Yet this must be so because the range for Fp is constrained to the range 0-1. I would argue that to understand flood risk you also need to measure the ascending limb of the flows (the rapidity of the rise in response to rainfall patterns). I think would be probably be necessary to have indices for both the ascending and the descending limb as they are rarely symmetrical. Attempting to parameterise Fp for the various flow pathways seems to me to add complexity but not much insight given

their focus on large catchments where the relative importance of different flow paths can vary considerably across space and temporally. I did not find Figure 1 helpful as an illustration of the causal pathway or for placing the wide range of factors into their context. It has too much detail presented at the same level, is not adequately explained, and has too many terms that are not explained in the caption. A figure should be able to stand on its own with its caption, but in this case it does not, even with an extensive caption. A part of the problem is the use of two sets of arrows with different meanings in ways that are confusing. A more common way of representing this kind of diagram is a flow chart where the factors that influence (solid arrows) the (relative) magnitude of the water flows (hollow blue arrows) are represented in ways which make their role much clearer. I do not understand why rainfall is not made the start of the diagram and why the caption ends with number 0 rather than beginning there as one would intuitively expect. It is also not clear that land cover has various influences at both the "plot" level (whatever that may be) AND at the hillslope/landscape level. It is also not clear to me why there is a blue water flow arrow directly from rainfall to #2 and to #3 without passing through the landscape (and why #2 is in brackets). Why are some of the arrows broad and others not? Is the "triangle" to the left of human population density an arrow? Why are human population density and topography (subsidence) linked? What is the relevance of subsidence? Why is topography placed here and not within the sets #0 and #1 given its importance as a factor in the generation and flow of surface and subsurface runoff at both plot and hillslope levels? For a paper that attempts to explain how land cover changes affects catchment flow responses I find it inexplicable that there is almost no reference to: (a) the very extensive body of hillslope hydrology research into flow pathways and the temporal effects of different water partitioning and surface/subsurface on flow response to rainfall inputs; and (b) how hillslope responses might scale up to larger parts of catchments and large catchments. Even the brief mention of the different ways in which overland flow can be generated (e.g. Hortonian versus variable (saturated) source areas) fails to cite the original research papers and the insights they provide in the catchment responsiveness. Despite reading section 2

a few times I am still not entirely clear on the logic of the various deductions that are made about low flow, seasonality and the influence of varying Fp on the form of the hydrograph. Perhaps this is because the text is not always very clear. For example, the authors present the following (page 7 line 19 onwards):

"If we consider the sum of river flow over a sufficiently long period, we can expect $\Sigma Qt$ to closely approximate $\Sigma Qt\text{-}1$, and thus $\Sigma Qt = Fp\ \Sigma Qt\text{-}1 + \Sigma\varepsilon$ (equation 2) From this relationship we obtain a first way of estimating the Fp value if a complete hydrograph is available: $Fp = 1 - \Sigma\varepsilon/\ \Sigma Qt$ (equation 3)"

The only way I can derive equation 3 from 2 is to assume that $\Sigma Qt\text{-}1 = \Sigma Qt$ and so $\Sigma Qt$ can be substituted for $\Sigma Qt\text{-}1$. However, if this is so, then the only way equation 2 can hold is if Fp = 1 and $\Sigma\varepsilon = 0$. If this is so, how can this relationship then be used to estimate Fp? Or am I missing something here? In section 2.4 I assume that a model with a set input of daily rainfall and flow responses to that rainfall was used to create the ascending flow limbs so that Fp values could be used to generate the descending flow limb? And so that Fp could be varied? I also had similar difficulty in following parts of the methods section. For example, on pg 12 line 9 the term Qadd is abruptly introduced without an adequate explanation of its meaning. This is followed by the 'apparent Qadd' and Fp,try, again with no proper explanation. I should not have to go and find the paper cited (in fact a user manual) or to download the spreadsheet for an adequate explanation of the terms or to find a proper explanation of the FlowPer algorithm.

B) The second part of the paper deals with the application of the GenRiver model for assessing the impacts of land cover on river flow and its attribution and detectability. The entire model and its application is only introduced in the methods but its structure and use should really be described already in the introduction. The model is said to be spatially explicit but it is not clear how that is realised in practice (i.e. it a distributed model?). Tables 1-4 do not provide an adequate description of the study catchments – what does dominant land cover mean? Although the authors note the importance

of knowing what changes where in a catchment in relation to flow paths and times and attributing responses to changes and other factors, Table 1 does not give any indication even of how much of each land cover there is the baseline situation. Why not provide summaries or maps? Nor are we given information on where, in relation to this baseline state, the changes in land cover are made for the different scenarios. Why not provide a summary or a map? In Table 4 there is a repeated use of "some" in describing the changes made. This to me is not acceptable. We are not given an adequate explanation of how the single values of each of the 13 parameters of the GenRiver (Table 2) were obtained. Those parameter values are all ones that would vary a great deal spatially and with different land cover types (e.g. interception), but only a single value is given with no indication of their variability in the study catchments or how representative each value is. Providing definitions of the terms in a user manual the reader would have to look up is simply not acceptable. Table 3 also gives values for three important parameters for each of the land cover types with no explanation of what their sources and ranges are (BTW surely interception [Table 2] differs between forest and annual crops and so is land cover specific?). Table 3 also introduces the term relative drought threshold with no explanation of what it means and how the model uses it. The legitimacy, accuracy and representativeness of these values, together with the land cover changes, are critical to our confidence in the model outputs and thus in the analysis of the detectability and attribution of the changes in flows to changes in land cover. A study should be repeatable and this hypothetical modelling exercise certainly is not given the information included in the paper.

In summary I am not entirely sure what to recommend overall. The idea of deriving a simple but robust measure of flow change (i.e. flow responsiveness) which can be causally related to land cover changes is sound, and necessary. Flow persistence (Fp, recession) is an interesting measure and can be related to changes in the relative importance of different water flow paths, but it is also evident that it is not straightforward to derive and could be masked by the effects of location and catchment heterogeneity. I do think that a measure of the flow recession is not sufficient, the nature of the

whole response to rainfall needs to be assessed for flood risk. The flow persistence component of the paper needs careful thought to make sure that the measure(s) are clearly and thoroughly explained. Even so, I still am left with the question of whether a simpler approach would not be to examine the slopes of the flow recession curves (in relation to rainfall event sizes and sequences [antecedent conditions]) for possible shifts due to land cover changes. Alternatively, using shifts in the flow duration curves as measures of changes in the relative importance of flow pathways, as has been done elsewhere, would be more effective and understandable. Another alternative would be to use the relationship between rainfall event sizes and sequences (e.g. antecedent wetness) and flow response to those events and sequences to infer changes.

The modelling component needs a lot more information to back up the chosen parameter values for both the hydrological (Table 2) and land cover-specific values, as well as specific information on the extent of the land cover changes and their spatial configurations. It also needs to provide information on how well the outputs it generates for the different land cover types compare with the findings of other studies (i.e. how well does the model perform). Overall I need more information on the model structure and setup to interpret how well it performs in this application. This would require expanding the paper substantially.

Overall, my conclusion is that perhaps this paper attempts to cover too much ground and should be two papers: - One on the issue of catchment responsiveness to rainfall as a measure of land cover change, including flow persistence - One on modelling of the effects of land cover change on river flow responses and the difficulties of detecting and attributing changes in flow responsiveness to changes in land cover (and relating this back to the changes in the relative importance of flow paths linked to the changes in land cover).

---

## Referee Comment (RC2) · Anonymous Referee #2 · 25 Apr 2016

As I understand the article, the authors are attempting to develop a single measure of watershed health called 'flow persistence' (Fp). This Fp parameter measures the volatility of daily river flow in response to land cover change within large catchments. One of the key objectives of the study is to determine the value of specific land cover types in terms of flood mitigation. The study itself is broken down into two phases: (1) the derivation of a river flow algorithm, and (2) the application of the algorithm within four watersheds with different rainfall and land cover characteristics. The key points that need to be addressed include: 1. A better justification that flow predictability does in fact correspond with watershed health; 2. A better explanation of the flow persistence derivation; and 3. A much more thorough explanation of the Fp algorithm application

within the four catchments. The study addresses a significant point of contention in the literature: the influence of land cover (particularly forests) and flooding at the watershed scale. If the Fp model is properly justified and performs adequately, then it would undoubtedly increase our understanding of the linkages between land cover and flood risk. The benefits of such an approach are clear as it would make for a much more parsimonious model of river flow that would greatly enhance the monitoring and prioritization of specific landscape management decisions. That said, the paper requires substantial work to adequately address the points above and may need to be split into two separate papers. I will address the three points I mentioned above in greater detail below. There are quite a few of typographical and grammatical errors in the paper, but I will leave these alone for now as the paper requires substantial work.

Point 1 In the paper the authors use persistence, predictability, and watershed health interchangeably. One of the key assumptions of the paper and previous watershed rehabilitation efforts is that increasing the presence of natural land covers (particularly forests and wetlands) will restore the natural flood regime with lower peak flows and less damaging flood events. The authors do a good job of documenting previous studies that have illustrated the complexity of the linkage between reforestation and river flows. Moreover, the ability of wetlands and riparian forests to absorb rainfall, slow streamflow, and attenuate peak flows is supported by many studies and is fairly well understood. However, these types of stream corridor ecosystems also require a particular type of disturbance regime that creates opportunities for species specific recruitment processes and establishes landscape and topographic heterogeneity that are critical components of watershed health. These disturbance regimes are often characterized by variable flow patterns with various flood magnitudes required for specific types of ecosystem level processes. Most efforts to create a stable and predictable flood regime have been anthropogenic in origin through engineered based interventions like dams, and retention and detention ponds, which are also some of the primary drivers behind the degradation of watershed health. A perfectly stable flow regime could, theoretically, be established by a highly integrated system of engineered solutions (albeit until

they are either overwhelmed by a storm or undermined by system failure) within a very ecologically degraded watershed. Likewise, there could, theoretically, be examples of ecologically well connected and healthy watersheds with fairly volatile and unpredictable daily flow regimes. To overcome this, the authors need to discuss what exactly watershed health means and whether or not a predictable flow regime is the product of an ecosystem service. I could see an argument in which the shape of a storm specific hydrograph within a healthy watershed should be fairly predictable, however, to study this would preclude the advantages proposed by this paper (i.e. the application of the Fp algorithm in data sparse regions). I also agree that more human development and less natural systems generally leads to more flashy river flows as a result of decreased buffering capacity, however this study examines flow rates at daily intervals which washes out the ability to assess this linkage. Or maybe I'm missing something. The authors do point out later in the paper that Fp of zero (i.e. low predictability) would be the result of erratic rainfall (page 7 first paragraph). This is somewhat confusing because most of the introduction and discussion is focused on using Fp as a way to summarize "complex land use mosaics". Two paragraphs later the authors state "a decrease of Fp indicates watershed degradation." So how much of the decrease in Fp is explained by watershed degradation as opposed to just more erratic rainfall? I know the authors say that the GenRiver model is spatially explicit, but this is a little vague. Does this mean that spatial autocorrelation in precipitation is controlled for or that the model is spatially distributed? I understand that to have an Fp equal to 0 would require erratic rainfall, but the authors need to be consistent when describing what proxy measurements that Fp is suitable for. Figure 1 gets at the interconnection between many different elements that influence the hydrological cycle, and the authors break up the components into ecosystem structure, function, and human land use/perceived ecosystem service. However, I find the figure difficult to navigate and poorly described in the study. The different color arrows with different shades and outlines is one of main culprits of the confusion. The graphic needs to be simplified, it should probably start with rainfall, and terms like "plot-level" should either be defined or removed.

Point 2 An Fp value ranging from 0-1 essentially represents the buffering capacity of the watershed, but there are also characteristics that influence how rapidly water reaches the stream. In this sense, Fp is only represents half of the picture. The authors go on to create separate Fp's for each flow pathway, which is probably necessary for large catchments as each flow pathway likely do have large influences over space and time. However, this seems to be overcomplicating a model that was originally being created out of need for greater parsimony. If these pathway specific indices are necessary, then more discussion and justification is required in the text. The authors use vague language that needs some more clarification. Line 19 on page 7 contains "flow over a sufficiently long period". What is a sufficiently long period? Wouldn't a sufficiently long period wash out the "flashy" fluctuations that the authors are trying to explain with changing land cover/watershed degradation? If the 'sufficiently long period' is preventing what the study is attempting to explain, then I do not see how equation three could be derived. Maybe I'm missing something, but wouldn't the stochastic term represent all unexplained variations in the predicted river flow? Line 28 on page 7 explains that the stochastic term is equal to the sum of peak flows. Couldn't other unpredicted river flows have other anthropogenic origins that contribute to the river flow stochasticity (e.g. dam operations/failure, irrigation, urban water use, etc.)? The authors also mention new variables like Qadd and Fp,try without adequately discussing what they actually represent.

Point 3 Figures 2-9 were very readable and for the most part stand on their own, however the table were poorly formatted and vague. Table 1 does not provide land cover proportions by land cover type (other than forest). Percent developed land, existing flood control infrastructure, and population would all be helpful information. Not sure what 'dominant land cover' means. Do these watersheds have a history of damaging floods? Why were the parameters in table 2 chosen? The authors do not provide an adequate discussion of how these parameter values were estimated. Why were the defaults in GenRiver used for each of the land cover types in table 3? What process or methodology did GenRiver use to estimate these values? What does 'relative drought

threshold' mean? The use of the word 'some' in table 4 is simply too vague when describing the scenarios. The reader is left wondering what the magnitude of change would be within each of these scenarios. All of the information in tables 1-4 are critical components of the GenRiver model. The legitimacy and accuracy of the model is weak without proper documentation and justification of the underlying model parameters. The authors must correct this if we are to have any confidence in the results. Table 5 and its corresponding discussion regarding the sample sizes required to reject a null hypothesis is interesting, but not enough information was given to make this section clear. The methodology is clear enough, but the implications were not really discussed. The statement beginning on line 21 on page 14: "In practice, that means that empirical evidence that survives statistical tests will not emerge, even though effects on watershed health are real" is vague and needs some more clarification. Lastly, in table 6, the authors provide broken links to the detailed reports of rainfall and river flow data. Moreover, there is very little discussion on the accuracy and metadata of each of these data sources, all of which have different origins.

Summary Overall, I think that this study addresses a critical knowledge gap with important implications. However, the conceptual foundation regarding watershed health and flow predictability requires a closer examination. The derivation of Fp and the process used to create the GenRiver model parameters needs more discussion, clarification, and justification if the reader is to have any confidence in the results. I think that if the authors were to accomplish these revisions then the paper would simply be too long and cover too much ground. Breaking the research into two separate papers is probably a better course with one focusing more on the conceptualization and creation of the Fp term and one on the application of it within the GenRiver framework.

---

## Author Comment (AC1) · 24 May 2016

Interactive comment on "Flood risk reduction and flow buffering as ecosystem services: a flow persistence indicator for watershed health" by M. van Noordwijk et al.

With both reviews showing interest in the line of argument, but indicating incomplete understanding of the various steps in the analysis, we will follow the suggestions of both reviewers to split the manuscript into two, the first part describing the theory (recursive flow models and its parameters), the second applications to a number of watersheds of contrasting characteristics. Working titles would be: "Flood risk reduction and flow buffering as ecosystem services: I. Theory on a flow persistence indicator for watershed health, II. Applications in four contrasting watersheds in Southeast Asia". We

will benefit from the many specific comments by both reviewers in clarifying the overall flow of the argument and the details of its presentation.

1. Anonymous Referee #1

R1.1 The paper presents an interesting attempt to develop a simple measure of flow persistence and address the issue of the ability to detect and attribute variations in daily riverflows to the effects of land-cover change in large catchments (riverbasins), a well-known issue in hydrology. Author's response AR1.1: Thank you for the positive suggestions and interest. We realize, however, that we may need to be more clear on the primary aim: exploring the 'information content' of an empirically derived indicator of watershed health that does not require data other than the temporal pattern of river flow at a point of interest. R1.2 The authors then use a model for estimating flow persistence to try to demonstrate how difficult it can be to identify the effects of land-use change in four tropical catchments. Their illustration of the sample sizes needed to identify effects is also interesting but difficult to interpret as too little information is given on the catchments used as examples. AR1.2 We will have to provide further detail on the catchments, and the possible reasons for their differences in response to land cover change – see below where Tables 2-4 are discussed. R1.3 I am in full agreement that the issues of detectability and attribution are important for hydrologists to investigate because decision makers need evidence that catchment restoration can reduce flood risks and increase flow persistence by redirecting water from flow paths with rapid responses to rainfall to those with slow paths. This evidence can then be used to demonstrate that those benefits are being realised. AR1.3 Thanks, this is indeed the overall direction of the argument. R1.4 The authors also note that these effects are well documented in hillslope and small catchment studies but there is little evidence of such effects at in large catchments (river basins). AR1.4 Agreed R1.5 I have no problem with their argument that some form of measure of hydrological change is important, but (a) their method of deriving the persistence indicator and (b) of applying their model to the four catchments is not adequately explained. AR1.5 We will

have to provide further detail then – as discussed below R1.6 Overall the paper needs substantial work and possibly reworking in to two papers. I have divided my review into two sections based on the two main components of the paper: (a) flow persistence and (b) flow change detection and attribution. There are some typographic errors but I have not gone into these as the paper needs rework. AR1.6 Thanks for the suggestion – as the method is not easily explained without practical examples, we would prefer to keep the full story in a single paper, but the suggested structure can help in the rewrite. R1.7 A) The one part of the paper addresses both flood risk and flow buffering by measuring aspects of the flow responsiveness of a catchment using a simple index (Fp) of the flow persistence. Flow persistence is defined as identical to the 'recession constant' (pg 7 line 7). However, I would argue that flow persistence is only half the picture, what is needed is actually a measure of flow responsiveness to rainfall because flooding can depend on how rapidly the flow increases versus factors that constrain that flow and cause water levels to rise. Flow responsiveness is also directly influenced by antecedent wetness, rainfall event depth and duration, and other factors which they do discuss but do not seem to incorporate in their approach. AR1.7 We agree with the reviewer that flow persistence is focused on half the story (peak flows and subsequent decay), and that the speed at which peak flows are attained matters beyond what the peak itself is. Details on this speed will depend heavily on the specific space-time pattern of rainfall, and will require flow measurements at least hourly time-scale. In many cases daily records are the only thing that exists empirically, and we can't say much about this first part. Detailed models that use space-time pattern analysis can probably –provide reasonable inference, but will require many parameters, beyond what our 'parsimonious' targets allow. We will expand the discussion on these aspects.

R1.8 Maybe I missed it, but I did not find a clear statement that Fp is only being calculated for the descending limb of a hydrograph. Yet this must be so because the range for Fp is constrained to the range 0-1. I would argue that to understand flood risk you also need to measure the ascending limb of the flows (the rapidity of the

rise in response to rainfall patterns). I think would be probably be necessary to have indices for both the ascending and the descending limb as they are rarely symmetrical. Zz AR1.8 On further thought, as may need to start from a more general case where Fp varies along the hydrological year, and only the dry season Fp equals the recession constant, as normally defined. We agree that the Fp parameter (at daily scale) alone is not sufficient to predict flood dynamics at shorter timespans (e.g. hours), as details of spatial and temporal storm patterns interact with characteristics of the streambed, beyond what Fp captures. However, the finding that the proportion of fresh rainfall (minus soil water storage capacity linked to preceding Et) that comes down as riverflow is (1-Fp), allows us to infer an important component of flood predictions: the peak daily flow volume (given rainfall). We agree that a further empirical parameter at hourly (or similar timescale) can add further value – but for the empirical data sets we used only daily records are available. Indeed we don't assume that the ascending and descending limbs are symmetrical. R1.9 Attempting to parameterise Fp for the various flow pathways seems to me to add complexity but not much insight given their focus on large catchments where the relative importance of different flow paths can vary considerably across space and temporally. AR1.9 This part is presented as an aid in the interpretation of the aggregate Fp, not as a way of empirically deriving it. This probably will have to be more clearly stated. R1.10 I did not find Figure 1 helpful as an illustration of the causal pathway or for placing the wide range of factors into their context. It has too much detail presented at the same level, is not adequately explained, and has too many terms that are not explained in the caption. A figure should be able to stand on its own with its caption, but in this case it does not, even with an extensive caption. A part of the problem is the use of two sets of arrows with different meanings in ways that are confusing. A more common way of representing this kind of diagram is a flow chart where the factors that influence (solid arrows) the (relative) magnitude of the water flows (hollow blue arrows) are represented in ways which make their role much clearer. I do not understand why rainfall is not made the start of the diagram and why the caption ends with number 0 rather than beginning there as one would

intuitively expect. It is also not clear that land cover has various influences at both the "plot" level (whatever that may be) AND at the hillslope/landscape level. It is also not clear to me why there is a blue water flow arrow directly from rainfall to #2 and to #3 without passing through the landscape (and why #2 is in brackets). Why are some of the arrows broad and others not? Is the "triangle" to the left of human population density an arrow? Why are human population density and topography (subsidence) linked? What is the relevance of subsidence? Why is topography placed here and not within the sets #0 and #1 given its importance as a factor in the generation and flow of surface and subsurface runoff at both plot and hillslope levels? AR1.10 We accept the limitations of Fig 1 in its current form and will improve it based on the comments made. It is meant at conceptual level, rather than as specification of a quantifiable model. Subsidence due to groundwater extraction in urban areas of high population density is a specific problem for a number of cities built on floodplains (such as Jakarta and Bangkok). It is a rather specific situation, but economically important flooding risks that were at first attributed to changes in upland land use are now understood to be generated in the urban areas. We'll add a brief explanation and references to this point of detail. R1.11 For a paper that attempts to explain how land cover changes affects catchment flow responses I find it inexplicable that there is almost no reference to: (a) the very extensive body of hillslope hydrology research into flow pathways and the temporal effects of different water partitioning and surface/subsurface on flow response to rainfall inputs; and (b) how hillslope responses might scale up to larger parts of catchments and large catchments. Even the brief mention of the different ways in which overland flow can be generated (e.g. Hortonian versus variable (saturated) source areas) fails to cite the original research papers and the insights they provide in the catchment responsiveness. AR1.11 Our primary purpose with the paper is to evaluate the information content of a metric that is derived form observations of river flow alone – we are indeed aware of the large body of work (and various models) at hillslope scale, and will add some references, but the paper is not meant to be a review of all we know about 'floods', but to evaluate a very parsimonious model, that can in a single

index capture important first-order predictive value of the influence the watershed has, as modifier of stochastic rainfall. R1.12 Despite reading section 2 a few times I am still not entirely clear on the logic of the various deductions that are made about low flow, seasonality and the influence of varying Fp on the form of the hydrograph. Perhaps this is because the text is not always very clear. For example, the authors present the following (page 7 line 19 onwards): "If we consider the sum of river flow over a sufficiently long period, we can expect $\Sigma Q_t$ to closely approximate $\Sigma Q_{t-1}$, and thus $\Sigma Q_t = F_p \Sigma Q_{t-1} + \Sigma\varepsilon$ (equation 2) From this relationship we obtain a first way of estimating the Fp value if a complete hydrograph is available: $F_p = 1 - \Sigma\varepsilon / \Sigma Q_t$ (equation 3)" The only way I can derive equation 3 from 2 is to assume that $\Sigma Q_{t-1} = \Sigma Q_t$ and so $\Sigma Q_t$ can be substituted for $\Sigma Q_{t-1}$. However, if this is so, then the only way equation 2 can hold is if $F_p = 1$ and $\Sigma\varepsilon = 0$. If this is so, how can this relationship then be used to estimate Fp? Or am I missing something here? In section 2.4 I assume that a model with a set input of daily rainfall and flow responses to that rainfall was used to create the ascending flow limbs so that Fp values could be used to generate the descending flow limb? And so that Fp could be varied? I also had similar difficulty in following parts of the methods section. For example, on pg 12 line 9 the term Qadd is abruptly introduced without an adequate explanation of its meaning. This is followed by the 'apparent Qadd' and Fp,try, again with no proper explanation. I should not have to go and find the paper cited (in fact a user manual) or to download the spread-sheet for an adequate explanation of the terms or to find a proper explanation of the FlowPer algorithm. AR1.12 This is a crucial point in the derivation,

If we indeed assume $\Sigma Q_t = \Sigma Q_{t-1}$, we obtain $\Sigma Q_t = F_p \Sigma Q_t + \Sigma\varepsilon$ and hence $\Sigma\varepsilon = (1 - F_p)(\Sigma P - \Sigma E)$. The easiest way to obtain this relationship at the level of annual sums, is to have it hold true at event level: $\varepsilon = (1 - F_p)(P - E_x)$ with the caveat that the way $\Sigma E$ is partitioned over terms calculated at each day with rain ($\Sigma E_x$) may require some nuance (where not all antecedent ET is compensated in a single storm on soils that don't easily rewet, for example).

The key point, however, is that the stochastic (P – Ex) term is to be multiplied with (1 – Fp), which leads to direct influences on peak flows (if Fp does not vary with Q).

We will have to be more explicit in the way Qadd is calculated for each Fp,try value: Every pair (Qt, Qt+1) yields an estimate of Qadd derived from Qadd = Qt+1 – Fp,try Qt Each Fp,try value thus generates a frequency distribution of inferred Qadd values (some of which will be negative for relatively high Fp,try values), and the algorithm retains the value that minimizes Var(Qadd).

We will adjust the text to make this clearer, while the spreadsheet version is available for anybody who wants to try it on empirical data. R1.13 B) The second part of the paper deals with the application of the GenRiver model for assessing the impacts of land cover on river flow and its attribution and detectability. The entire model and its application is only introduced in the methods but its structure and use should really be described already in the introduction. The model is said to be spatially explicit but it is not clear how that is realised in practice (i.e. it a distributed model?). AR1.13 The use of the model to explore Fp derived from hydrographs for scenarios other than current land use should indeed be more explicitly announced in the introduction – but it remains a 'tool' for exploring Fp properties of hydrographs, rather than being a focus on its own. If we were to separate the manuscript into two, the GenRiver part would probably have to come first, so that the Fp discussion can use its results. With a full specification of the model available for who wants to get into the detail, we will increase the description of key features that likely influence the results obtained. R1.14 Tables 1-4 do not provide an adequate description of the study catchments – what does dominant land cover mean? Although the authors note the importance of knowing what changes where in a catchment in relation to flow paths and times and attributing responses to changes and other factors, Table 1 does not give any indication even of how much of each land cover there is the baseline situation. Why not provide summaries or maps? Nor are we given information on where, in relation to this baseline state, the changes in land cover are made for the different scenarios. Why not provide a summary or a

map? In Table 4 there is a repeated use of "some" in describing the changes made. This to me is not acceptable. AR1.14 Thanks for the suggestion – we will add a map that visualizes the baseline situation, and a table of what the scenario's mean in local context. Supplementary material that gives all the requested details, while maintaining the overall flow of the current manuscript is probably our preference at this stage. A point of warning may be needed here, as the case study catchments have not been subject to a multi-year intensive measurement campaign of the type that reviewer probably finds necessary to fully trust results. We do provide the level. Of correspondence between Fp's derived from measured and modelled hydrographs, and on that basis present the further model results as illustrations of what can be expected, rather than as statements of fact. R1.15 We are not given an adequate explanation of how the single values of each of the 13 parameters of the GenRiver (Table 2) were obtained. Those parameter values are all ones that would vary a great deal spatially and with different land cover types (e.g. interception), but only a single value is given with no indication of their variability in the study catchments or how representative each value is. Providing defiżnitions of the terms in a user manual the reader would have to look up is simply not acceptable. Table 3 also gives values for three important parameters for each of the land cover types with no explanation of what their sources and ranges are (BTW surely interception [Table 2] differs between forest and annual crops and so island cover specific?). Table3 also introduces the term relative drought threshold with no explanation of what it means and how the model uses it. The legitimacy, accuracy and representativeness of these values, together with the land cover changes, are critical to our confidence in the model outputs and thus in the analysis of the detectability and attribution of the changes in flows to changes in land cover. A study should be repeatable and this hypothetical modelling exercise certainly is not given the information included in the paper. In summary I am not entirely sure what to recommend overall. The idea of deriving a simple but robust measure of flow change (i.e. flow responsiveness) which can be causally related to land cover changes is sound, and necessary. AR1.15 We will provide full specification of the parameter values used for the calculations (and store runs in a data depository), while the model is freely downloadable, so we'll meet reasonable standards of repeatability – but of course further tests of the Fp summary statistic on hydrographs obtained with other models for other situations are likely needed before this method can be more widely accepted by the community. We hope that the discussion gives a fair assessment of the level of evidence, avoiding an oversell. R1.16 Flow persistence (Fp, recession) is an interesting measure and can be related to changes in the relative importance of different water flow paths, but it is also evident that it is not straightforward to derive and could be masked by the effects of location and catchment heterogeneity. I do think that a measure of the flow recession is not sufficient, the nature of the whole response to rainfall needs to be assessed for flood risk. The flow persistence component of the paper needs careful thought to make sure that the measure(s) are clearly and thoroughly explained. Even so, I still am left with the question of whether a simpler approach would not be to examine the slopes of the flow recession curves (in relation to rainfall event sizes and sequences [antecedent conditions]) for possible shifts due to land cover changes. Alternatively, using shifts in the flow duration curves as measures of changes in the relative importance of flow pathways, as has been done elsewhere, would be more effective and understandable. AR1.16 It seems that reviewer presents a multi-parameter description as 'simpler' than our single metric? Again, our focus is to assess the information content of a metric of flow predictability that appears to align well with the way downstream observers describe and experience changes in the conditions in upper watersheds. If our primary focus would be to assess the effects of land use change on flood risks as such, for any of the catchments studied or for the wider geographic domain in its totality, we would probably embark on further data collection... A key point here is that in many situations (historical) rainfall data are inadequate – at best a few point records are available, but no spatially weighted average, and there are many degrees of freedom in getting reasonable answers for wrong reasons in multiparameter models. R1.17 Another alternative would be to use the relationship between rainfall event sizes and sequences (e.g. antecedent wetness)

and flow response to those events and sequences to infer changes. The modelling component needs a lot more information to back up the chosen parameter values for both the hydrological (Table 2) and land cover-specific values, as well as specific information on the extent of the land cover changes and their spatial configurations. It also needs to provide information on how well the outputs it generates for the different land cover types compare with the findings of other studies (i.e. how well does the model perform). Overall I need more information on the model structure and setup to interpret how well it performs in this application. This would require expanding the paper substantially. AR1.17 We agree that the current description of the GenRiver model is not sufficient to fully compare its performance with the substantial range of other models, most of which require considerably more parameters. We do believe that the exploration of how hydrographs for alternative land use scenarios can be summarized in the Fp statistic adds value to our discussion on what we can and cannot expect of this parameter, and how it can play a role as 'metric of watershed quality', as the title emphasizes. R1.18 Overall, my conclusion is that perhaps this paper attempts to cover too much ground and should be two papers: - One on the issue of catchment responsiveness to rainfall as a measure of land cover change, including flow persistence - One on modelling of the effects of land cover change on river flow responses and the difficulties of detecting and attributing changes in flow responsiveness to changes in land cover (and relating this back to the changes in the relative importance of flow paths linked to the changes in land cover). AR1.18 Maybe reviewer has read more into the case studies than we intended – we agree that more detail (in an appendix) of the model results is needed, while a more comprehensive assessment of what land use change can mean in the specific locations waits further study.

Anonymous Referee #2 R2.1. Summary Overall, I think that this study addresses a critical knowledge gap with important implications. However, the conceptual foundation regarding watershed health and flow predictability requires a closer examination. The derivation of Fp and the process used to create the GenRiver model parameters needs more discussion, clarification, and

justification if the reader is to have any confidence in the results. I think that if the authors were to accomplish these revisions then the paper would simply be too long and cover too much ground. Breaking the research into two separate papers is probably a better course with one focusing more on the conceptualization and creation of the Fp term and one on the application of it within the GenRiver framework.

AR2.1 Thanks for the interest, we will follow the suggestion to split the manuscript into two parts "I. Theory, II. Applications", and appreciate the detailed suggestions and comments.

R2.2. As I understand the article, the authors are attempting to develop a single measure of watershed health called 'flow persistence' (Fp). This Fp parameter measures the volatility of daily river flow in response to land cover change within large catchments. AR2.2 Maybe the manuscript needs to be more clear in the steps involved: Fp is a measure of 'volatility' (or its complement), that can be used to quantify one aspect of the impacts, at multiple scales, of land cover change. R2.3. One of the key objectives of the study is to determine the value of specific land cover types in terms of flood mitigation. The study itself is broken down into two phases: (1) the derivation of a river flow algorithm, and (2) the application of the algorithm within four watersheds with different rainfall and land cover characteristics. AR2.3 In the current manuscript the Fp definition in terms of a recursive river flow model is presented as step 0, before the empirical steps on how actual (or simulated) flow data can be used to derive an estimate of Fp (step 1), illustrated with applications in four case studies (step 2). R2.4. The key points that need to be addressed include: 1. A better justification that flow predictability does in fact correspond with watershed health; 2. A better explanation of the flow persistence derivation; and 3. A much more thorough explanation of the Fp algorithm application within the four catchments. AR2.4 We thank reviewer for the positive suggestions and will try to clarify these points in a resubmission. "does in fact correspond with watershed health", is however a complex question, as "watershed health" hasn't been satisfactorily quantified in absolute terms anywhere. What we claim

is that changes in Fp, towards lower values match "degradation" and towards higher values match "restoration", from a downstream perspective. The metric matches a common way of describing degradation as loss of predictability. In unpacking the concept, however, we find that the specific pattern of rainfall in a given year interacts with the condition of the watershed in generating a river flow pattern, as captured in Fp. Our conclusion that multiple years of "paired catchment" type data (different watershed conditions, same rainfall pattern) are needed to be reject null-hypotheses that land cover use effects are neutral. So – we qualify our claim that Fp is an "indicator", not a "measure" of watershed health. But, so far we don't know of a better simple indicator. We accept that steps 2 and 3 need improvement in terms of clarity and documentation. Splitting the manuscript in two parts, as suggested by reviewer 1 will help us do so.

R2.5. The study addresses a significant point of contention in the literature: the influence of land cover (particularly forests) and flooding at the watershed scale. If the Fp model is properly justified and performs adequately, then it would undoubtedly increase our understanding of the linkages between land cover and flood risk. The benefits of such an approach are clear as it would make for a much more parsimonious model of river flow that would greatly enhance the monitoring and prioritization of specific landscape management decisions. AR2.5 We thank reviewer for the interest – indeed our target is a key parameter for a parsimonious model that can communicate key functional property of the way a watershed functions, given variable rainfall. R2.6. That said, the paper requires substantial work to adequately address the points above and may need to be split into two separate papers. I will address the three points I mentioned above in greater detail below. There are quite a few of typographical and grammatical errors in the paper, but I will leave these alone for now as the paper requires substantial work. AR2.6 As reviewer 1 also suggested splitting the paper along similar lines, we will follow this suggestion. R2.7. Point 1 In the paper the authors use persistence, predictability, and watershed health interchangeably. AR2.7 We aim to use "persistence" as a first descriptor of what Fp measures (the part of todays flow that can be counted on for tomorrow, regardless of additional precipitation); 1-Fp controls
the part of new rainfall that adds to the stream, and as such controls the predictabil-ity of overall flow; the way that Fp is an indicator of watershed health is at a further level of interpretation. We will scrutinize which words are used where in the process. R2.8. One of the key assumptions of the paper and previous watershed rehabilitation efforts is that increasing the presence of natural land covers (particularly forests and wetlands) will restore the natural flood regime with lower peak flows and less dam-aging flood events. AR2.8 Actually we take this as testable hypothesis rather than a priori assumption R2.9. The authors do a good job of documenting previous stud-ies that have illustrated the complexity of the linkage between reforestation and river flows. Moreover, the ability of wetlands and riparian forests to absorb rainfall, slow streamflow, and attenuate peakflows is supported by many studies and is fairly well understood. However, these types of stream corridor ecosystems also require a particular type of disturbance regime that creates opportunities for species specific recruitment processes and establishes landscape and topographic heterogeneity that are critical components of watershed health. These disturbance regimes are often characterized by variable flow patterns with various flood magnitudes required for specific types of ecosystem level processes. Most efforts to create a stable and pre-dictable flood regime have been anthropogenic in origin through engineered based interventions like dams, and retention and detention ponds, which are also some of the primary drivers behind the degradation of watershed health. A perfectly stable flow regime could, theoretically, be established by a highly integrated system of engineered solutions (albeit until they are either overwhelmed by a storm or undermined by system failure) within a very ecologically degraded watershed. Likewise, there could, theoret-ically, be examples of ecologically well connected and healthy watersheds with fairly volatile and unpredictable daily flow regimes. To overcome this, the authors need to discuss what exactly watershed health means and whether or not a predictable flow regime is the product of an ecosystem service. I could see an argument in which the shape of a storm specific hydrograph within a healthy watershed should be fairly pre-dictable, however, to study this would preclude the advantages proposed by this paper

(i.e. the application of the Fp algorithm in data sparse regions). I also agree that more human development and less natural systems generally leads to more flashy river flows as a result of decreased buffering capacity, however this study examines flow rates at daily intervals which washes out the ability to assess this linkage. Or maybe I'm missing something. The authors do point out later in the paper that Fp of zero (i.e. low predictability) would be the result of erratic rainfall (page 7 first paragraph). This is somewhat confusing because most of the introduction and discussion is focused on using Fp as a way to summarize "complex land use mosaics". Two paragraphs later the authors state "a decrease of Fp indicates watershed degradation." So how much of the decrease in Fp is explained by watershed degradation as opposed to just more erratic rainfall? I know the authors say that the GenRiver model is spatially explicit, but this is a little vague. Does this mean that spatial autocorrelation in precipitation is controlled for or that the model is spatially distributed? I understand that to have an Fp equal to 0 would require erratic rainfall, but the authors need to be consistent when describing what proxy measurements that Fp is suitable for. AR2.9 Thanks for these thoughts. We agree that stabilizing riverflow beyond what the "natural condition Fp" indicates is not without risk for the ecological functions of the river and its biota, and will add some comments to this effect. Quantitative estimates of Fp derived from a limited sampling period do depend on specifics of the actual rainfall. As rainfall is not generally known at the required spatial and temporal resolution to disentangle these relationships, we have to accept that Fp does not only respond to the land use mosaic, but also to rainfall. We'll check whether this can be stated more clearly upfront. R2.10. Figure 1 gets at the interconnection between many different elements that influence the hydrological cycle, and the authors break up the components into ecosystem structure, function, and human land use/perceived ecosystem service. However, I find the figure difficult to navigate and poorly described in the study. The different color arrows with different shades and outlines is one of main culprits of the confusion. The graphic needs to be simplified, it should probably start with rainfall, and terms like "plot-level" should either be defined or removed. AR2.10 Thanks, we will try to simplify the graphic R2.11. Point2. An Fp value ranging from 0-1 essentially represents the buffering capacity of the watershed, but there are also characteristics that influence how rapidly water reaches the stream. In this sense, Fp is only represents half of the picture. AR2.11 We don't quite understand the reviewer here: Fp quantifies a property of flow dynamics, and integrates over multiple aspects of the subsystems along the rainfall-vegetation-soil-streams pathway. The Fp concept is close to 'buffering' – but this is itself a scale-dependent concept (variability of outflow relative to variability of inflow, depending on time sacale and system boundaries). R2.12. The authors go on to create separate Fp's for each flow pathway, which is probably necessary for large catchments as each flow pathway likely do have large influences over space and time. However, this seems to be overcomplicating a model that was originally being created out of need for greater parsimony. If these pathway specific indices are necessary, then more discussion and justification is required in the text. AR2.12 We offer the weighted average of pathway-specific Fp values as an additional way of interpreting results, hoping that for some readers this will help understand what Fp is. It is not an essential component of the main argument, and we will state this more clearly. R2.13. The authors use vague language that needs some more clarification. Line 19 on page 7 contains "flow over a sufficiently long period". What is a sufficiently long period? Wouldn't a sufficiently long period wash out the "flashy" fluctuations that the authors are trying to explain with changing land cover/watershed degradation? If the 'sufficiently long period' is preventing what the study is attempting to explain, then I do not see how equation three could be derived. Maybe I'm missing something, but wouldn't the stochastic term represent all unexplained variations in the predicted river flow? Line 28 on page 7 explains that the stochastic term is equal to the sum of peak flows. Couldn't other unpredicted river flows have other anthropogenic origins that contribute to the river flow stochasticity (e.g. dam operations/failure, irrigation, urban water use, etc.)? The authors also mention new variables like Qadd and Fp,try without adequately discussing what they actually represent. AR2.13 We obviously created some confusion here and will try to more clearly separate the recursive

model of river flow (and its associated Fp) as a "principle", before getting into issues of data and empirical estimates of Fp in defined data sets.

R2.14. Point 3 Figures 2-9 were very readable and for the most part stand on their own, however the table were poorly formatted and vague. Table 1 does not provide land cover proportions by land cover type (other than forest). Percent developed land, existing flood control infrastructure, and population would all be helpful information. Not sure what 'dominant land cover' means. Do these watersheds have a history of damaging floods? AR2.14 Thanks, we will improve the presentation of Table 1 and define the terms used. R2.15. Why were the parameters in table 2 chosen? The authors do not provide an adequate discussion of how these parameter values were estimated. Why were the defaults in GenRiver used for each of the land cover types in table 3? What process or methodology did GenRiver use to estimate these values? What does 'relative drought threshold' mean? AR2.15 As also commented on by reviewer 1, the reference to the existing manual of the GenRiver model is clearly not sufficient here, and the model will have to be summarized in its key equations and assumptions, before we can use it to illustrate how Fp can be interpreted for alternative land cover scenarios.

R2.16. The use of the word 'some' in table 4 is simply too vague when describing the scenarios. The reader is left wondering what the magnitude of change would be within each of these scenarios. All of the information in tables 1-4 are critical components of the GenRiver model. The legitimacy and accuracy of the model is weak without proper documentation and justification of the underlying model parameters. The authors must correct this if we are to have any confidence in the results. AR2.16 We agree that further detail is needed here – however, the primary target here is "sensitivity analysis" of the way Fp will respond to changes within a plausible range, not to get into detail on any of the watersheds as such. R2.17. Table 5 and its corresponding discussion regarding the sample sizes required to reject a null hypothesis is interesting, but not enough information was given to make this section clear. The methodology is

clear enough, but the implications were not really discussed. The statement beginning on line 21 on page 14: "In practice, that means that empirical evidence that survives statistical tests will not emerge, even though effects on watershed health are real" is vague and needs some more clarifiĄcation. AR2.17 Thanks for the interest in these results – we will bring in some quantitative terms in these overall, qualitative conclusions R2.18. Lastly, in table 6, the authors provide broken links to the detailed reports of rainfall and river flow data. Moreover, there is very little discussion on the accuracy and metadata of each of these data sources, all of which have different origins. AR2.18 Unfortunately the website to which links are provided has been off-line for some time, we will double check all links function again in a resubmitted manuscript.

Please also note the supplement to this comment:
http://www.hydrol-earth-syst-sci-discuss.net/hess-2015-538/hess-2015-538-AC1-supplement.pdf

---

## Author Response (AR1)

HESS 2015-538

Dear editors, we are grateful for the opportunity to resubmit the manuscript in a version that address all the valuable suggestions made by reviewers. We have followed the advice to split the manuscript into two parts – which will, however, have to be read in conjunction; both are included here in a single file, as advised.

Our specific responses to all suggestions are listed in a table below

|  | Reviewer's comments | Authors' responses |
|---|---|---|
| Reviewer 1 | | |
| 1 | The paper presents an interesting attempt to develop a simple measure of flow persistence and address the issue of the ability to detect and attribute variations in daily riverflows to the effects of land-cover change in large catchments (riverbasins), a well-known issue in hydrology. | Thanks! We aim to derive an indicator of watershed health that mainly requires a series of data on rainfall and the temporal pattern of river flow at a point of interest. |
| 2 | The authors then use a model for estimating flow persistence to try to demonstrate how difficult it can be to identify the effects of land-use change in four tropical catchments. Their illustration of the sample sizes needed to identify effects is also interesting but difficult to interpret as too little information is given on the catchments used as examples | We have provided further details on the subject. |
| 3 | I am in full agreement that the issues of detectability and attribution are important for hydrologists to investigate because decision makers need evidence that catchment restoration can reduce flood risks and increase flow persistence by redirecting water from flow paths with rapid responses to rainfall to those with slow paths. This evidence can then be used to demonstrate that those benefits are being realised. | This is indeed the overall direction of the paper. |
| 4 | The authors also note that these effects are well documented in hillslope and small catchment studies but there is little evidence of such effects at in large catchments (river basins). | Agreed |
| 5 | I have no problem with their argument that some form of measure of hydrological change is important, but (a) their method of deriving the persistence indicator and (b) of applying their model to the four catchments is not adequately explained. | We have provided further details on the subject. See further explanation below. |
| 6 | I have divided my review into two sections based on the two main components of the paper: (a) flow persistence and (b) flow change detection and attribution. There are some typographic errors but I have not gone into these as the paper needs rework. | Thanks – we hope the current version addresses all concerns and suggestions and is clean |

| 7 | A) The one part of the paper addresses both flood risk and flow buffering by measuring aspects of the flow responsiveness of a catchment using a simple index (Fp) of the flow persistence. Flow persistence is defined as identical to the 'recession constant' (pg 7 line 7). However, I would argue that flow persistence is only half the picture, what is needed is actually a measure of flow responsiveness to rainfall because flooding can depend on how rapidly the flow increases versus factors that constrain that flow and cause water levels to rise. Flow responsiveness is also directly influenced by antecedent wetness, rainfall event depth and duration, and other factors which they do discuss but do not seem to incorporate in their approach. | We agree with flow persistence is focused on half the story (peak flows and subsequent decay), and that the speed at which peak flows are attained matters beyond the peak itself. However, since our aim is to derive watershed indicator using parsimonious information, we could not study the flow responsiveness to rainfall at timescales less than a day. If shorter-term dynamics matter, their study will require detail information on the specific space-time pattern of rainfall and flow measurements at least on hourly time-scale. Our study focuses on situation where daily records are the only information exists empirically. We explained and discussed further in our paper (see e.g. line 423-425). |
|---|---|---|
| 8 | Maybe I missed it, but I did not find a clear statement that Fp is only being calculated for the descending limb of a hydrograph. Yet this must be so because the range for Fp is constrained to the range 0-1. I would argue that to understand flood risk you also need to measure the ascending limb of the flows (the rapidity of the rise in response to rainfall patterns). I think would be probably be necessary to have indices for both the ascending and the descending limb as they are rarely symmetrical. | We agree that the Fp parameter (at daily scale) alone is not sufficient to predict flood dynamics at shorter timespans (e.g. hours), as details of spatial and temporal storm patterns interact with characteristics of the streambed, beyond what Fp captures. However, the finding that the proportion of fresh rainfall (minus soil water storage capacity linked to preceding Et) that comes down as riverflow is (1-Fp), allows us to infer an important component of flood predictions: the peak daily flow volume (given rainfall). We agree that a further empirical parameter at hourly (or similar timescale) can add further value – but for the empirical data sets we used only daily records are available. Indeed we don't assume that the ascending and descending limbs are symmetrical. |
| 9 | Attempting to parameterise Fp for the various flow pathways seems to me to add complexity but not much insight given their focus on large catchments where the relative importance of different flow paths can vary considerably across space and temporally. | This part is presented as an aid in the interpretation of the aggregate Fp, not as a way of empirically deriving it. We have tried to state this more clearly in the revised manuscript, section 2.3 (lines 288-312). |

| 10 | I did not find Figure 1 helpful as an illustration of the causal pathway or for placing the wide range of factors into their context. It has too much detail presented at the same level, is not adequately explained, and has too many terms that are not explained in the caption. A figure should be able to stand on its own with its caption, but in this case it does not, even with an extensive caption. A part of the problem is the use of two sets of arrows with different meanings in ways that are confusing. A more common way of representing this kind of diagram is a flow chart where the factors that influence (solid arrows) the (relative) magnitude of the water flows (hollow blue arrows) are represented in ways which make their role much clearer. I do not understand why rainfall is not made the start of the diagram and why the caption ends with number 0 rather than beginning there as one would intuitively expect. It is also not clear that land cover has various influences at both the "plot" level (whatever that may be) AND at the hillslope/landscape level. It is also not clear to me why there is a blue water flow arrow directly from rainfall to #2 and to #3 without passing through the landscape (and why #2 is in brackets). Why are some of the arrows broad and others not? Is the "triangle" to the left of human population density an arrow? Why are human population density and topography (subsidence) linked? What is the relevance of subsidence? Why is topography placed here and not within the sets #0 and #1 given its importance as a factor in the generation and flow of surface and subsurface runoff at both plot and hillslope levels? | We have rearranged the elements in Fig 1, and hope it now supports the understanding of the relations we try to highlight in the manuscript.

We have replaced "plot" by "patch".

Subsidence due to groundwater extraction in urban areas of high population density is a specific problem for a number of cities built on floodplains (such as Jakarta and Bangkok). It is a rather specific situation, but economically important flooding risks that were at first attributed to changes in upland land use are now understood to be generated in the urban areas. We have add a brief explanation and reference to this point of detail (line 154-157). |
|---|---|---|
| 11 | For a paper that attempts to explain how land cover changes affects catchment flow responses I find it inexplicable that there is almost no reference to: (a) the very extensive body of hillslope hydrology research into flow pathways and the temporal effects of different water partitioning and surface/subsurface on flow response to rainfall inputs; and (b) how hillslope responses might scale up to larger parts of catchments and large catchments. Even the brief mention of the different ways in which overland flow can be | Our primary purpose with the paper is not to explain how land cover changes affect hillslope hydrology, but rather to start at the other end and evaluate the information content of a metric that is derived form observations of river flow alone – we are indeed aware of the large body of work (and various models) at hillslope scale, and have added some further references in section 2.3, but the paper is not meant to be a review of all we know about 'floods', but to evaluate |

| | | |
|---|---|---|
| | generated (e.g. Hortonian versus variable (saturated) source areas) fails to cite the original research papers and the insights they provide in the catchment responsiveness. | a very parsimonious model, that can in a single index capture important first-order predictive value of the influence the watershed has, as modifier of stochastic rainfall. |
| 12 | Despite reading section 2 a few times I am still not entirely clear on the logic of the various deductions that are made about low flow, seasonality and the influence of varying Fp on the form of the hydrograph. Perhaps this is because the text is not always very clear. For example, the authors present the following (page 7 line 19 onwards): "If we consider the sum of river flow over a sufficiently long period, we can expect $\Sigma Qt$ to closely approximate $\Sigma Qt-1$, and thus $\Sigma Qt = Fp\ \Sigma Qt-1 + \Sigma\varepsilon$ (equation 2) From this relationship we obtain a first way of estimating the Fp value if a complete hydrograph is available: $Fp = 1 - \Sigma\varepsilon/\ \Sigma Qt$ (equation 3)" The only way I can derive equation 3 from 2 is to assume that $\Sigma Qt-1 = \Sigma Qt$ and so $\Sigma Qt$ can be substituted for $\Sigma Qt-1$. However, if this is so, then the only way equation 2 can hold is if $Fp = 1$ and $\Sigma\varepsilon = 0$. If this is so, how can this relationship then be used to estimate Fp? Or am I missing something here? In section 2.4 I assume that a model with a set input of daily rainfall and flow responses to that rainfall was used to create the ascending flow limbs so that Fp values could be used to generate the descending flow limb? And so that Fp could be varied? I also had similar difficulty in following parts of the methods section. For example, on pg 12 line 9 the term Qadd is abruptly introduced without an adequate explanation of its meaning. This is followed by the 'apparent Qadd' and Fp,try, again with no proper explanation. I should not have to go and find the paper cited (in fact a user manual) or to download the spreadsheet for an adequate explanation of the terms or to find a proper explanation of the FlowPer algorithm. | This is a crucial point in the derivation, we hope that a number of changes we made in the presentation make the text easier to follow.

We hope that replacing the notation $\varepsilon$ that may have, inadvertently, suggested that this is a negligible term by $Q_a$ the misunderstanding is avoided.

If we indeed assume $\Sigma_1^t\ Q_t = \Sigma_0^{t-1}\ Q_t$, we obtain $\Sigma Qt = Fp\ \Sigma Qt + \Sigma Q_a$ and hence $\Sigma Q_a = (1 - Fp)(\ \Sigma P - \Sigma E)$. The easiest way to obtain this relationship at the level of annual sums, is to have it hold true at event level: $Q_a = (1 - Fp)(\ P - Ex)$ with the caveat that the way $\Sigma E$ is partitioned over terms calculated at each day with rain ($\Sigma Ex$) may require some nuance (where not all antecedent ET is compensated in a single storm on soils that don't easily rewet, for example).

The key point, however, is that the stochastic $(P - Ex)$ term is to be multiplied with $(1 - Fp)$, which leads to direct influences on peak flows (if Fp does not vary with Q).

We have made the text more explicit in the way $Q_a$ is calculated for each Fp,try value: Every pair $(Qt, Qt+1)$ yields an estimate of Qadd derived from $Q_a = Q_{t+1} - F_{p,try}\ Qt$ Each $F_{p,try}$ value thus generates a frequency distribution of inferred $Q_a$ values (some of which will be negative for |

| | | relatively high Fp,try values), and the algorithm retains the value that minimizes Var(Qa). |
|---|---|---|
| 13 | The second part of the paper deals with the application of the GenRiver model for assessing the impacts of land cover on river flow and its attribution and detectability. The entire model and its application is only introduced in the methods but its structure and use should really be described already in the introduction. The model is said to be spatially explicit but it is not clear how that is realised in practice (i.e. it a distributed model?). | The use of the model to explore Fp derived from hydrographs for scenarios other than current land use indeed had to be more explicitly announced in the introduction – which we have done. However, it remains a 'tool' for exploring Fp properties of hydrographs, rather than being a focus on its own. By separating the manuscript into two parts, further explanation has been added on the the GenRiver model in part II. With a full specification of the model available for who wants to get into the detail, we have increased the description of key features that likely influence the results obtained. |
| 14 | Tables 1-4 do not provide an adequate description of the study catchments – what does dominant land cover mean? Although the authors note the importance of knowing what changes where in a catchment in relation to flow paths and times and attributing responses to changes and other factors, Table 1 does not give any indication even of how much of each land cover there is the baseline situation. Why not provide summaries or maps? Nor are we given information on where, in relation to this baseline state, the changes in land cover are made for the different scenarios. Why not provide a summary or a map? In Table 4 there is a repeated use of "some" in describing the changes made. This to me is not acceptable. | Thanks for the suggestion – we have add a map as Fig. 1 in Part II that visualizes the baseline situation, and a table of (plus figure as appendix 2 of part II) on what the scenario's mean in local context. A point of warning may be needed here, as the case study catchments have not been subject to a multi-year intensive measurement campaign of the type that reviewer probably finds necessary to fully trust results. We do provide the level. Of correspondence between Fp's derived from measured and modelled hydrographs, and on that basis present the further model results as illustrations of what can be expected, rather than as statements of fact. |
| 15 | We are not given an adequate explanation of how the single values of each of the 13 parameters of the GenRiver (Table 2) were obtained. Those parameter values are all ones that would vary a great deal spatially and with different land cover types (e.g. interception), but only a single value is given with no indication of their variability in the study catchments or how representative each value is. Providing definitions of the terms in a user manual the | We have provided full specification of the parameter values used for the calculations, while the model is freely downloadable, so we'll meet reasonable standards of repeatability – but of course further tests of the Fp summary statistic on hydrographs obtained with other models for other situations are likely needed before this method can be more widely accepted by the community. |

| | reader would have to look up is simply not acceptable. Table 3 also gives values for three important parameters for each of the land cover types with no explanation of what their sources and ranges are (BTW surely interception [Table 2] differs between forest and annual crops and so island cover specific?). Table3 also introduces the term relative drought threshold with no explanation of what it means and how the model uses it. The legitimacy, accuracy and representativeness of these values, together with the land cover changes, are critical to our confidence in the model outputs and thus in the analysis of the detectability and attribution of the changes in flows to changes in land cover. A study should be repeatable and this hypothetical modelling exercise certainly is not given the information included in the paper. In summary I am not entirely sure what to recommend overall. The idea of deriving a simple but robust measure of flow change (i.e. flow responsiveness) which can be causally related to land cover changes is sound, and necessary. | We hope that the discussion gives a fair assessment of the level of evidence, avoiding an oversell. |
|---|---|---|
| 16 | Flow persistence (Fp, recession) is an interesting measure and can be related to changes in the relative importance of different water flow paths, but it is also evident that it is not straightforward to derive and could be masked by the effects of location and catchment heterogeneity. I do think that a measure of the flow recession is not sufficient, the nature of the whole response to rainfall needs to be assessed for flood risk. The flow persistence component of the paper needs careful thought to make sure that the measure(s) are clearly and thoroughly explained. Even so, I still am left with the question of whether a simpler approach would not be to examine the slopes of the flow recession curves (in relation to rainfall event sizes and sequences [antecedent conditions]) for possible shifts due to land cover changes. Alternatively, using shifts in the flow duration curves as measures of changes in the relative importance of flow pathways, as has been done elsewhere, would be more effective and understandable. | It seems that reviewer presents a multi-parameter description as 'simpler' than our single metric? Again, our focus is to assess the information content of a metric of flow predictability that appears to align well with the way downstream observers describe and experience changes in the conditions in upper watersheds. If our primary focus would be to assess the effects of land use change on flood risks as such, for any of the catchments studied or for the wider geographic domain in its totality, we would probably embark on further data collection… A key point here is that in many situations (historical) rainfall data are inadequate – at best a few point records are available, but no spatially weighted average, and there are many degrees of freedom in getting reasonable answers for wrong reasons in multiparameter models. |

| | | |
|---|---|---|
| 17 | Another alternative would be to use the relationship between rainfall event sizes and sequences (e.g. antecedent wetness) and flow response to those events and sequences to infer changes. The modelling component needs a lot more information to back up the chosen parameter values for both the hydrological (Table 2) and land cover-specific values, as well as specific information on the extent of the land cover changes and their spatial configurations. It also needs to provide information on how well the outputs it generates for the different land cover types compare with the findings of other studies (i.e. how well does the model perform). Overall I need more information on the model structure and setup to interpret how well it performs in this application. This would require expanding the paper substantially. | We have added information on the model in the methods and appendix and provided the main parameters with its values in Table 2 and 3. |
| 18 | Overall, my conclusion is that perhaps this paper attempts to cover too much ground and should be two papers: - One on the issue of catchment responsiveness to rainfall as a measure of land cover change, including flow persistence - One on modelling of the effects of land cover change on river flow responses and the difficulties of detecting and attributing changes in flow responsiveness to changes in land cover (and relating this back to the changes in the relative importance of flow paths linked to the changes in land cover). | As suggested we have split the paper into two: 1) on flow persistence concept as indicator of watershed health and 2) on the application of flow persistence (Fp) concept on 4 watersheds.  We believe summarizing hydrographs resulting from different land use scenarios into Fp can play a role as 'metric of watershed health'. |
| Reviewer 2 | | |
| 1 | Summary Overall, I think that this study addresses a critical knowledge gap with important implications. However, the conceptual foundation regarding watershed health and flow predictability requires a closer examination. The derivation of Fp and the process used to create the GenRiver model parameters needs more discussion, clarification, and justification if the reader is to have any confidence in the results. I think that if the authors were to accomplish these revisions then the paper would simply be | Thanks for the interest, we have followed the suggestion to split the manuscript into two parts "I. Theory, II. Applications", and appreciate the detailed suggestions and comments. |

| | | |
|---|---|---|
| | too long and cover too much ground. Breaking the research into two separate papers is probably a better course with one focusing more on the conceptualization and creation of the Fp term and one on the application of it within the GenRiver framework. | |
| 2 | As I understand the article, the authors are attempting to develop a single measure of watershed health called 'flow persistence' (Fp). This Fp parameter measures the volatility of daily river flow in response to land cover change within large catchments. | We have tried to make the manuscript more clear in the steps involved: Fp is a measure of 'volatility' (or its complement), that can be used to quantify one aspect of the impacts, at multiple scales, of land cover change. |
| 3 | One of the key objectives of the study is to determine the value of specific land cover types in terms of flood mitigation. The study itself is broken down into two phases: (1) the derivation of a river flow algorithm, and (2) the application of the algorithm within four watersheds with different rainfall and land cover characteristics. | In the current manuscript the Fp definition in terms of a recursive river flow model is presented as step 0, before the empirical steps on how actual (or simulated) flow data can be used to derive an estimate of Fp (step 1), illustrated with applications in four case studies (step 2). |
| 4 | The key points that need to be addressed include: 1. A better justification that flow predictability does in fact correspond with watershed health; 2. A better explanation of the flow persistence derivation; and 3. A much more thorough explanation of the Fp algorithm application within the four catchments. | We thank reviewer for the positive suggestions and have I tried to clarify these points in the resubmission. "does in fact correspond with watershed health", is however a complex question, as "watershed health" hasn't been satisfactorily quantified in absolute terms anywhere. What we claim is that changes in Fp, towards lower values match "degradation" and towards higher values match "restoration", from a downstream perspective. The metric matches a common way of describing degradation as loss of predictability. In unpacking the concept, however, we find that the specific pattern of rainfall in a given year interacts with the condition of the watershed in generating a river flow pattern, as captured in Fp. Our conclusion that multiple years of "paired catchment" type data (different |

| | | watershed conditions, same rainfall pattern) are needed to reject null-hypotheses that land cover use effects are neutral. So – we qualify our claim that Fp is an "indicator", not a "measure" of watershed health. But, so far we don't know of a better simple indicator. |
|---|---|---|
| 5 | The study addresses a significant point of contention in the literature: the influence of land cover (particularly forests) and flooding at the watershed scale. If the Fp model is properly justified and performs adequately, then it would undoubtedly increase our understanding of the linkages between land cover and flood risk. The benefits of such an approach are clear as it would make for a much more parsimonious model of river flow that would greatly enhance the monitoring and prioritization of specific landscape management decisions. | We thank reviewer for the interest – indeed our target is a key parameter for a parsimonious model that can communicate key functional property of the way a watershed functions, given variable rainfall. |
| 6 | That said, the paper requires substantial work to adequately address the points above and may need to be split into two separate papers. I will address the three points I mentioned above in greater detail below. There are quite a few of typographical and grammatical errors in the paper, but I will leave these alone for now as the paper requires substantial work. | As reviewer 1 also suggested splitting the paper along similar lines, we have followed this suggestion. |
| 7 | Point 1 In the paper the authors use persistence, predictability, and watershed health interchangeably. | We hope the current text is clear in presenting "persistence" as a first descriptor of what Fp measures (the part of todays flow that can be counted on for tomorrow, regardless of additional precipitation); 1-Fp controls the part of new rainfall that adds to the stream, and as such controls the predictability of overall flow; the way that Fp is an indicator of watershed health is at a further level of interpretation. |
| 8 | One of the key assumptions of the paper and previous watershed rehabilitation efforts is that increasing the presence of natural land covers (particularly forests and wetlands) will restore the natural flood regime with lower peak flows and less damaging flood events. | Actually we took this to be a testable hypothesis rather than a priori assumption. |

| 9 | The authors do a good job of documenting previous studies that have illustrated the complexity of the linkage between reforestation and river flows. Moreover, the ability of wetlands and riparian forests to absorb rainfall, slow streamflow, and attenuate peakflows is supported by many studies and is fairly well understood. However, these types of stream corridor ecosystems also require a particular type of disturbance regime that creates opportunities for species specific recruitment processes and establishes landscape and topographic heterogeneity that are critical components of watershed health. These disturbance regimes are often characterized by variable flow patterns with various flood magnitudes required for specific types of ecosystem level processes. Most efforts to create a stable and predictable flood regime have been anthropogenic in origin through engineered based interventions like dams, and retention and detention ponds, which are also some of the primary drivers behind the degradation of watershed health. A perfectly stable flow regime could, theoretically, be established by a highly integrated system of engineered solutions (albeit until they are either overwhelmed by a storm or undermined by system failure) within a very ecologically degraded watershed. Likewise, there could, theoretically, be examples of ecologically well connected and healthy watersheds with fairly volatile and unpredictable daily flow regimes. To overcome this, the authors need to discuss what exactly watershed health means and whether or not a predictable flow regime is the product of an ecosystem service. I could see an argument in which the shape of a storm specific hydrograph within a healthy watershed should be fairly predictable, however, to study this would preclude the advantages proposed by this paper (i.e. the application of the Fp algorithm in data sparse regions). I also agree that more human development and less natural systems generally leads to more flashy river flows as a result of decreased buffering capacity, however this study examines flow rates at daily intervals which washes out the ability to assess this linkage. Or maybe I'm | Thanks for these thoughts. We agree that stabilizing riverflow beyond what the "natural condition Fp" indicates is not without risk for the ecological functions of the river and its biota, and have added some comments to this effect (line 466-471).

Quantitative estimates of Fp derived from a limited sampling period do depend on specifics of the actual rainfall. As rainfall is not generally known at the required spatial and temporal resolution to disentangle these relationships, we have to accept that Fp does not only respond to the land use mosaic, but also to rainfall. We have stated this more clearly upfront. |

| | | |
|---|---|---|
| | missing something. The authors do point out later in the paper that Fp of zero (i.e. low predictability) would be the result of erratic rainfall (page 7 first paragraph). This is somewhat confusing because most of the introduction and discussion is focused on using Fp as a way to summarize "complex land use mosaics". Two paragraphs later the authors state "a decrease of Fp indicates watershed degradation." So how much of the decrease in Fp is explained by watershed degradation as opposed to just more erratic rainfall? I know the authors say that the GenRiver model is spatially explicit, but this is a little vague. Does this mean that spatial autocorrelation in precipitation is controlled for or that the model is spatially distributed? I understand that to have an Fp equal to 0 would require erratic rainfall, but the authors need to be consistent when describing what proxy measurements that Fp is suitable for. | |
| 10 | Figure 1 gets at the interconnection between many different elements that influence the hydrological cycle, and the authors break up the components into ecosystem structure, function, and human land use/perceived ecosystem service. However, I find the figure difficult to navigate and poorly described in the study. The different color arrows with different shades and outlines is one of main culprits of the confusion. The graphic needs to be simplified, it should probably start with rainfall, and terms like "plot-level" should either be defined or removed. | Thanks, we have tried to simplify the graphic |
| 11 | Point2. An Fp value ranging from 0-1 essentially represents the buffering capacity of the watershed, but there are also characteristics that influence how rapidly water reaches the stream. In this sense, Fp is only represents half of the picture. | We don't quite understand the reviewer here: Fp quantifies a property of flow dynamics, and integrates over multiple aspects of the subsystems along the rainfall-vegetation-soil-streams pathway. The Fp concept is close to 'buffering' – but this is itself a scale-dependent concept (variability of outflow relative to variability of inflow, depending on time scale and system boundaries). |

| 12 | The authors go on to create separate Fp's for each flow pathway, which is probably necessary for large catchments as each flow pathway likely do have large influences over space and time. However, this seems to be overcomplicating a model that was originally being created out of need for greater parsimony. If these pathway specific indices are necessary, then more discussion and justification is required in the text. | We offer the weighted average of pathway-specific Fp values as an additional way of interpreting results, hoping that for some readers this will help understand what Fp is. It is not an essential component of the main argument, and we will state this more clearly. |
|----|----|----|
| 13 | R2.13. The authors use vague language that needs some more clarification. Line 19 on page 7 contains "flow over a sufficiently long period". What is a sufficiently long period? Wouldn't a sufficiently long period wash out the "flashy" fluctuations that the authors are trying to explain with changing land cover/watershed degradation? If the 'sufficiently long period' is preventing what the study is attempting to explain, then I do not see how equation three could be derived. Maybe I'm missing something, but wouldn't the stochastic term represent all unexplained variations in the predicted river flow? Line 28 on page 7 explains that the stochastic term is equal to the sum of peak flows. Couldn't other unpredicted river flows have other anthropogenic origins that contribute to the river flow stochasticity (e.g. dam operations/failure, irrigation, urban water use, etc.)? The authors also mention new variables like Qadd and Fp,try without adequately discussing what they actually represent. | We obviously created some confusion here and have tried to more clearly separate the recursive model of river flow (and its associated Fp) as a "principle", before getting into issues of data and empirical estimates of Fp in defined data sets. |
| 14 | Point 3 Figures 2-9 were very readable and for the most part stand on their own, however the table were poorly formatted and vague. Table 1 does not provide land cover proportions by land cover type (other than forest). Percent developed land, existing flood control infrastructure, and population would all be helpful information. Not sure what 'dominant land cover' means. Do these watersheds have a history of damaging floods? | Thanks, we have improved the presentation of Table 1 in part II and defined the terms used, with references to further studies. |
| 15 | Why were the parameters in table 2 chosen? The authors do not provide an adequate discussion of how these parameter values | As also commented on by reviewer 1, the reference to the existing manual of the GenRiver model was clearly not |

| | were estimated. Why were the defaults in GenRiver used for each of the land cover types in table 3? What process or methodology did GenRiver use to estimate these values? What does 'relative drought threshold' mean? | sufficient here, and we have now summarized key equations and assumptions of the model, before using it to illustrate how Fp can be interpreted for alternative land cover scenarios. |
|---|---|---|
| 16 | The use of the word 'some' in table 4 is simply too vague when describing the scenarios. The reader is left wondering what the magnitude of change would be within each of these scenarios. All of the information in tables 1-4 are critical components of the GenRiver model. The legitimacy and accuracy of the model is weak without proper documentation and justification of the underlying model parameters. The authors must correct this if we are to have any confidence in the results. | In Appendix 2 we have added information on the associated land use distributions of the different land use scenario described in Table 4. |
| 17 | Table 5 and its corresponding discussion regarding the sample sizes required to reject a null hypothesis is interesting, but not enough information was given to make this section clear. The methodology is clear enough, but the implications were not really discussed. The statement beginning on line 21 on page 14: "In practice, that means that empirical evidence that survives statistical tests will not emerge, even though effects on watershed health are real" is vague and needs some more clarification. | Thanks for the interest in these results – we have brought in some quantitative terms in these overall, qualitative conclusions |
| 18 | Lastly, in table 6, the authors provide broken links to the detailed reports of rainfall and river flow data. Moreover, there is very little discussion on the accuracy and metadata of each of these data sources, all of which have different origins. | Unfortunately the website to which links are provided has been off-line for some time, we have double checked all links function again in a resubmitted manuscript. |

---

## Referee Report (RR1)

Referee Report

**General Comments:**

Both of these articles are relevant and could provide contributions to decision makers, as well as those working with ecosystem services and flood risk. The authors adequately addressed all of my concerns within the original submission making the subsequent paper much more suitable for publication. The separation of the original article into two greatly strengthens the study.

**Specific Comments:**

**Abstract:** Abstracts need to be shortened. The abstracts are far too verbose as they stand. A more simplified abstract that excludes detailed discussions of equations/parameters will make them clearer and more engaging.

**PART I**

**Figure 1:** This figure is much improved and far easier to follow, however it seems a little blurry. Misspelling of pathways at line 727. No explanation of step 10 in figure description.

**Figure 4:** This figure is blurry making the small text difficult to read.

**Avoid using contractions:** For example, "don't" should be "do not"

**Lined 158:** change doess to does.

**Line 196:** "it's" should be "its"

**Line 212:** "The probably simplest" to "A simple"

**Line 227-239**: The wording of this sentence is confusing and requires some revisions.

**Line 456:** "en" should be "an"

**Line 534-536:** Spell out authors' full names.

**PART II**

An explanation of the general land cover characteristics for each watershed would be helpful. Table 1 provides an element of confusion regarding the proportion of forested land that needs to be acknowledged.

Needs a conclusion section that summarizes the study, discusses implications, and acknowledges limitations and future research directions.

**Line 817:** add comma after part

**Line 832:** removed comma after intensity and response

**Line 841:** add "the" after "we consider"

**Line 848:** change patchlevel to "the patch level"

**Line 855 & 883:** change "land-cover" to land "cover" (and throughout paper)

**Line 868:** change "Fig." to "Figure" (and throughout paper); "provides" to "provide"; and remove "are"

**Line 886:** change "land-use" to "land use" (and throughout paper)

**Line 937:** unsure what dace is supposed to mean

**Line 945:** add "the" before "measuring"

**Line 987:** no supplementary information given…

**Table 1:** What is the differentiation between "forest" within land cover type and "natural forest" at the bottom of the table? For Bialo and Mae Chaem they are equal, but are different for the other two. An explanation for this must be given as the proportion of forested land is one of the primary drivers behind flow predictability.

**Figure 1:** blurry

**Figure 5:** Why are the water balance percentages different for the NatFor scenario when Figure App2 shows that the NatFor scenario is 100% for all watersheds?

**Appendix 2:** no proportions for Mae Chaem AgFor are given.

---

## Author Response (AR2)

HESS-2015-538 Response to reviewers

**General:** We appreciate the chance to respond to the two reviewers, as the original manuscript underwent major changes in splitting it in two parts. We're pleased that the current focus of both papers appears to work well. The further comments of reviewer 2 are of definite help in fine-tuning the manuscripts. Reviewer 1 still has major doubts or questions on the core of the method and concept we describe here – and we from our side are challenged to understand where and how we create the apparent misunderstandings that the reviewer articulates.

The core of the argument here seems to be:

The manuscript states:  $Q_t = F_p Q_{t-1} + Q_{a,t}$

Reviewer states: Assuming that  $Q_{a,t} = 0$  for now, then  $Q_t = F_p Q_{t-1}$  and this can't be true for  $F_p$  restricted to the 0-1 range, so the equation can't be right...

But, that's why there is the term  $Q_{a,t}$ . We've tried to understand whether in the text leading up to the first equation we've given the impression that  $Q_{a,t}$  is 'negligible', but we don't see where we did set the reviewer on the wrong track.

Yet we have revised some of the text introducing the concept, and hope that a fresh look at this all by the reviewer could lead to more understanding of what we propose.

Detailed response to reviewer comments

| Reviewer 1.                                      | Response                                           |
|--------------------------------------------------|----------------------------------------------------|
| I think the restructuring and responses to the   | Thanks                                             |
| reviewer comments on the previous submission     |                                                    |
| have materially improved the MS relative to the  |                                                    |
| previous one.                                    |                                                    |
| I have only reviewed the first paper of the two  |                                                    |
| though because I have encountered issues that    |                                                    |
| need to be resolved.                             |                                                    |
| My overall comment is that much of the           | There have of course been many discussions of      |
| lengthy introduction on the various possible     | the type reviewer prefers, and some are quoted     |
| interpretations of flow persistence (Fp or the   | here. Yet, it is not clear how the success of such |
| slope of the recession curve) adds little value  | interventions can be measured. Out focus here      |
| and could even confuse readers. My preference    | is what a stakeholder/observer who 'only' has      |
| would be reduce the lengthy detail and           | access to data on the daily dynamics of river      |
| digressions, and focus on why and how            | flow can infer about conditions upstream, and      |
| catchments respond temporally to rainfall,       | how he/she could interpret changes in the          |
| what shifts in those responses may mean, and     | performance parameter that we propose.             |
| how an understanding of response mechanisms      | That's the stated purpose of the paper, and        |
| can lead to actions aimed at recovering          | that's what we do. If reviewer wants to see a      |
| catchment function.                              | different paper, she/he may need different         |
|                                                  | authors with access to different data.             |
| It seems that the authors have failed to grasp a | We have indeed failed to grasp this argument,      |
| major comment I had on the previous version.     | because we think it is based on reviewer not       |
| In my opinion this MUST be addressed before      | grasping the argument we made. We hope             |
| this version can be taken any further. I thought | reviewer can reconsider the perspective.           |
| my comment was straightforward and easily        |                                                    |
| understood, but it seems the authors have        |                                                    |

| failed to grasp the issue so I will try again. The
whole study is predicated on finding a simple
index of flow persistence which can be
measured over time to detect whether land-use
changes are altering the responsiveness of a
catchment to rainfall input. Fine and good. An | We concur that the rise in hydrographs is faster                                            |
|--------------------------------------------------------------------------------------------------------------------------------------------------------------------------------------------------------------------------------------------------------------------------------------------------|---------------------------------------------------------------------------------------------|
| the flows out of a catchment initially rise after a
rainfall event and then decrease again, with the                                                                                                                                                                                          | than the subsequent decline – and found
(based on water balance logic) a way that the    |
| decrease following what is known as a                                                                                                                                                                                                                                                            | two are linked: the increase (at daily                                                      |
| recession curve. The rise is typically more rapid                                                                                                                                                                                                                                                | observation scale) is $(1-F_p)$ times the effective                                         |
| context, I now take their equation (1) which is                                                                                                                                                                                                                                                  | rainfall, while the decrease is proportional to $F_p$                                       |
| Ot = En Ot - 1 + Oa t                                                                                                                                                                                                                                                                            |                                                                                             |
| Assuming that $Qa,t = 0$ for now, then $Qt = FpQt$ -                                                                                                                                                                                                                                             | What gives reviewer reason to make this                                                     |
| 1 (i.e. the flow at time t is related to the flow at                                                                                                                                                                                                                                             | assumption?                                                                                 |
| time t-1 by Fp). For this equation to be true for                                                                                                                                                                                                                                                | Discos understand that if 0 > 0, the tarm 0                                                 |
| 1) En must be >1                                                                                                                                                                                                                                                                                 | Please, understand that if $Q_t > Q_{t-1}$ the term $Q_{a,t}$                               |
| Yet they only deal with values of Ep in the range                                                                                                                                                                                                                                                | Please the explanation is that there is also                                                |
| from 0 to 1. So the Fp they are describing must                                                                                                                                                                                                                                                  | term Qa,t in the equation                                                                   |
| only calculated on the falling flows. Yet this is                                                                                                                                                                                                                                                |                                                                                             |
| not mentioned or described anywhere in the                                                                                                                                                                                                                                                       | The caption of Fig. 2 refers to "unimodal rainfall                                          |
| paper even though they explicitly note that Fp                                                                                                                                                                                                                                                   | regime – we have provided further detail on                                                 |
| Is equivalent to a recession constant (line 222).                                                                                                                                                                                                                                                | now a stochastic rainfall time series is used                                               |
| The authors provide data that show how                                                                                                                                                                                                                                                           | here to derive $(P_t-E_{tx})$ , while increments in now                                     |
| tesponses in a catchinent (Figure 2 and assoc                                                                                                                                                                                                                                                    | (the $Q_{a,t}$ term) are calculated as (1- $F_p$ ) ( $P_t$ - $E_{tx}$ ).                    |
| magnitude of both the rices and falls) increases                                                                                                                                                                                                                                                 | $h_p = 1.0$ we have a constant now throughout the year, without any increments or decreases |
| vet there are no values of En>1. This needs                                                                                                                                                                                                                                                      | and 'nerfect' huffering. Values of E. above 1.0                                             |
| explanation                                                                                                                                                                                                                                                                                      | and perfect burnering. Values of $r_p$ above 1.0                                            |
|                                                                                                                                                                                                                                                                                                  | equations we developed                                                                      |
| At some points they discuss the term Oa t as                                                                                                                                                                                                                                                     | Yes both statements are correct                                                             |
| though it were stochastic, at other points it                                                                                                                                                                                                                                                    |                                                                                             |
| seems that Oa.t is used to account for all flows                                                                                                                                                                                                                                                 |                                                                                             |
| greater than some level of base flow (implying                                                                                                                                                                                                                                                   |                                                                                             |
| that Qa,t is ≥0).                                                                                                                                                                                                                                                                                |                                                                                             |
| (BTW My understanding is that if Qa,t
represents a proportion of the observed flow, it                                                                                                                                                                                                        | Our use of "stochastic" is aligned with its
common definition:                           |
| is not actually stochastic although there may be                                                                                                                                                                                                                                                 | https://en.wikipedia.org/wiki/Stochastic as                                                 |
| factors that give it a degree of stochasticity?)                                                                                                                                                                                                                                                 | "the physical systems in which the values of                                                |
|                                                                                                                                                                                                                                                                                                  | parameters, measurements, expected input,                                                   |
|                                                                                                                                                                                                                                                                                                  | and disturbances are uncertain. "; it doesn't                                               |
|                                                                                                                                                                                                                                                                                                  | mean that stochastic terms are unbounded.                                                   |
|                                                                                                                                                                                                                                                                                                  | The idea that a term can't be stochastic                                                    |
|                                                                                                                                                                                                                                                                                                  | because it can be expressed as a fraction of the                                            |
|                                                                                                                                                                                                                                                                                                  | sum of that term and another one would, we                                                  |
|                                                                                                                                                                                                                                                                                                  | think, not hold up to scrutiny. In that case                                                |
|                                                                                                                                                                                                                                                                                                  | stochasticity could not exist, as it can always be expressed as a fraction.                 |

| If Qa,t represents all non-base-flows, then all                                                                                                                                                                                                                                                                                                                                                                                                                                                                                                                                                                                                                                                                                                                                                                                                                                                                                                                                                                                        | Almost correct, Qa,t indeed represents all non-
base-flows – but there are no cases with $E > 1$                                                                                                                                                                                                                                                                                                                                                                                                                                                                                                                                                                                                                                                          |
|----------------------------------------------------------------------------------------------------------------------------------------------------------------------------------------------------------------------------------------------------------------------------------------------------------------------------------------------------------------------------------------------------------------------------------------------------------------------------------------------------------------------------------------------------------------------------------------------------------------------------------------------------------------------------------------------------------------------------------------------------------------------------------------------------------------------------------------------------------------------------------------------------------------------------------------------------------------------------------------------------------------------------------------|--------------------------------------------------------------------------------------------------------------------------------------------------------------------------------------------------------------------------------------------------------------------------------------------------------------------------------------------------------------------------------------------------------------------------------------------------------------------------------------------------------------------------------------------------------------------------------------------------------------------------------------------------------------------------------------------------------------------------------------------------------------|
| estimate $\Omega_2$ t (which is what Linfer from lines                                                                                                                                                                                                                                                                                                                                                                                                                                                                                                                                                                                                                                                                                                                                                                                                                                                                                                                                                                                 | the way we have defined the terms of our                                                                                                                                                                                                                                                                                                                                                                                                                                                                                                                                                                                                                                                                                                                     |
| 243-246)?                                                                                                                                                                                                                                                                                                                                                                                                                                                                                                                                                                                                                                                                                                                                                                                                                                                                                                                                                                                                                              | equations.                                                                                                                                                                                                                                                                                                                                                                                                                                                                                                                                                                                                                                                                                                                                                   |
| Did they effectively vary Qa,t to get the results                                                                                                                                                                                                                                                                                                                                                                                                                                                                                                                                                                                                                                                                                                                                                                                                                                                                                                                                                                                      | Reviewer probably refers to Fig. 2 here. In that                                                                                                                                                                                                                                                                                                                                                                                                                                                                                                                                                                                                                                                                                                             |
| they present in Figure 1? If so, this needs to be                                                                                                                                                                                                                                                                                                                                                                                                                                                                                                                                                                                                                                                                                                                                                                                                                                                                                                                                                                                      | case: we have modified the figure to show both                                                                                                                                                                                                                                                                                                                                                                                                                                                                                                                                                                                                                                                                                                               |
| explained.                                                                                                                                                                                                                                                                                                                                                                                                                                                                                                                                                                                                                                                                                                                                                                                                                                                                                                                                                                                                                             | base flow and Qa,t                                                                                                                                                                                                                                                                                                                                                                                                                                                                                                                                                                                                                                                                                                                                           |
| I also question then why they do not simply                                                                                                                                                                                                                                                                                                                                                                                                                                                                                                                                                                                                                                                                                                                                                                                                                                                                                                                                                                                            | Yes, the $F_p$ is close to one of several flow                                                                                                                                                                                                                                                                                                                                                                                                                                                                                                                                                                                                                                                                                                               |
| describe this approach as flow separation                                                                                                                                                                                                                                                                                                                                                                                                                                                                                                                                                                                                                                                                                                                                                                                                                                                                                                                                                                                              | separation techniques – but that terminology                                                                                                                                                                                                                                                                                                                                                                                                                                                                                                                                                                                                                                                                                                                 |
| technique with the aim of quantifying the                                                                                                                                                                                                                                                                                                                                                                                                                                                                                                                                                                                                                                                                                                                                                                                                                                                                                                                                                                                              | might come with strong perceptions on how it                                                                                                                                                                                                                                                                                                                                                                                                                                                                                                                                                                                                                                                                                                                 |
| catchment responsiveness rather than the flow                                                                                                                                                                                                                                                                                                                                                                                                                                                                                                                                                                                                                                                                                                                                                                                                                                                                                                                                                                                          | should be done. We prefer to present our                                                                                                                                                                                                                                                                                                                                                                                                                                                                                                                                                                                                                                                                                                                     |
| persistence.                                                                                                                                                                                                                                                                                                                                                                                                                                                                                                                                                                                                                                                                                                                                                                                                                                                                                                                                                                                                                           | from a time series of daily flow reserves and                                                                                                                                                                                                                                                                                                                                                                                                                                                                                                                                                                                                                                                                                                                |
|                                                                                                                                                                                                                                                                                                                                                                                                                                                                                                                                                                                                                                                                                                                                                                                                                                                                                                                                                                                                                                        | then discuss where and how this differs from                                                                                                                                                                                                                                                                                                                                                                                                                                                                                                                                                                                                                                                                                                                 |
|                                                                                                                                                                                                                                                                                                                                                                                                                                                                                                                                                                                                                                                                                                                                                                                                                                                                                                                                                                                                                                        | what has been done before. In the hone that it                                                                                                                                                                                                                                                                                                                                                                                                                                                                                                                                                                                                                                                                                                               |
|                                                                                                                                                                                                                                                                                                                                                                                                                                                                                                                                                                                                                                                                                                                                                                                                                                                                                                                                                                                                                                        | may help readers like reviewer 1, we have                                                                                                                                                                                                                                                                                                                                                                                                                                                                                                                                                                                                                                                                                                                    |
|                                                                                                                                                                                                                                                                                                                                                                                                                                                                                                                                                                                                                                                                                                                                                                                                                                                                                                                                                                                                                                        | if it would help the reviewer we have used the                                                                                                                                                                                                                                                                                                                                                                                                                                                                                                                                                                                                                                                                                                               |
|                                                                                                                                                                                                                                                                                                                                                                                                                                                                                                                                                                                                                                                                                                                                                                                                                                                                                                                                                                                                                                        | flow separation language at an earlier point in                                                                                                                                                                                                                                                                                                                                                                                                                                                                                                                                                                                                                                                                                                              |
|                                                                                                                                                                                                                                                                                                                                                                                                                                                                                                                                                                                                                                                                                                                                                                                                                                                                                                                                                                                                                                        | the revised manuscript.                                                                                                                                                                                                                                                                                                                                                                                                                                                                                                                                                                                                                                                                                                                                      |
| To my mind, they have not adequately                                                                                                                                                                                                                                                                                                                                                                                                                                                                                                                                                                                                                                                                                                                                                                                                                                                                                                                                                                                                   | We appreciate your 'persistence', but hope that                                                                                                                                                                                                                                                                                                                                                                                                                                                                                                                                                                                                                                                                                                              |
| explained these key points, so until I get an                                                                                                                                                                                                                                                                                                                                                                                                                                                                                                                                                                                                                                                                                                                                                                                                                                                                                                                                                                                          | the current version avoids the misunderstan-                                                                                                                                                                                                                                                                                                                                                                                                                                                                                                                                                                                                                                                                                                                 |
| understandable explanation I cannot accept the                                                                                                                                                                                                                                                                                                                                                                                                                                                                                                                                                                                                                                                                                                                                                                                                                                                                                                                                                                                         | ding on which, we believe, your issues were                                                                                                                                                                                                                                                                                                                                                                                                                                                                                                                                                                                                                                                                                                                  |
| paper.                                                                                                                                                                                                                                                                                                                                                                                                                                                                                                                                                                                                                                                                                                                                                                                                                                                                                                                                                                                                                                 | based.                                                                                                                                                                                                                                                                                                                                                                                                                                                                                                                                                                                                                                                                                                                                                       |
| Some other points:                                                                                                                                                                                                                                                                                                                                                                                                                                                                                                                                                                                                                                                                                                                                                                                                                                                                                                                                                                                                                     |                                                                                                                                                                                                                                                                                                                                                                                                                                                                                                                                                                                                                                                                                                                                                              |
| Lines 119-138 provides a discussion of whether                                                                                                                                                                                                                                                                                                                                                                                                                                                                                                                                                                                                                                                                                                                                                                                                                                                                                                                                                                                         | We added a brief reference to issues of                                                                                                                                                                                                                                                                                                                                                                                                                                                                                                                                                                                                                                                                                                                      |
| changes in land cover lead to changes in flows.                                                                                                                                                                                                                                                                                                                                                                                                                                                                                                                                                                                                                                                                                                                                                                                                                                                                                                                                                                                        | detectability: "Detectability of effects depends                                                                                                                                                                                                                                                                                                                                                                                                                                                                                                                                                                                                                                                                                                             |
|                                                                                                                                                                                                                                                                                                                                                                                                                                                                                                                                                                                                                                                                                                                                                                                                                                                                                                                                                                                                                                        |                                                                                                                                                                                                                                                                                                                                                                                                                                                                                                                                                                                                                                                                                                                                                              |
| They note that this has been shown in small                                                                                                                                                                                                                                                                                                                                                                                                                                                                                                                                                                                                                                                                                                                                                                                                                                                                                                                                                                                            | on their relative size, the accuracy of the                                                                                                                                                                                                                                                                                                                                                                                                                                                                                                                                                                                                                                                                                                                  |
| They note that this has been shown in small catchments but not in large catchments. Yet                                                                                                                                                                                                                                                                                                                                                                                                                                                                                                                                                                                                                                                                                                                                                                                                                                                                                                                                                | on their relative size, the accuracy of the measurement devices, background variability of                                                                                                                                                                                                                                                                                                                                                                                                                                                                                                                                                                                                                                                                   |
| They note that this has been shown in small
catchments but not in large catchments. Yet
they do not discuss the simple issue of                                                                                                                                                                                                                                                                                                                                                                                                                                                                                                                                                                                                                                                                                                                                                                                                                                                                                                  | on their relative size, the accuracy of the
measurement devices, background variability of
the signal and length of observation period."                                                                                                                                                                                                                                                                                                                                                                                                                                                                                                                                                                                                               |
| They note that this has been shown in small
catchments but not in large catchments. Yet
they do not discuss the simple issue of
detectability given the accuracy of the flow
recording cystem. The design of most large                                                                                                                                                                                                                                                                                                                                                                                                                                                                                                                                                                                                                                                                                                                                                                                                    | on their relative size, the accuracy of the
measurement devices, background variability of
the signal and length of observation period."                                                                                                                                                                                                                                                                                                                                                                                                                                                                                                                                                                                                               |
| They note that this has been shown in small
catchments but not in large catchments. Yet
they do not discuss the simple issue of
detectability given the accuracy of the flow
recording system. The design of most large
weirs is simply not suitable for accurately.                                                                                                                                                                                                                                                                                                                                                                                                                                                                                                                                                                                                                                                                                                                                                    | on their relative size, the accuracy of the
measurement devices, background variability of
the signal and length of observation period."                                                                                                                                                                                                                                                                                                                                                                                                                                                                                                                                                                                                               |
| They note that this has been shown in small
catchments but not in large catchments. Yet
they do not discuss the simple issue of
detectability given the accuracy of the flow
recording system. The design of most large
weirs is simply not suitable for accurately
measuring base-flows or relatively small                                                                                                                                                                                                                                                                                                                                                                                                                                                                                                                                                                                                                                                                                                         | on their relative size, the accuracy of the
measurement devices, background variability of
the signal and length of observation period."                                                                                                                                                                                                                                                                                                                                                                                                                                                                                                                                                                                                               |
| They note that this has been shown in small
catchments but not in large catchments. Yet
they do not discuss the simple issue of
detectability given the accuracy of the flow
recording system. The design of most large
weirs is simply not suitable for accurately
measuring base-flows or relatively small
changes in flows, which means that changes                                                                                                                                                                                                                                                                                                                                                                                                                                                                                                                                                                                                                                                           | on their relative size, the accuracy of the
measurement devices, background variability of
the signal and length of observation period."                                                                                                                                                                                                                                                                                                                                                                                                                                                                                                                                                                                                               |
| They note that this has been shown in small
catchments but not in large catchments. Yet
they do not discuss the simple issue of
detectability given the accuracy of the flow
recording system. The design of most large
weirs is simply not suitable for accurately
measuring base-flows or relatively small
changes in flows, which means that changes
have to affect most of the catchment to be                                                                                                                                                                                                                                                                                                                                                                                                                                                                                                                                                                                                             | on their relative size, the accuracy of the
measurement devices, background variability of
the signal and length of observation period."                                                                                                                                                                                                                                                                                                                                                                                                                                                                                                                                                                                                               |
| They note that this has been shown in small
catchments but not in large catchments. Yet
they do not discuss the simple issue of
detectability given the accuracy of the flow
recording system. The design of most large
weirs is simply not suitable for accurately
measuring base-flows or relatively small
changes in flows, which means that changes
have to affect most of the catchment to be
detectable and accurately quantifiable.                                                                                                                                                                                                                                                                                                                                                                                                                                                                                                                                                                  | on their relative size, the accuracy of the
measurement devices, background variability of
the signal and length of observation period."                                                                                                                                                                                                                                                                                                                                                                                                                                                                                                                                                                                                               |
| They note that this has been shown in small
catchments but not in large catchments. Yet
they do not discuss the simple issue of
detectability given the accuracy of the flow
recording system. The design of most large
weirs is simply not suitable for accurately
measuring base-flows or relatively small
changes in flows, which means that changes
have to affect most of the catchment to be
detectable and accurately quantifiable.
In line 247-249 they describe Qa,t as being                                                                                                                                                                                                                                                                                                                                                                                                                                                                                                                   | on their relative size, the accuracy of the
measurement devices, background variability of
the signal and length of observation period."
Reviewer is probably working in an                                                                                                                                                                                                                                                                                                                                                                                                                                                                                                                                                                         |
| They note that this has been shown in small
catchments but not in large catchments. Yet
they do not discuss the simple issue of
detectability given the accuracy of the flow
recording system. The design of most large
weirs is simply not suitable for accurately
measuring base-flows or relatively small
changes in flows, which means that changes
have to affect most of the catchment to be
detectable and accurately quantifiable.
In line 247-249 they describe Qa,t as being
interpretable as effective rainfall. My                                                                                                                                                                                                                                                                                                                                                                                                                                                                        | on their relative size, the accuracy of the
measurement devices, background variability of
the signal and length of observation period."
Reviewer is probably working in an
environment where "events" are clearly                                                                                                                                                                                                                                                                                                                                                                                                                                                                                                                               |
| They note that this has been shown in small
catchments but not in large catchments. Yet
they do not discuss the simple issue of
detectability given the accuracy of the flow
recording system. The design of most large
weirs is simply not suitable for accurately
measuring base-flows or relatively small
changes in flows, which means that changes
have to affect most of the catchment to be
detectable and accurately quantifiable.
In line 247-249 they describe Qa,t as being
interpretable as effective rainfall. My
understanding effective rainfall is that is this is                                                                                                                                                                                                                                                                                                                                                                                                                 | on their relative size, the accuracy of the
measurement devices, background variability of
the signal and length of observation period."
Reviewer is probably working in an
environment where "events" are clearly
separated, whereas we start from daily flow                                                                                                                                                                                                                                                                                                                                                                                                                                                                                |
| They note that this has been shown in small
catchments but not in large catchments. Yet
they do not discuss the simple issue of
detectability given the accuracy of the flow
recording system. The design of most large
weirs is simply not suitable for accurately
measuring base-flows or relatively small
changes in flows, which means that changes
have to affect most of the catchment to be
detectable and accurately quantifiable.
In line 247-249 they describe Qa,t as being
interpretable as effective rainfall. My
understanding effective rainfall is that is this is
simple a measure of how much rainfall is taken                                                                                                                                                                                                                                                                                                                                                               | on their relative size, the accuracy of the
measurement devices, background variability of
the signal and length of observation period."
Reviewer is probably working in an
environment where "events" are clearly
separated, whereas we start from daily flow
observations as such not knowing time-space                                                                                                                                                                                                                                                                                                                                                                                                                                 |
| They note that this has been shown in small
catchments but not in large catchments. Yet
they do not discuss the simple issue of
detectability given the accuracy of the flow
recording system. The design of most large
weirs is simply not suitable for accurately
measuring base-flows or relatively small
changes in flows, which means that changes
have to affect most of the catchment to be
detectable and accurately quantifiable.
In line 247-249 they describe Qa,t as being
interpretable as effective rainfall. My
understanding effective rainfall is that is this is
simple a measure of how much rainfall is taken
up by the soil. Only a daily time step (which is                                                                                                                                                                                                                                                                                                           | on their relative size, the accuracy of the
measurement devices, background variability of
the signal and length of observation period."
Reviewer is probably working in an
environment where "events" are clearly
separated, whereas we start from daily flow
observations as such not knowing time-space
details of rainfall. We thus don't know the time                                                                                                                                                                                                                                                                                                                                                                             |
| They note that this has been shown in small
catchments but not in large catchments. Yet
they do not discuss the simple issue of
detectability given the accuracy of the flow
recording system. The design of most large
weirs is simply not suitable for accurately
measuring base-flows or relatively small
changes in flows, which means that changes
have to affect most of the catchment to be
detectable and accurately quantifiable.
In line 247-249 they describe Qa,t as being
interpretable as effective rainfall. My
understanding effective rainfall is that is this is
simple a measure of how much rainfall is taken
up by the soil. Only a daily time step (which is
what they argue is logical to use) much of this                                                                                                                                                                                                                                                        | on their relative size, the accuracy of the
measurement devices, background variability of
the signal and length of observation period."
Reviewer is probably working in an
environment where "events" are clearly
separated, whereas we start from daily flow
observations as such not knowing time-space
details of rainfall. We thus don't know the time
of rainfall relative to the time of observing river                                                                                                                                                                                                                                                                                                                      |
| They note that this has been shown in small
catchments but not in large catchments. Yet
they do not discuss the simple issue of
detectability given the accuracy of the flow
recording system. The design of most large
weirs is simply not suitable for accurately
measuring base-flows or relatively small
changes in flows, which means that changes
have to affect most of the catchment to be
detectable and accurately quantifiable.
In line 247-249 they describe Qa,t as being
interpretable as effective rainfall. My
understanding effective rainfall is that is this is
simple a measure of how much rainfall is taken
up by the soil. Only a daily time step (which is
what they argue is logical to use) much of this
water will not appear as flow over 24 hours                                                                                                                                                                                                         | on their relative size, the accuracy of the
measurement devices, background variability of
the signal and length of observation period."
Reviewer is probably working in an
environment where "events" are clearly
separated, whereas we start from daily flow
observations as such not knowing time-space
details of rainfall. We thus don't know the time
of rainfall relative to the time of observing river
flow, nor the time it takes for rivers (at the                                                                                                                                                                                                                                                                    |
| They note that this has been shown in small
catchments but not in large catchments. Yet
they do not discuss the simple issue of
detectability given the accuracy of the flow
recording system. The design of most large
weirs is simply not suitable for accurately
measuring base-flows or relatively small
changes in flows, which means that changes
have to affect most of the catchment to be
detectable and accurately quantifiable.
In line 247-249 they describe Qa,t as being
interpretable as effective rainfall. My
understanding effective rainfall is that is this is
simple a measure of how much rainfall is taken
up by the soil. Only a daily time step (which is
what they argue is logical to use) much of this
water will not appear as flow over 24 hours
unless the catchment is already saturated or                                                                                                                                                         | on their relative size, the accuracy of the
measurement devices, background variability of
the signal and length of observation period."
Reviewer is probably working in an
environment where "events" are clearly
separated, whereas we start from daily flow
observations as such not knowing time-space
details of rainfall. We thus don't know the time
of rainfall relative to the time of observing river
flow, nor the time it takes for rivers (at the
point of observation) to respond.                                                                                                                                                                                                                               |
| They note that this has been shown in small
catchments but not in large catchments. Yet
they do not discuss the simple issue of
detectability given the accuracy of the flow
recording system. The design of most large
weirs is simply not suitable for accurately
measuring base-flows or relatively small
changes in flows, which means that changes
have to affect most of the catchment to be
detectable and accurately quantifiable.
In line 247-249 they describe Qa,t as being
interpretable as effective rainfall. My
understanding effective rainfall is that is this is
simple a measure of how much rainfall is taken
up by the soil. Only a daily time step (which is
what they argue is logical to use) much of this
water will not appear as flow over 24 hours
unless the catchment is already saturated or
has some rapid flow response mechanisms (i.e.                                                                                                        | on their relative size, the accuracy of the
measurement devices, background variability of
the signal and length of observation period."
Reviewer is probably working in an
environment where "events" are clearly
separated, whereas we start from daily flow
observations as such not knowing time-space
details of rainfall. We thus don't know the time
of rainfall relative to the time of observing river
flow, nor the time it takes for rivers (at the
point of observation) to respond.
Effective rainfall is normally defined from the
paremention of what reaches rivers                                                                                                                                      |
| They note that this has been shown in small
catchments but not in large catchments. Yet
they do not discuss the simple issue of
detectability given the accuracy of the flow
recording system. The design of most large
weirs is simply not suitable for accurately
measuring base-flows or relatively small
changes in flows, which means that changes
have to affect most of the catchment to be
detectable and accurately quantifiable.
In line 247-249 they describe Qa,t as being
interpretable as effective rainfall. My
understanding effective rainfall is that is this is
simple a measure of how much rainfall is taken
up by the soil. Only a daily time step (which is
what they argue is logical to use) much of this
water will not appear as flow over 24 hours
unless the catchment is already saturated or
has some rapid flow response mechanisms (i.e.
rapid interflow). So it will not be adequately
measured by Oa t which poods to be made           | on their relative size, the accuracy of the
measurement devices, background variability of
the signal and length of observation period."
Reviewer is probably working in an
environment where "events" are clearly
separated, whereas we start from daily flow
observations as such not knowing time-space
details of rainfall. We thus don't know the time
of rainfall relative to the time of observing river
flow, nor the time it takes for rivers (at the
point of observation) to respond.
Effective rainfall is normally defined from the
perspective of what reaches rivers.                                                                                                                                     |
| They note that this has been shown in small
catchments but not in large catchments. Yet
they do not discuss the simple issue of
detectability given the accuracy of the flow
recording system. The design of most large
weirs is simply not suitable for accurately
measuring base-flows or relatively small
changes in flows, which means that changes
have to affect most of the catchment to be
detectable and accurately quantifiable.
In line 247-249 they describe Qa,t as being
interpretable as effective rainfall. My
understanding effective rainfall is that is this is
simple a measure of how much rainfall is taken
up by the soil. Only a daily time step (which is
what they argue is logical to use) much of this
water will not appear as flow over 24 hours
unless the catchment is already saturated or
has some rapid flow response mechanisms (i.e.
rapid interflow). So it will not be adequately
measured by Qa,t which needs to be made
clear  | on their relative size, the accuracy of the
measurement devices, background variability of
the signal and length of observation period."
Reviewer is probably working in an
environment where "events" are clearly
separated, whereas we start from daily flow
observations as such not knowing time-space
details of rainfall. We thus don't know the time
of rainfall relative to the time of observing river
flow, nor the time it takes for rivers (at the
point of observation) to respond.
Effective rainfall is normally defined from the
perspective of what reaches rivers.
We are considering the statistical properties of
the O 2 frequency distribution rather than                        |
| They note that this has been shown in small
catchments but not in large catchments. Yet
they do not discuss the simple issue of
detectability given the accuracy of the flow
recording system. The design of most large
weirs is simply not suitable for accurately
measuring base-flows or relatively small
changes in flows, which means that changes
have to affect most of the catchment to be
detectable and accurately quantifiable.
In line 247-249 they describe Qa,t as being
interpretable as effective rainfall. My
understanding effective rainfall is that is this is
simple a measure of how much rainfall is taken
up by the soil. Only a daily time step (which is
what they argue is logical to use) much of this
water will not appear as flow over 24 hours
unless the catchment is already saturated or
has some rapid flow response mechanisms (i.e.
rapid interflow). So it will not be adequately
measured by Qa,t which needs to be made
clear. | on their relative size, the accuracy of the
measurement devices, background variability of
the signal and length of observation period."
Reviewer is probably working in an
environment where "events" are clearly
separated, whereas we start from daily flow
observations as such not knowing time-space
details of rainfall. We thus don't know the time
of rainfall relative to the time of observing river
flow, nor the time it takes for rivers (at the
point of observation) to respond.
Effective rainfall is normally defined from the
perspective of what reaches rivers.
We are considering the statistical properties of
the Q a frequency distribution, rather than
individual values. |

| There are a number of places where they abruptly introduce new symbols and fail to | We have made clear in the revised text that $Q_T$ and $Q_0$ are equal to $Q_t$ for t=T and t=0, |
|------------------------------------------------------------------------------------|-------------------------------------------------------------------------------------------------|
| explain them e.g. line 237 QT and Qo are not explained.                            | respectively                                                                                    |
| There are still several basic typographical errors                                 | Apologies (the spellchecker had been                                                            |
| should not be in a submitted MS                                                    | We had further help in the current ms version                                                   |
| Reviewer 2                                                                         |                                                                                                 |
| Referee Report General Comments: Both of                                           |                                                                                                 |
| these articles are relevant and could provide                                      |                                                                                                 |
| contributions to decision makers, as well as                                       |                                                                                                 |
| those working with ecosystem services and                                          |                                                                                                 |
| flood risk. The authors adequately addressed                                       |                                                                                                 |
| all of my concerns within the original                                             |                                                                                                 |
| submission making the subsequent paper much                                        |                                                                                                 |
| more suitable for publication. The separation                                      |                                                                                                 |
| of the original article into two greatly                                           |                                                                                                 |
| strengthens the study.                                                             |                                                                                                 |
| Specific Comments:                                                                 |                                                                                                 |
| Abstract: Abstracts need to be shortened. The                                      | Addressed                                                                                       |
| abstracts are far too verbose as they stand. A                                     |                                                                                                 |
| discussions of equations/parameters will make                                      |                                                                                                 |
| them clearer and more engaging.                                                    |                                                                                                 |
| PART I Figure 1: This figure is much improved                                      | Addressed, explanation of step 10 added in the                                                  |
| and far easier to follow, however it seems a                                       | caption (consistent with its reference in the                                                   |
| little blurry. Misspelling of pathways at line                                     | text)                                                                                           |
| 727. No explanation of step 10 in figure                                           |                                                                                                 |
| description.                                                                       |                                                                                                 |
|                                                                                    |                                                                                                 |
| Figure 4: This figure is blurry making the small                                   | Thanks, all corrections made                                                                    |
| text difficult to read. Avoid using contractions:                                  |                                                                                                 |
| For example, "don't" should be "do not" Lined                                      |                                                                                                 |
| 158: change doess to does. Line 196: It's                                          |                                                                                                 |
| Line 212: "The probably simplest" to "A simple"                                    | Thanks                                                                                          |
| Line 227-239: The wording of this sentence is                                      | Adjusted                                                                                        |
| confusing and requires some revisions.                                             | hujusteu                                                                                        |
| Line 456: "en" should be "an"                                                      | Thanks                                                                                          |
| Line 534-536: Spell out authors' full names.                                       | Addressed                                                                                       |
| PART II An explanation of the general land                                         |                                                                                                 |
| cover characteristics for each watershed would                                     |                                                                                                 |
| be helpful.                                                                        |                                                                                                 |
| Table 1 provides an element of confusion                                           |                                                                                                 |
| regarding the proportion of forested land that                                     |                                                                                                 |
| needs to be acknowledged.                                                          |                                                                                                 |
| Needs a conclusion section that summarizes                                         | We have added a section "conclusions" that                                                      |
| the study, discusses implications, and                                             | summarizes the discussion. Further research                                                     |
| acknowledges limitations and future research                                       | directions might be more appropriate in the                                                     |
| directions.                                                                        | discussion section.                                                                             |
| Line 817: add comma after part                                                     |                                                                                                 |

| Line 832: removed comma after intensity and      | Addressed                                         |
|--------------------------------------------------|---------------------------------------------------|
| response                                         |                                                   |
| Line 841: add "the" after "we consider"          | Addressed                                         |
| Line 848: change patchlevel to "the patch level" | Addressed                                         |
| Line 855 & 883: change "land-cover" to land      | Addressed                                         |
| "cover" (and throughout paper)                   |                                                   |
| Line 868: change "Fig." to "Figure" (and         | Addressed                                         |
| throughout paper); "provides" to "provide";      |                                                   |
| and remove "are"                                 |                                                   |
| Line 886: change "land-use" to "land use" (and   | Addressed                                         |
| throughout paper)                                |                                                   |
| Line 937: unsure what dace is supposed to        | Addressed                                         |
| mean                                             |                                                   |
| Line 945: add "the" before "measuring"           | Addressed                                         |
| Line 987: no supplementary information given     | Addressed                                         |
| Table 1: What is the differentiation between     | Addressed                                         |
| "forest" within land cover type and "natural     |                                                   |
| forest" at the bottom of the table? For Bialo    |                                                   |
| and Mae Chaem they are equal, but are            |                                                   |
| different for the other two. An explanation for  |                                                   |
| this must be given as the proportion of forested |                                                   |
| land is one of the primary drivers behind flow   |                                                   |
| predictability.                                  |                                                   |
| Figure 1: blurry                                 | Addressed                                         |
| Figure 5: Why are the water balance              | The sites have different rainfall patterns, soils |
| percentages different for the NatFor scenario    | and landscape properties, all influencing the     |
| when Figure App2 shows that the NatFor           | water balance.                                    |
| scenario is 100% for all watersheds?             |                                                   |
| Appendix 2: no proportions for Mae Chaem         | This land use type does not exist in Mae          |
| AgFor are given.                                 | Chaem, according the data we used.                |

| 1 | Flood risk reduction and flow buffering as ecosystem services:                             | Formatted: English (United Kingdom)     |
|---|--------------------------------------------------------------------------------------------|-----------------------------------------|
| 2 | I. Theory on a flow persistence indicator for watershed health                             |                                         |
| 3 | Meine van Noordwijk 1,2 , Lisa Tanika 1 , Betha Lusiana 1 |                                         |
| 4 | [1] {World Agroforestry Centre (ICRAF), SE Asia program, Bogor, Indonesia}                 |
| 5 | [2] {Wageningen University, Plant Production Systems, Wageningen, the Netherlands}         |
| 6 | Correspondence to: Meine van Noordwijk ( m.vannoordwijk@cgiar.org )                 |
| 7 |                                                                                            |                                         |

**8 Abstract 1**

9 We present and discuss a candidate here for a single parameter representation of the 10 complex concept of watershed quality that does align short and long term responses, 11 and provides bounds to the levels of unpredicataibility. Flow buffering in landscapes is 12 commonly interpreted as ecosystem service, but needs quantification, as f.Flood damage 13 reflects insufficient adaptation of human presence and activity to location and variability 14 (inherent plus induced) of river flow. Increased variability and reduced predictability of 15 river flow is a common sign, in public discourse, of degrading watersheds, combining 16 increased flooding risk and reduced low flows. Flow buffering in landscapes is 17 commonly interpreted as ecosystem service, but needs quantification. Geology, 18 landscape form, soil porosity, litter layer and surface features, drainage pathways, 19 vegetation and space-time patterns of rainfall interact in complex space-time patterns of 20 riverflowriver flow, but the anthropogenic aspects tend to get discussed on a one-21 dimensional scale of degradation and restoration. A strong tradition in public discourse 22 associates changes on such degradation-restoration axis with binary deforestation-23 reforestation shifts. changes in tree cover and/or forest qualityE, but the empirical 24 evidence for such link that may exist at high spatial resolution may not be a safe basis 25 for securing required flow buffering in landscapes at large. - Capturing the relationship 26 between the space-time patterns of rainfall and riverflow in a single buffering indicator 27 can help the way empirical evidence is summarized and projected change in land use 28 change scenarios is evaluated. Where space-time details of rainfall remain unknown, a 29 simpler approach is needed. We present and discuss a candidate here for a single 30 parameter representation of the complex concept of watershed quality that does align 31 short and long term responses, and provides bounds to the levels of unpredicatibility. 32 We define a The dimensionless FlowPer parameter  $F_p$  that  $(F_p)$  represents predictability 33 of river flow in a recursive flow model. Analysis suggests that buffering has two 34 interlinked effects: a smaller fraction of fresh rainfall enters the streams, and flow 35 becomes more persistent, in that the ratio of the flow on subsequent days has a higher 36 minimum level. It is defined through a recursive model of river flow,  $Q_t = F_p Q_{t-1} + (1 - 1) Q_{t-1}$ 37  $F_p$ )(Pt - Etx), that relates the flow Q on day t to that on the previous day (Qt-1), and a term 38 that reflects precipitation P on the day itself and evapotranspiration E in a preceding 39 time period, with Q, P and E expressed in mm d4. When summed over one or more

| 40 | years, this recursive model reflects the water balance ( $\sum Q = \sum P - \sum E$ ), once changes in           |
|----|------------------------------------------------------------------------------------------------------------------|
| 41 | the storage term that can dominate short term dynamics become negligible. $\mathrm{F}_{\mathrm{p}}$ varies       |
| 42 | between 0 and 1, and can be derived from a time series of measured (or modeled) river                            |
| 43 | flow data. In a parsimonious interpretation that aligns with data sets that only exist of                        |
| 44 | (daily) records of riverflow, the spatially averaged precipitation term $P_{t}$ and preceding                    |
| 45 | cumulative evapotranspiration since previous rain $E_{tx}\xspace$ are treated as constrained but                 |
| 46 | unknown, stochastic variables. Without knowing when peak flows occur, the balance                                |
| 47 | equation suggests that a decrease in $\mathrm{F}_{\mathrm{p}}$ from 0.9 to 0.8 means peak flow doubling from     |
| 48 | 10 to 20% of peak rainfall (minus its accompanying $E_{\rm tx}$ ). Flood duration has a nonlinear                |
| 49 | response to increases in $F_{\text{p}},$ as low $F_{\text{p}}$ values lead to high peak flow of short duration,  |
| 50 | and at high $F_{\rm p}$ values thresholds of flooding may never be reached. In a numerical                       |
| 51 | example a decrease in $F_p$ led at most to an increase in expected flood duration by 3 days.                     |
| 52 | As a potential indicator of watershed health (or quality), the $F_{\text{p}}$ metric (or its change              |
| 53 | over time from what appears to be the local norm) matches local knowledge concepts,                              |
| 54 | captures key aspects of the river flow dynamic and can be unambiguously derived from                             |
| 55 | empirical river flow data. Further exploration of responsiveness of $\mathrm{F}_{\mathrm{p}}$ to the interaction |
| 56 | of land cover and the specific realization of space-time patterns of rainfall in a limited                       |
| 57 | observation period is needed to test the interpretation of $F_{\underline{p}}$ as indicator of watershed         |
| 58 | health (or quality) in the way this is degrading or restoring through land cover change                          |
| 59 | and modifications of the overland and surface flow pathways, given inherent properties                           |
| 60 | such as geology, geomorphology and climate.                                                                      |

**1** Introduction**

T

| 62 | Degradation of watersheds and its consequences for river flow regime and flooding intensity             |
|----|---------------------------------------------------------------------------------------------------------|
| 63 | and frequency are a widespread concern (Brauman et alet ale 2007; Bishop and Pagiola, 2012;             |
| 64 | Winsemius et alet ale 2013). Current watershed rehabilitation programs that focus on                    |
| 65 | increasing tree cover in upper watersheds are only partly aligned with current scientific               |
| 66 | evidence of effects of large-scale tree planting on streamflow (Ghimire et alet alet 2014;              |
| 67 | Malmer et alet alet alet alet alet alet alet a                                                          |
| 68 | al., 2010). The relationship between floods and change in forest quality and quantity, and the          |
| 69 | availability of evidence for such a relationship at various scales has been widely discussed over       |
| 70 | the past decades (Andréassian, 2004; Bruijnzeel, 2004; Bradshaw et alet alet alet alet alet alet alet a |
|    |                                                                                                         |

| 71  | alat al 2000) Magguraments in | Cote d'Ivoire | for example   | showed strong | scale dependence |
|-----|-------------------------------|---------------|---------------|---------------|------------------|
| 1/1 | $\frac{1}{1}$                 |               | TOI CAAIIDIC. | Showed Shong  | scale dependence |

|        | Formatted: English (United Kingdom), Subscript |
|--------|------------------------------------------------|
|        | Formatted: English (United Kingdom)            |
| h      | Formatted: English (United Kingdom)            |
| //     | Formatted: English (United Kingdom)            |
| - ///  | Formatted: English (United Kingdom)            |
| /// ٨  | Formatted: English (United Kingdom)            |
| X      | Formatted: English (United Kingdom)            |
| / // X | Formatted: English (United Kingdom)            |
| // /X  | Formatted: English (United Kingdom)            |
| ///    | Formatted: English (United Kingdom)            |
| X      | Formatted: English (United Kingdom)            |
| / X    | Formatted: English (United Kingdom)            |
| M      | Formatted: English (United Kingdom)            |
|        | Formatted: English (United Kingdom)            |
|        | Formatted: English (United Kingdom)            |
| 1      | Formatted: English (United Kingdom)            |
| -1     | Formatted: English (United Kingdom)            |
|        | Formatted: English (United Kingdom)            |
|        | Formatted: English (United Kingdom)            |

of runoff from 30-50% at 1 m2 point scale, to 4% at 130 ha watershed scale, linked to spatial 72 73 variability of soil properties plus variations in rainfall patterns (Van de Giesen et alet al., 2000). 74 The ratio between peak and average flow decreases from headwater streams to main rivers in a predictable manner; while mean annual discharge scales with (area)1.0, maximum river flow 75 76 was found to scale with (area)0.7 on average (Rodríguez-Iturbe and Rinaldo, 2001; van Noordwijk et alet al., 1998). The determinants of peak flows are thus scale-dependent, with 77 78 space-time correlations in rainfall interacting with subcatchment-level flow buffering in 79 peakflows at any point along the river. Whether and where peakflowpeak flows lead to flooding 80 depends on the capacity of the rivers to pass on peakflowpeak flows towards downstream lakes 81 or the sea, assisted by riparian buffer areas with sufficient storage capacity (Baldasarre et alet 82 al., 2013); reducing local flooding risk by increased drainage increases flooding risk 83 downstream, challenging the nested-scales management of watersheds to find an optimal spatial 84 distribution, rather then minimization, of flooding probabilities. Well-studied effects of forest 85 conversion on peak flows in small upper stream catchments (Alila et alet ale 2009) do not 86 necessarily translate to flooding downstream. As summarized by Beck et alet alet (2013) meso-87 to macroscale catchment studies (>1 and >10 000 km2, respectively) in the tropics, subtropics, 88 and warm temperate regions have mostly failed to demonstrate a clear relationship between 89 river flow and change in forest area. Lack of evidence cannot be firmly interpreted as evidence 90 for lack of effect, however. Detectability of effects depends on their relative size, the accuracy 91 of the measurement devices, background variability of the signal and length of observation 92 period. A recent econometric study for Peninsular Malaysia by Tan-Soo et alet al. (2014) 93 concluded that, after appropriate corrections for space-time correlates in the data-set for 31 94 meso- and macroscale basins (554-28,643 km2), conversion of inland rain forest to 95 monocultural plantations of oil palm or rubber increased the number of flooding days reported, 96 but not the number of flood events, while conversion of wetland forests to urban areas reduced 97 downstream flood duration. This Malaysian study may be the first credible empirical evidence 98 at this scale. The difference between results for flood duration and flood frequency and the 99 result for draining wetland forests warrant further scrutiny. Consistency of these findings with 00 river flow models based on a water balance and likely pathways of water under the influence 101 of change in land cover and land use has yet to be shown. Two recent studies for Southern 102 China confirm the conventional perspective that deforestation increases high flows, but are 103 contrasting in effects of reforestation. Zhou et alet ale (2010) analyzed analysed a 50-year data 104 set for Guangdong Province in China and concluded that forest recovery had not changed the

**Formatted: English (United Kingdom)**

| Formatted: English (United Kingdom) |
|-------------------------------------|
| Formatted: English (United Kingdom) |
|                                     |
| Formatted: English (United Kingdom) |
|                                     |
| Formatted: English (United Kingdom) |
|                                     |

105 annual water yield (or its underpinning water balance terms precipitation and 106 evapotransipiration evapotranspiration), but had a statistically significant positive effect on dry 107 108 (6983 km2) in subtropical China that while historical deforestation had decreased the 109 magnitudes of low flows (daily flows  $\leq Q95\%$ ) by 30.1%, low flows were not significantly 110 improved by reforestation. They concluded that recovery of low flows by reforestation may 111 take much longer time than expected probably because of severe soil erosion and resultant loss 112 of soil infiltration capacity after deforestation. Changes in riverflow river flow patterns over a 113 limited period of time can be the combined and interactive effects of variations in the local 114 rainfall regime, land cover effects on soil structure and engineering modifications of water flow, that can be teased apart with modelling tools (Ma et alet al., 2014). 115

116 Lacombe et alet ale (2015) documented that the hydrological effects of natural regeneration 117 differ from those of plantation forestry, while forest statistics don't not normally differentiate 118 between these different land covers. In a regression study of the high and low flow regimes in 119 the Volta and Mekong river basins Lacombe and McCartney (2016) found that in the variation 120 among tributaries various aspects of land cover and land cover change had explanatory power. 121 Between the two basins, however, these aspects differed. In the Mekong basin variation in forest 22 cover had no direct effect on flows, but extending paddy areas resulted in a decrease in 123 downstream low flows, probably by increasing evapotranspiration in the dry season. In the 24 Volta River Basin, the conversion of forests to crops (or a reduction of tree cover in the existing 125 parkland system) induced greater downstream flood flows. This observation is aligned with the 126 experimental identification of an optimal, intermediate tree cover from the perspective of 127 groundwater recharge in parklands in Burkina Faso (Ilstedt et alet al. 2016).

The statistical challenges of attribution of cause and effect in such data-sets are considerable with land use/land cover interacting with spatially and temporally variable rainfall, geological

- 130 configuration and the fact that land use is not changing in random fashion or following any pre-
- randomized design (Alila et alet alx, 2009; Rudel et alet alx, 2005). Hydrologieical analysis
- relationships between the change in forest cover or urban area, and change in various flow
- 134 characteristics, despite indications that regrowing forests increased evapotranspiration. Yet, the
- concept of a 'regulating function' on river flow regime for forests and other semi-natural
- 136 ecosystems is widespread. The considerable human and economic costs of flooding at locations

| Formatted: English (United Kingdom) |
|-------------------------------------|
|                                     |
| Formatted: English (United Kingdom) |
| Formatted: English (United Kingdom) |

**Formatted: English (United Kingdom)**

| Formatted: English (United Kingdom) |
|-------------------------------------|
| Formatted: English (United Kingdom) |

137 and times beyond where this is expected make the presumed 'regulating function' on flood 138 reduction of high value (Brauman et alet alet 2007) - if only we could be sure that the effect is real, beyond the local scales (< 10 km2) of paired catchments where ample direct empirical 139 140 proof exists (Bruijnzeel, 1990, 2004). These observations imply that percent tree cover (or other 141 forest related indicators) is probably not a good metric for judging the ecosystem services 142 provided by a watershed (of different levels of 'health'), and that a metric more directly 143 reflecting changes in river flow may be needed. Here we will explore a simple recursive model 144 of river flow (van Noordwijk et alet ale 2011) that (i) is focused on (loss of) predictability, (ii) 145 can account for the types of results obtained by the cited recent Malaysian study (Tan-Soo et 146 alet ale 2014), and (iii) may constitute a suitable performance indicator to monitor watershed 147 'health' through time.

⇒ <del>Fig.</del>Figure 1

148

Figure 1 is compatible with a common dissection of risk as the product of hazard, exposure and vulnerability. Extreme discharge events plus river-level engineering co-determine hazard, while

exposure depends on topographic position interacting with human presence, and vulnerability

152 can be modified by engineering at a finer scale and be further reduced by advice to leave an

area in high-risk periods. A recent study (Jongman et alet alet alet 2015) found that human fatalities

and material losses between 1980 and 2010 expressed as a share of the exposed population and

gross domestic product were decreasing with rising income. The planning needed to avoid

extensive damage requires quantification of the risk of higher than usual discharges, especially

157 at the upper tail end of the flow frequency distribution.

158 The statistical scarcity, per definition, of 'extreme events' and the challenge of data collection 159 where they do occur, make it hard to rely on empirical data as such. Existing data on flood 160 frequency and duration, as well as human and economic damage are influenced by topography, 161 human population density and economic activity, interacting with engineered infrastructure 162 (step 4 and 5 in Fig.Figure 1), as well as the extreme rainfall events that are their proximate 163 cause. Subsidence due to groundwater extraction in urban areas of high population density is a 164 specific problem for a number of cities built on floodplains (such as Jakarta and Bangkok), but 165 subsidence of drained peat areas has also been found to increase flooding risks elsewhere 166 (Sumarga et alet ale: 2016). Common hydrological analysis of flood frequency (called 1 in 10-167 , 1 in 100-, 1 in 1000-year flood events, for example) doess not separately attribute flood 168 magnitude to rainfall and land use properties, and analysis of likely change in flood frequencies in the context of climate change adaptation has been challenging (Milly et alet al., 2002; Ma et 169

170 alet  $al_{\bar{x},v}$  2014). There is a lack of simple performance indicators for watershed health at its point 171 of relating precipitation P and river flow Q (step 2 in Figure 1) that align with local 172 observations of river behavior behaviour and concerns about its change and that can reconcile 173 local, public/policy and scientific knowledge, thereby helping negotiated change in watershed 174 management (Leimona et alet alex 2015). The behaviorbehaviour of rivers depends on many 175 climatic (step 1 in Figure 1) and terrain factors (step 7-9 in Figure 1) that make it a challenge 176 to differentiate between anthropogenically induced ecosystem structural change and soil 177 degradation (step 7a) on one hand and intrinsic variability on the other. Arrow 10 in Figure 1 178 represents the direct influence of climate on vegetation, but also a possible reverse influence 179 (van Noordwijk et alet ale., 2015b). Hydrological models tend to focus on predicting 180 hydrographs at one or more temporal scales, and are usually tested on data-sets from limited 181 locations. Despite many decades (if not centuries) of hydrological modelingmodelling, current 182 hydrologic theory, models and empirical methods have been found to be largely inadequate for 183 sound predictions in ungauged basins (Hrachowitz et alet alr., 2013). Efforts to resolve this 184 through harmonization of modelling strategies have so far failed. Existing models differ in the 185 number of explanatory variables and parameters they use, but are generally dependent on 186 empirical data of rainfall that are available for specific measurement points but not at the spatial 187 resolution that is required for a close match between measured and modeled modelled river flow. 188 189 degrees of freedom and too many opportunities for getting right answers for wrong reasons if 190 used for empirical calibration (Beven, 2011). Parsimonious, parameter-sparse models are 191 appropriate for the level of evidence available to constrain them, but these parameters are 192 themselves implicitly influenced by many aspects of existing and changing features of the 193 watershed, making it hard to use such models for scenario studies of interacting land use and 94 climate change. Here we present a more direct approach deriving a metric of flow predictability 195 that can bridge local concerns and concepts to quantified hydrologic function: the 'flow 96 persistence' parameter (step 2 in Figure 1). 197 In this contribution to the debate we will first define the metric 'flow persistence' in the context

of temporal autocorrelation of river flow and then derive a way to estimate its numerical value. In part II we will apply the algorithm to river flow data for a number of contrasting meso-scale watersheds. In the discussion of this paper we will consider the new flow persistence metric in terms of three groups of criteria for usable knowledge (Clark et alct alc., 2011; Lusiana et alct Formatted: English (United Kingdom)

| Formatted: English (United Kingdom) |   |
|-------------------------------------|---|
| Formatted: English (United Kingdom) |   |
| Formatted: English (United Kingdom) | _ |

| 202 | al; 2011; Leimona et alet al; 2015) based on salience (1,2), credibility (3,4) and legitimacy                                                                                                                                                                                                                                                                                                                                                                                                                                                                                                                                                                                                                                                                                                                                                                                                                                                                                                                                                                                                                                                                                                                                                                                                                                                                                                                                                                                                                                                                                                                                                                                                                                                                                                                                                                                                                                                                                                                                                                                                                                                                                                                                                                                                                                                                                                                                                                                                                                                                                                                                                                                                                                                                                                                                                                                                                                                                                                                                                                                                                                                                                                                                                                                                                                                                                                                                                                                                                                                                                                                                                | Formatted: English (United Kingdom)        |
|-----|----------------------------------------------------------------------------------------------------------------------------------------------------------------------------------------------------------------------------------------------------------------------------------------------------------------------------------------------------------------------------------------------------------------------------------------------------------------------------------------------------------------------------------------------------------------------------------------------------------------------------------------------------------------------------------------------------------------------------------------------------------------------------------------------------------------------------------------------------------------------------------------------------------------------------------------------------------------------------------------------------------------------------------------------------------------------------------------------------------------------------------------------------------------------------------------------------------------------------------------------------------------------------------------------------------------------------------------------------------------------------------------------------------------------------------------------------------------------------------------------------------------------------------------------------------------------------------------------------------------------------------------------------------------------------------------------------------------------------------------------------------------------------------------------------------------------------------------------------------------------------------------------------------------------------------------------------------------------------------------------------------------------------------------------------------------------------------------------------------------------------------------------------------------------------------------------------------------------------------------------------------------------------------------------------------------------------------------------------------------------------------------------------------------------------------------------------------------------------------------------------------------------------------------------------------------------------------------------------------------------------------------------------------------------------------------------------------------------------------------------------------------------------------------------------------------------------------------------------------------------------------------------------------------------------------------------------------------------------------------------------------------------------------------------------------------------------------------------------------------------------------------------------------------------------------------------------------------------------------------------------------------------------------------------------------------------------------------------------------------------------------------------------------------------------------------------------------------------------------------------------------------------------------------------------------------------------------------------------------------------------------------------|--------------------------------------------|
| 203 | (5-7):                                                                                                                                                                                                                                                                                                                                                                                                                                                                                                                                                                                                                                                                                                                                                                                                                                                                                                                                                                                                                                                                                                                                                                                                                                                                                                                                                                                                                                                                                                                                                                                                                                                                                                                                                                                                                                                                                                                                                                                                                                                                                                                                                                                                                                                                                                                                                                                                                                                                                                                                                                                                                                                                                                                                                                                                                                                                                                                                                                                                                                                                                                                                                                                                                                                                                                                                                                                                                                                                                                                                                                                                                                       | Formatted: English (United Kingdom)        |
| 004 | 1 Does flow percistance relate to important espects of watershed behaviorhelewiour?                                                                                                                                                                                                                                                                                                                                                                                                                                                                                                                                                                                                                                                                                                                                                                                                                                                                                                                                                                                                                                                                                                                                                                                                                                                                                                                                                                                                                                                                                                                                                                                                                                                                                                                                                                                                                                                                                                                                                                                                                                                                                                                                                                                                                                                                                                                                                                                                                                                                                                                                                                                                                                                                                                                                                                                                                                                                                                                                                                                                                                                                                                                                                                                                                                                                                                                                                                                                                                                                                                                                                          | Formatted: English (United Kingdom)        |
| 204 | 1. Does now persistence relate to important aspects of watersned <del>behaviorbehaviourt</del>                                                                                                                                                                                                                                                                                                                                                                                                                                                                                                                                                                                                                                                                                                                                                                                                                                                                                                                                                                                                                                                                                                                                                                                                                                                                                                                                                                                                                                                                                                                                                                                                                                                                                                                                                                                                                                                                                                                                                                                                                                                                                                                                                                                                                                                                                                                                                                                                                                                                                                                                                                                                                                                                                                                                                                                                                                                                                                                                                                                                                                                                                                                                                                                                                                                                                                                                                                                                                                                                                                                                        | Formatted: English (United Kingdom)        |
| 205 | 2. Does it 2 s quantification help to select management actions?                                                                                                                                                                                                                                                                                                                                                                                                                                                                                                                                                                                                                                                                                                                                                                                                                                                                                                                                                                                                                                                                                                                                                                                                                                                                                                                                                                                                                                                                                                                                                                                                                                                                                                                                                                                                                                                                                                                                                                                                                                                                                                                                                                                                                                                                                                                                                                                                                                                                                                                                                                                                                                                                                                                                                                                                                                                                                                                                                                                                                                                                                                                                                                                                                                                                                                                                                                                                                                                                                                                                                                  | Formatted: English (United Kingdom)        |
| 206 | 3. Is there consistency of numerical results?                                                                                                                                                                                                                                                                                                                                                                                                                                                                                                                                                                                                                                                                                                                                                                                                                                                                                                                                                                                                                                                                                                                                                                                                                                                                                                                                                                                                                                                                                                                                                                                                                                                                                                                                                                                                                                                                                                                                                                                                                                                                                                                                                                                                                                                                                                                                                                                                                                                                                                                                                                                                                                                                                                                                                                                                                                                                                                                                                                                                                                                                                                                                                                                                                                                                                                                                                                                                                                                                                                                                                                                                |                                            |
| 207 | 4. How sensitive is it to bias and random error in data sources?                                                                                                                                                                                                                                                                                                                                                                                                                                                                                                                                                                                                                                                                                                                                                                                                                                                                                                                                                                                                                                                                                                                                                                                                                                                                                                                                                                                                                                                                                                                                                                                                                                                                                                                                                                                                                                                                                                                                                                                                                                                                                                                                                                                                                                                                                                                                                                                                                                                                                                                                                                                                                                                                                                                                                                                                                                                                                                                                                                                                                                                                                                                                                                                                                                                                                                                                                                                                                                                                                                                                                                             |                                            |
| 208 | 5. Does it match local knowledge?                                                                                                                                                                                                                                                                                                                                                                                                                                                                                                                                                                                                                                                                                                                                                                                                                                                                                                                                                                                                                                                                                                                                                                                                                                                                                                                                                                                                                                                                                                                                                                                                                                                                                                                                                                                                                                                                                                                                                                                                                                                                                                                                                                                                                                                                                                                                                                                                                                                                                                                                                                                                                                                                                                                                                                                                                                                                                                                                                                                                                                                                                                                                                                                                                                                                                                                                                                                                                                                                                                                                                                                                            |                                            |
| 209 | 6. Can it be used to empower local stakeholders of watershed management?                                                                                                                                                                                                                                                                                                                                                                                                                                                                                                                                                                                                                                                                                                                                                                                                                                                                                                                                                                                                                                                                                                                                                                                                                                                                                                                                                                                                                                                                                                                                                                                                                                                                                                                                                                                                                                                                                                                                                                                                                                                                                                                                                                                                                                                                                                                                                                                                                                                                                                                                                                                                                                                                                                                                                                                                                                                                                                                                                                                                                                                                                                                                                                                                                                                                                                                                                                                                                                                                                                                                                                     |                                            |
| 210 | 7. Can it inform local risk management?                                                                                                                                                                                                                                                                                                                                                                                                                                                                                                                                                                                                                                                                                                                                                                                                                                                                                                                                                                                                                                                                                                                                                                                                                                                                                                                                                                                                                                                                                                                                                                                                                                                                                                                                                                                                                                                                                                                                                                                                                                                                                                                                                                                                                                                                                                                                                                                                                                                                                                                                                                                                                                                                                                                                                                                                                                                                                                                                                                                                                                                                                                                                                                                                                                                                                                                                                                                                                                                                                                                                                                                                      |                                            |
| 211 | Questions 3 and 4 will get specific attention in part II.                                                                                                                                                                                                                                                                                                                                                                                                                                                                                                                                                                                                                                                                                                                                                                                                                                                                                                                                                                                                                                                                                                                                                                                                                                                                                                                                                                                                                                                                                                                                                                                                                                                                                                                                                                                                                                                                                                                                                                                                                                                                                                                                                                                                                                                                                                                                                                                                                                                                                                                                                                                                                                                                                                                                                                                                                                                                                                                                                                                                                                                                                                                                                                                                                                                                                                                                                                                                                                                                                                                                                                                    | Formatted: Check spelling and grammar      |
|     |                                                                                                                                                                                                                                                                                                                                                                                                                                                                                                                                                                                                                                                                                                                                                                                                                                                                                                                                                                                                                                                                                                                                                                                                                                                                                                                                                                                                                                                                                                                                                                                                                                                                                                                                                                                                                                                                                                                                                                                                                                                                                                                                                                                                                                                                                                                                                                                                                                                                                                                                                                                                                                                                                                                                                                                                                                                                                                                                                                                                                                                                                                                                                                                                                                                                                                                                                                                                                                                                                                                                                                                                                                              | Formatted: English (United Kingdom)        |
| 212 | 2 Recursive river flow model and flow persistence                                                                                                                                                                                                                                                                                                                                                                                                                                                                                                                                                                                                                                                                                                                                                                                                                                                                                                                                                                                                                                                                                                                                                                                                                                                                                                                                                                                                                                                                                                                                                                                                                                                                                                                                                                                                                                                                                                                                                                                                                                                                                                                                                                                                                                                                                                                                                                                                                                                                                                                                                                                                                                                                                                                                                                                                                                                                                                                                                                                                                                                                                                                                                                                                                                                                                                                                                                                                                                                                                                                                                                                            |                                            |
|     |                                                                                                                                                                                                                                                                                                                                                                                                                                                                                                                                                                                                                                                                                                                                                                                                                                                                                                                                                                                                                                                                                                                                                                                                                                                                                                                                                                                                                                                                                                                                                                                                                                                                                                                                                                                                                                                                                                                                                                                                                                                                                                                                                                                                                                                                                                                                                                                                                                                                                                                                                                                                                                                                                                                                                                                                                                                                                                                                                                                                                                                                                                                                                                                                                                                                                                                                                                                                                                                                                                                                                                                                                                              |                                            |
| 213 | 2.1 Basic equations                                                                                                                                                                                                                                                                                                                                                                                                                                                                                                                                                                                                                                                                                                                                                                                                                                                                                                                                                                                                                                                                                                                                                                                                                                                                                                                                                                                                                                                                                                                                                                                                                                                                                                                                                                                                                                                                                                                                                                                                                                                                                                                                                                                                                                                                                                                                                                                                                                                                                                                                                                                                                                                                                                                                                                                                                                                                                                                                                                                                                                                                                                                                                                                                                                                                                                                                                                                                                                                                                                                                                                                                                          |                                            |
| 214 | One of the easiest-to-observe aspects of a river is its day-to-day fluctuation in waterlevelwater                                                                                                                                                                                                                                                                                                                                                                                                                                                                                                                                                                                                                                                                                                                                                                                                                                                                                                                                                                                                                                                                                                                                                                                                                                                                                                                                                                                                                                                                                                                                                                                                                                                                                                                                                                                                                                                                                                                                                                                                                                                                                                                                                                                                                                                                                                                                                                                                                                                                                                                                                                                                                                                                                                                                                                                                                                                                                                                                                                                                                                                                                                                                                                                                                                                                                                                                                                                                                                                                                                                                            |                                            |
| 215 | level, related to the volumetric flow (discharge) via rating curves (Maidment, 1992). Without

---

## Author Response (AR3)

Dear Editor

We were pleased to note the progress that both reviewers have now recommended publication of the manuscript and we certainly appreciate the rigour of the review and editorial process.

| Editor Decision: Publish subject to revisions (further review by Editor and Referees) (29 Oct 2016) by Prof. Jan Seibert Comments to the Author: Thanks for your efforts with the revisions, which clarified a number of points. At the same time, reading your manuscript again, I also realized that there are still a number of issues: | Our responses: In response to your remaining comments we have made some further changes to the manuscript that we hope address the remaining issues: |
|---|---|
| 1) The runoff values you present in figures 2-4 seem unrealistically high. Could you please comment on these values. As it is not just one figure, I am afraid there is something fundamentally wrong - or I misunderstand something. | Indeed the example presented in Figures 2 and 3 was from an exceptionally wet climate, with about 6000 mm y-1 of rainfall. We have replaced the figure with one for a 1600 mm y-1 example which is more typical of the humid tropics. We have not altered Fig 4 which includes some high flow rates. |
| 2) Your approach basically is similar what people previously have used as flow separation techniques. While I can see your point that you are using these techniques here with a different goal, I still think that it for the sake of scientific clarity is mandatory to clearly link to the previous work and to better describe what is similar/different with your approach. | Thank you for the suggestion. We have further explored the relationship between the Fp method and existing flow separation procedures and added a paragraph discussing two new figures: one that illustrates how different methods interpret a given hydrograph, and one that compares interannual variation in the different metrics derived from the four catchments described in further detail in Paper 2. We added: "As indicated, the $F_p$ method is related to earlier methods used in streamflow hydrograph separation of base flow and quick flow. While textbooks (Ward and Robinson, 2000; Hornberger et al 2014) tend to be critical of the lack of objectivity of graphical methods, algorithms are used for deriving the minimum flow in a fixed or sliding period of reference as base flow (Sloto and Crouse, 1996; Furey and Gupta, 2001). The time interval used for deriving the minimum flow depends on catchment size. Figure 6 compares results for a hydrograph of a single year of one of the catchments described in more detail in paper II. While there is agreement on most of what is indicated ass baseflow, the short term response to peaks in the flow differ, with baseflow in the $F_p$ method more rapidly increasing after peak events. When compared across multiple years |

for the four catchments describe in detail in paper II, there is partial agreement in the way interannual variation is described in each catchment, while numerical values are similar, but the ratio of what is indicated as baseflow according to the $F_p$ method and according to standard hydrograph separation varies from 1.05 to 0.86."

We also added:
"Recursive models that describe flow in a next time interval on the basis of a fraction of that in the preceding time interval with a term for additional flow due to additional rainfall have been used in analysis of peak flow event before, with time intervals as short as 1 minute rather than the 1 day we use here (Rose, 2004). Through reference to an overall mass balance a relationship similar to what we found here ($F_p$ times preceding flow plus $1 - F_p$ times recent inputs) was also used in such models. To our knowledge, the method we describe here at daily timescales has not been used before."

And

"The idea that the form of the storage-discharge function can be estimated from analysis of streamflow fluctuations has been explored before for a class of catchments in which discharge is determined by the volume of water in storage (Kirchner, 2009). Such catchments behave as simple first-order nonlinear dynamical systems and can be characterized in a single-equation rainfall-runoff model that predicted streamflow, in a test catchment in Wales, as accurately as other models that are much more highly parameterized. This model of the dQ/dt versus Q relationship can also be analytically inverted; thus, it can be used to "do hydrology backward," that is, to infer time series of whole-catchment precipitation directly from fluctuations in streamflow. The slope of the log-log relationship between flow recession (dQ/dt) and Q that Kirchner (2009) used is conceptually similar to the $F_p$ metric we derived here, but the specific algorithm to derive the parameter from empirical data differs. Estimates of dQ/dt are sensitive to noise in the measurement of Q and the possibly frequent and small

| | increases in Q can be separated from the expected flow recession in the algorithm we presented here." |
|---|---|
| | And |
| | "Seifert and Beven (2009) discussed the increase in predictive skill of models depending on the amount of location-specific data that can be used to constrain them. They found that the ensemble prediction of multiple models for a single location clearly outperformed the predictions using single parameter sets and that surprisingly little runoff data was necessary to identify model parameterizations that provided good results for "ungauged" test periods in cases where actual measurements were available. Their results indicated that a few runoff measurements can contain much of the information content of continuous runoff time series. The way these conclusions might be modified if continuous measurements for limited time periods, rather than separated single data points on river flow could be used, remains to be explored. Their study indicated that results may differ significantly between catchments and critical tests of $F_p$ across multiple situations are obviously needed, as paper II will provide. " |
| 3) Fp is affected by both rainfall time series and catchment characteristics. It is therefore not clear to me whther Fp really is a good measure of catchment status since Fp also could change without any catchment change if rainfall distributions change. | Indeed, that is the conclusion we formulate in the second paper, based on the case studies analysed. Where $F_p$ describes behaviour of a river with direct relevance for downstream populations, it is reflecting the "health" of the way a watershed interacts with its climate, rather than the land cover as such. |
| | We added: |
| | "In conclusion, the $F_p$ metric appears to allow an efficient way of summarizing complex landscape processes into a single parameter that reflects the effects of landscape management within the context of the local climate. If rainfall patterns change but the landscape does not, the resultant flow patterns may reflect a change in watershed health (van Noordwijk et al., 2016)." |
| 4) Human impacts can both increase and decrease Fp, again, I am wondering whether Fp really is a good measure of 'watershed health' | We try to provide empirical evidence to help readers answer this question for themselves. The "health" concept we use is a comprehensive one of the way climate, watershed and engineering interventions interact on functional aspects of river flow. |

| | |
|---|---|
| 5) You claim that Fp is related to flood risks. can this be shown on real data? I am not so sure I would agree. Your approach basically assumes that catchment changes result in linear changes in flood risks, but this is obviously often not the case. For instance, if a medium-size reservoir is build, this would increase Fp, but the largest floods would hardly be influenced (because then the reservoir is filled anyway). I guess, another issue here is that the Fp values are based on the continuous, average catchment behavior, whereas floods are single extreme events. | We added: "Flood risks are themselves nonlinearly and in strongly topography-specific ways related to the volume of river flow after extreme rainfall events. While the expected fraction of rainfall that contributes to direct flow is linearly related to rainfall via (1-Fp), flooding risk as such will have a non-linear relationship with rainfall, that depends on topography and antecedent rainfall. Catchment changes, such as increases or decreases in percentage tree cover, will generally have a non-linear relationship with $F_p$ as well as with flooding risks."

In the rivers we considered there has been no major dams or reservoirs installed, and where they do exist the specific operating rules need to be included in any model. We have clarified this restriction of the empirical data in the discussion.
"The "health" concept we use is a comprehensive one of the way climate, watershed and engineering interventions interact on functional aspects of river flow. In the catchments we considered in part II there have been no major dams or reservoirs installed. Ma et al (2014) described a method to separate these three influences on river flow. Where these do exist the specific operating rules of reservoirs need to be included in any model and these can have a major influence on downstream flow, depending on the primary use for power generation, dry season irrigation or stabilizing river flow for riverine transport." |
| 6) Your approach is not at all applicable to catchments with snow. This should be stated even clearer. | We had stated such, but made it even clearer – working in the tropics one may tend to forget the importance of snowmelt for river flow and flooding in your part of the world… We added: "
[revised manuscript text omitted]

```

---

## Author Response (AR4)

hess-2015-538

Dear authors,
thanks for these clarifications. However, I am afraid some of the replies left me rather more confused than before. Below I reiterate and further explain my concerns. As these concerns are fundamental to the approach put forward in you manuscript it is crucial to resolve these.

Best regards,
Jan Seibert

Dear editor

Thank you for the continued interest in the manuscript and the further question that may help reduce misunderstandings that our text inadvertently caused. We are pleased to note that the technical aspects of what Fp measures, how it can be derived and used are now settled.

The core of the debate now seems to be the character of what a "watershed health" concept could/should entail. The World Health Organization (WHO) defined health in its broader sense in its 1948 constitution as "a state of complete physical, mental, and social well-being". The word "health" here emphasizes the resultant outcome of an interaction between physiology and environment – not unlike the way we use it here for watershed health. Human health is a holistic concept that is evaluated in its current environment – someone vulnerable to asthma maybe healthy if living in the mountains of Switzerland, and, with the same inherent physical characteristics, very ill if living in Beijing or Delhi with current smog levels.

We added text in the introduction: "Figure 1 clarified how ecosystem structure, ecosystem function and human land use interact in causal loops that can lead to flood damage, its control and/or prevention. A holistic 'watershed health' concept may have to include all three elements, similar to the way a human health concept involves a physical condition of the human body responding to the external environment (e.g. levels of air pollution) and human behaviour that influences exposure and its consequences. The definition of human health has evolved over time. Human health was seen as a state of normal function that could be disrupted from time to time by disease. In 1948 the World Health Organization

(1958) proposed a definition that aimed higher, linking health to well-being, in terms of physical, mental, and social aspects, and not merely the absence of disease and infirmity. Health became seen as the ability to maintain homeostasis and recover from injury, but remained embedded in the environment in which humans function. Various geographically defined watershed health concepts are in use (see for example https://www.epa.gov/hwp/healthy-watersheds-projects-region-5; City of Fort Collins, 2015), employing a range of specific indicators. The spatial aspects of such indicators tend to combine the underlying geology, geomorphology, vegetation and climate, with aspects of anthropogenic change. There is space for a new synthetic indicator or metric."

1)      The runoff values you present in figures 2-4 seem unrealistically high. Could you please comment on these values. As it is not just one figure, I am afraid there is something fundamentally wrong - or I misunderstand something.

We have corrected the scale of Fig. 4A as this was expressed as $m^3 s^{-1}$ (rather than mm $d^{-1}$ as figures 2 and 3); division by the catchment area does not influence the numerical results for $F_p$.

You now changed two of the figures, although the catchments remain to be on the very wet side. For one figure you kept the very high runoff (I guess there are not too many catchments with 6000mm/y worldwide). This makes me wonder whether you would suggest that this approach is suitable mainly for very wet catchments and to be less suitable to 'normal' or more arid catchments. If so this should be clarified.

We have added a further version of figures 1.2 and 1.3 that refer to a situation with 45 rainy days and a total river flow of about 600 mm/year, to address your question. Qualitatively the change in river flow pattern with a decrease in $F_p$ is similar, and the approach we propose is not restricted to the very wet side of the world in which the authors happen to live and work. Having said that, in situations with one or two rainfall events defining a whole hydrological year (as we found in an Australian catchment studied by a colleague in a cooperation project we had in the past), other ways of characterizing the catchment make more sense. Where there is a long and uninterrupted recession curve, not confounded by further rainfall events (or the possibility that such took place beyond the scarce precipitation data one has), one can derive the recession constant in a more direct

way. We have added a few sentences in the discussion to this effect.

2)      Ok

3)      $F_p$ is affected by both rainfall time series and catchment characteristics. It is therefore not clear to me whether $F_p$ really is a good measure of catchment status since $F_p$ also could change without any catchment change if rainfall distributions change.

Yes, just as someone can suffer from asthma when moving from the Swiss mountains to Beijing or Delhi – health is not just a physical state of the body, but the way it functions in a given climate.

In your reply you agree with my concern, but say that FP ..” is reflecting the "health" of the way a watershed interacts with its climate, rather than the land cover as such.". I am not sure I understand the term 'health' here. Basically you are saying a catchment could change from healthy to unhealthy if the climate changes. And it is not really the 'interacting' with the climate which changes. The interacting processes could still be the same, just the forcing changes. Let's assume that you for some reason suddenly have a constant rainfall, and thus runoff if we ignore E here, for the half of the year, where usually no floods occur. This would largely increase $F_p$ (more healthy) but the risk for flooding would stay the same in reality, wouldn't it?

We agree that "forcing" may be better here than "interacting" and have changed the wording in some parts of the text to reflect this.

Your thought experiment is valid. In part II we analysed $F_p$ per quarter of the hydrological year, and indeed found that the estimates for the drier parts of the year are generally higher than those for the wetter part. We added to the discussion of part 2: Choice of the part of the year for which $F_p$ changes are used as indicator may have to depend on the seasonal patterns of rainfall.

As we conclude in part 2, it is a change in $F_p$ rather than $F_p$ itself that provides an indication of a changing health, but requires multiple years of observations to be distinguished from effects to the realized rainfall pattern.

In checking literature that has used the "watershed health" wording, we came across a "flashiness index" that is worth mentioning.

Baker et al. (2004) defined a 'Richards-Baker Flashiness Index' as $\sum_1^n |(q_i - q_{i-1})| / \sum_1^n q_i$. It is based on a similar idea as $F_p$, but derived from the absolute value of observed changes in flow relative to the average flow. Figure 8 compares numerical results for this index with $F_p$ for a number of hydrographs constructed as in Fig. 2A. The two concepts are inversely related, but where $F_p$ is

constrained to the 0-1 interval, the R-B Flashiness Index can attain values up to 2.0, with the value for $F_p = 0$ reflecting the local rainfall regime and its probability of rainfall on subsequent days.

⇨ Figure 8

4)      Human impacts can both increase and decrease Fp, again, I am wondering whether Fp really is a good measure of 'watershed health'

Your reply "We try to provide empirical evidence to help readers answer this question for themselves." honestly did not convince me. I am still struggling to understand why Fp should be a good measure of catchment health, if human activities can shift it in both directions and this measure is heavily influenced by rainfall patterns. Leaving it to the reader to answer this fundamental question, does not seem to be appropriate.

Again, using an analogy of human health, the fact that human behaviour can both increase and decrease the well-functioning of a body generally indicated as health doesn't distract from the term health – or do we miss something?

We have reaffirmed the conclusion that watershed health is here characterized through the flow pattern it generates, leaving the attribution to land cover, rainfall pattern and engineering of that pattern and of changes in pattern  to further location-specific analysis, just as a symptom of a high body temperature can indicate health, but not diagnose the specific illness causing it.

5)      You claim that Fp is related to flood risks. can this be shown on real data? I am not so sure I would agree. Your approach basically assumes that catchment changes result in linear changes in flood risks, but this is obviously often not the case. For instance, if a medium-size reservoir is build, this would increase Fp, but the largest floods would hardly be influenced (because then the reservoir is filled anyway). I guess, another issue here is that the Fp values are based on the continuous, average catchment behavior, whereas floods are single extreme events.

I am afraid I am not convinced by your reply. Here I want to emphasize on the second part of my question: the way you define and compute Fp means that this value reflects the average behaviour of a catchment over long periods. In other words, Fp is also influenced by low flow periods, which should not influence the likelihood for floods.

As stated before, the $Q_t$ vs $Q_{t-1}$ graphs provide a visual test of the homogeneity of the data, with a possible shift in $F_p$ value at higher flow rates quickly noticeable. In the data sets we have so far explored, no evidence of such inhomogeneity appeared, although we cannot exclude that it can occur elsewhere.

We added in the discussion of paper I:

The most damaging floods in any landscape are the result of extreme events that, by their very nature, are hardly represented in any data set. Where $F_p$ is derived from empirical data, extrapolation to rainfall conditions beyond what occurred in the measurement period will, regardless of the method used, have increased uncertainty. The same applies to existing methods for estimating 1:100 or 1:1000 year floods.

And in paper II:

[revised manuscript text omitted]
. A holistic 'watershed health' concept may have to include all three elements, similar to the way a human health concept involves a physical condition of the human body responding to external environment (e.g. levels of air pollution) and human behaviour that influences exposure and its consequences. The definition of human health has evolved over time. Human health was seen as a state of normal function that could be disrupted from time to time by disease. In 1948 the World Health Organization (1958) proposed a definition that aimed higher, linking health to well-being, in terms of physical, mental, and social aspects, and not merely the absence of disease and infirmity. Health became seen as the ability to maintain homeostasis and recover from injury, but remained embedded in the environment in which humans function. Various geographically defined watershed health concepts are in use (see for example https://www.epa.gov/hwp/healthy-watersheds-projects-region-5; City of Fort Collins, 2015, employing a range of specific indicators. The spatial aspects of such indicators tend to combine the underlying geology, geomorphology, vegetation and climate, with aspects of anthropogenic change. There is space for a new synthetic indicator or metric.

[revised manuscript text omitted]

/
```

---

## Author Response (AR5)

**Editor Decision: Publish subject to revisions (further review by Editor and Referees)** (14 Nov 2016) by Prof. Jan Seibert
Comments to the Author:
Dear authors,
thanks for your quick reply. I am not sure I can follow your reasoning with the analogy to the human health. I can see the Fp value similar to the flashiness index as a quantification of streamflow variability, but still I am not convinced about the usefulness as health indicator. If two catchments with the same weather input vary in their Fp value, why should this mean that one is more healthy?
As I am confused and might miss a point here, I will in the next round ask an additional reviewer to have a look on the manuscript to get a fresh opinion.

Best regards,
Jan Seibert

Dear editor

As the word 'health' apparently caused more confusion than that it clarified, we have removed the word from the title and we hope that the manuscript now meets the requirements of the HESS journal

Best regards

Meine van Noordwijk

[revised manuscript text omitted]

/
```

---

## Author Response (AR6)

**HESS-2015 -538.** Author's response to the third reviewer, 31 Jan 2017

Dear editor

We have taken the advice and restructured the manuscript to strengthen the overall storyline, which
is focused on how flood risk reduction and flow buffering as ecosystem service can be quantified in
data-scarce environments, by comparison with flashiness index and base flow indicators. We have
revised the figures and captions throughout.

| Reviewer | Authors |
|---|---|
| I find the concept of this paper to be really interesting, but feel that the papers are still poorly organized, figures and tables need better labeling and updating, and that the authors still need to make a better case for what Fp adds to a given analysis (both in the text and through benchmarking in Part II). | Thanks for the interest – on re-reading the manuscript we realize that by elaborating points in response to various comments, the main storyline has indeed been lost at a number of places. We have overhauled the text, revised figures and captions and added a table that compares $F_p$ and flashiness index in part I, and an overall comparison table in the discussion of part II. |
| ***Major points are summarized as follows:***
 •    The connection from health, to floods, to Fp I think gets lost amongst both papers. I would almost change all references from health to refer to Fp as an indicator of alteration. The threat of flooding differs based on your location within a catchment (upstream or downstream) as well as many different watershed characteristics, including human impact. Also, as stated in a comment below, floods are healthy, so attributing their increase solely to human impact is perhaps missing some points. I think reframing this as an indicator of alteration could be useful. | We have further downplayed the 'health' aspect and refocused on floods as the primary 'salient' point of attention. The word 'health' is still found in the text, in reference to existing literature.
 We fully agree with emphasis of change in $F_p$ rather than $F_p$ itself as primary indicator -- this was indicated before, but is now the primary conclusion.
 There was some reference to the relevance of floods for downstream biota, and there is now a full paragraph in the discussion on this. We don't quite understand what you mean by "so attributing their increase solely to human impact". |
| •    I find it very striking that two large bodies of literature are missing from the paper – the work of Leroy Poff (Natural Flow Regime, etc), and reference to many of the (small) benchmark catchment studies in the US that linked forest harvest to streamflow responses – these would be interesting candidates for further testing of Fp. I understand that you are focusing on floods at the larger scale, but given the connections you draw between forest harvest/recovery and watershed response, it would be remiss to not reference these landmark papers. | There luckily are many more large bodies of literature missing from the paper… The concept of 'natural flow regimes' gets some mention. The older work in the US and Europe on changes in forest catchments gets mentioned primarily through the excellent reviews that have summarized the results. It would of course be interesting if somebody can test the $F_p$ metric on the existing US data, but we don't have access to these data.  Indeed, it is the larger scale floods that are more contested (and potential involve larger values). Meanwhile, a major difference between temperate and tropical watersheds in the absence of snowpacks and snowmelt in the latter now gets mentioned several times. |

| | |
|---|---|
| • Floods are a natural part of the flow regime – while Part II demonstrates that the deviation from these natural flow regimes through time is really what you are looking for, I think this theoretical approach would be worth stating and referencing upfront throughout Part I. Just like most of the indicators of hydrologic alteration literature, you are interested in a deviation from average. | We have done so – it is explicit in the terms 'degradation' and 'restoration' that a change over time is the core interest. Whether it is 'deviation from the average' or 'deviation from what has been the past distribution' is an issue for further debate. |
| • I am left to think that the Fp metric may be worthwhile for comparing within a catchment through time, but may not be effective at comparing across catchments, due to their heterogeneity in all of the effects summarized in Figure 1. It would be nice to include some discussion of the possible limitations of Fp. | We have further clarified that the interannual variation in $F_p$ versus flashiness index as well as 'base flow' has a pattern that differs between catchments, even with the small set (four) of examples discussed here. We have added a table 1 that provides strengths and weaknesses of both indicators. |
| • From Figure 2 (Part I), I am really left thinking that this indicator is a measure of flashiness | In a sense yes, but it also is a metric for base flow and it does correlate with the R-B Flashiness index, but it is not equal to it |
| • Part of the utility of this indicator over flashiness could be your ability to partition it between wet and dry seasons and different flow pathways. At first, the description in Part I was lost on me, but I understood it after reading Part 2. I would better organize this description to frame Fp as a flexible indicator that spans the empirical and modeling realms. Currently, this is somewhat described in Part I, but highlighting the different ways it could be used by a Figure, or by organizing the text better would be really useful. | Thanks for the suggestion, we hope that the current text makes these points more clear. We have added a new Figure 1 that spells out the criteria that a 'metric' must meet in order to find its place in the applied field of discussion between natural resource managers, the wider public and local stakeholders. These lead to the 7 questions framed at the end of the introduction, and used to structure the discussion of part I. |
| • As a new reviewer coming to this paper with fresh eyes, I found that many of the comments from previous reviewers were not addressed, especially points of ambiguity in the text .e.g, sufficiently long period (line 224), and the wording throughout section 2, which I found difficult to follow. It is unclear if changes were made to Figure 1, despite comments from reviewers to this effect. I also find this figure difficult to dissect. | Where you have been non-ambiguous in these comments we have addressed them. We don't quite understand what your issue is in section 2. This is indeed a technical account with rather precise wording. Figure 1 has certainly been changed from the earlier version in the HESS_D manuscript and the changes were appreciated by the one reviewer who provided suggestions. We have made further changes now, to connect it with the terms in Fig. 1. |

| Minor points: | |
|---|---|
| -I found interpreting the figures in part I of the paper based on their captions alone to be very difficult, especially Figures 2, 3 and 8 (and their formatting). Please revise these captions. | We have revised the captions – but the sequence of showing Fig 3 (in the 'methods') before figures 4 and 5 (in the results) is suboptimal for ease of under-standing. Hopefully readers will refer to the M&M section once they are interested in details, and will first look at 'results' |
| -All figures would benefit from some revising, especially using subscripts for the p in Fp, sizing font and axes text to the same size, labeling x and y axes, better labeling of Figures (Figure 4, Part II especially), and construction of figures (Figure 4, Part II – lines should not be used to connect different values – this implies a continuity, but these are different catchments) -I found the results/discussion of part I to be haphazard – if data is presented, whether constructed or real, it should be introduced and discussed prior to being included in the discussion. Furthermore, while I like the comparison of Fp to the flashiness index, this was introduced so late in the paper, and not touched on in the methods, and then not truly analyzed. The authors missed an opportunity to give some thought to how this flashiness index compares to Fp – if the flashiness index describes Fp, then why do we need Fp? Does it say the same thing as Fp? Constructive analysis would certainly make the case for Fp here. | Thanks, we have indeed taken a critical look at all and harmonized them.

We have brought the description of the algorithm used for former figures 2 and 3 into the 'methods' section, and described the graphs generated in the 'results'. The cost of this may be that the explanation of what was Figure 4 (now Fig. 3) will be harder to follow, as no numerical examples have been presented at this stage, but no linear text representation can work for all readers, and once in print readers can switch for and back and skip the technical sections until they have some general idea of what is being done here.

Indeed the flashiness index comparison was an 'add on' in the discussion, and is now fully integrated in the text. |
| Line 45: 30-50% of what? | Rainfall, corrected |
| Line 48: This is true at large scales, but differences in catchments of similar size would break this relationship down | Not sure we understand what you mean here. All statements in this part have references to the literature in which they are based. |
| Line 109: there are several historical papers out of the US that were the first to perform the paired catchment study, highlighting the effect of land cover change on streamflow across longer time periods – work at Coweeta, HJ Andrews, Hubbard Brook, and Fernow Forestes would be relevant | We state that there is indeed 'ample proof' at this scale and in this type of condition. As we are not writing a textbook with full historical perspective, we maintain that use of the review papers that summarized the studies and conclusions is appropriate. There is similarly an extensive literature on forest and floods in Indonesia in the 1920's and 1930's that will be relevant (and is probably lesser known than the well-cited US examples). |
| Line 127: I'm not sure I fully agree – what is the alternative? Mechanistic models? | We elaborated the text a bit here |

| | |
|---|---|
| Line 130 on: also soils | Added |
| Line 136: the rational method and curve number approach were developed to do just this | Yes, thanks – we have adjusted the text here and give more reference to CN and SWAT here. |
| Line 150: I would recommend including more citations here – there are many papers that have demonstrated this | The Hachrowitz paper is a multi-authored review of the PUB effort – we have added a few further references here. |
| Line 310: I think there is an incorrectly placed word on this line! | We hope it has disappeared by now… |
| Line 469: wording | Adjusted |
| Line 491: For Figures 6 and 7 – If the data is used in this paper, you should describe where it came from – I think these implications may be better explored in paper 2, or the data source should be described in paper 1. | We have indeed more fully incorporated this into the paper and use the actual flow data for the four catchments in Paper I, leaving the (model-based_) scenarios for paper II. |
| Line 508: This paper made several assumptions that also should be acknowledged – doing hydrology backward only works well under certain cases | We use this language as in the paper that is referenced – there is acknowledgement in the text surrounding this statement that there is further discussion on where and when it 'works' |
| I find Figure 1 to be too busy to follow! Consider reducing color, size, changing font, and adding arrows in such a way to better show the "flow" | We have reduced colour and harmonized lines |
| Part II:

Given that you wish to relate health to Fp, and its change through time to land cover, it seems as though Part II would almost benefit more with a comparison to flashiness indicators. If flashiness tells the same story, then what is the value of Fp? I still think the argument could be made that it allows some simple process-based analysis. The contribution of Fp in Part II needs to be more clear, and should be benchmarked against another indicator of hydrologic alteration, to show the clear value of using Fp over other indicators. | Thanks for these suggestions. We have added further comparisons of Fp and FI, for the LU change scenarios in the four catchments (while in Paper I this comparison is made for actual flow data). We have also added a new table (Table 6 in new numbering) that tries to summarize responses to the 7 initial questions for a range of 'indicators' |
| Lines 960 – 64 – these are relatively relaxed targets, and may miss peaks. Given that the emphasis of this paper is on high flows, you should include a hydrograph of your model to demonstrate that the model adequately matches high flows. As this would affect your Fp values, it may be why you end up with wide scatter in Figure 3. | We could add examples as supplementary material, but examples (especially for the Wai Besai watershed with the best data) are already available in http://www.worldagroforestry.org/output/genriver/download

The primary reason, we believe, for the scatter are limitations in the rainfall data, with an insufficient number of measurement stations for the given spatial heterogeneity of rainfall. |

| Lines 1011 on: please use words for scenarios instead of abbreviations | Modified |
|---|---|
| Line 1097: wording is confusing | Thanks, we modified the sentence |
| Table 5, please use more descriptive titles or cite your abbreviations – I cannot tell what these titles mean | We have used full words instead of abbreviation now. |

We have highlighted the major changes made in the text in yellow

[revised manuscript text omitted]

**A.** Interests ⇔ Understanding ⇔ Metrics multistakeholder resource management processes

➔ Monitoring ➔ Diagnosis ➔ Tradeoff analysis ➔ Innovation ➔ Scenarios ➔ Negotiations ➔

| | |
|---|---|
| **Basis of current land use policies:**
Deforestation ➔ increased flood risk
Reforestation ➔ reduced flood risk | *Forestry perspective* |
| *Ecohydrology perspective* | **Relationship between land cover & river flow** depends on complex interactions, non-linearities, partial reversibility, climate variability |
| **Engineering of river storage and flow** can control all relevant risks, once these are quantified | *Engineering perspective* |
| *Climate Change adaptation view* | **Climate change** creates new challenges, requiring costly adaptation measures, but climate policy and finance needs clear attribution, cause & effect links |
| **Local land users want river flow to be predictable** but also like to have flexibility in how land use is regulated as part of ecosystem services management | *Local landscape stakeholders* |

**B.**

I) Diagnostic tool to identify and prioritize 'issues' that are or should **Salience** be of public concern and require a policy response.
II) Help in selecting and monitoring management actions.

V. Match with local knowledge and existing policy frameworks.

**Legitimacy**

VI) Empowerment of local stakeholders of resource management through boundary work, bridging local know-ledge, science, and policy-making, and supporting negotiations among stakeholders; basis for wider monitoring and evaluation of conditions and trends, enhancing transparency of governance.

**Metric**

**Credibility**

III) Succinct representation of current understanding of system performance and options.

IV) Operational link with primary data, known statistical distributions and confidence intervals that allow assessment of change as part of, or beyond 'normal' ranges.

VII) Basis, as 'boundary object', of 'performance-based' contracts and widely supported commitments to resolve 'issues'.

[revised manuscript text omitted]

/

---

## Author Response (AR7)

hess-2015-538
March 2 2017

Response to reviewers
Dear Editor
We have addressed the remaining concern in the current version of the manuscript.

| Comments | Response |
|---|---|
| The text is reading well, and the authors have addressed all of my comments. The authors have clearly undertaken a very challenging task of parsing their original manuscript into two separate companion papers, and have done this admirably. I commend them for taking a step back at this late stage and synthesizing the text and redirecting their storyline. | Thank you for the positive evaluation of the most recent version of the manuscript |
| My remaining concerns are: | |
| (1) the phrasing of section 3.2, Numerical Examples (lines 372 – 380)
In section 3.2, the phrasing is a little confusing – leading with the method instead of why it was done. Monte Carlo simulations are tools used for different purposes – it would be helpful to know the purpose upfront in this paragraph. | The section now reads:
"For visualizing the effects of stochastic rainfall on river flow according to equation [1] a spreadsheet model that is available from the authors on request was used in 'Monte Carlo' simulations. Fixed values for $F_p$ were used in combination with a stochastic $Q_{a,t}$ value. The latter was obtained from a random generator (rand) with two settings for a (truncated) sinus-based daily rainfall probability: A) one for situations that have approximately 120 rainy days, and an annual Q of around 1600 mm, and B) one that leads to around 45 rainy days and an annual total around 600 mm. Maximum daily $Q_{a,t}$ was chosen as 60 mm in both cases. For the figures, realizations for various $F_p$ values were retained that were within 10% of this number of rainy days and annual flow total, to focus on the effects of $F_p$ as such. " |
| (2) Figures // Part I: | |
| -Minor, but Figure 1 part B could be cleaned up, e.g., lines in alignment (if desired), etc | Thanks, we have simplified it |
| -Figure 3: Correct subscript on all Fp's, use Qa instead of Qadd (or make consistent between figures and captions), words instead of abbreviations (e.g., StDev), Day of year starting from when (if Jan 1 – use Julian Day instead) | Thanks, we have checked figures and text for consistency on $Q_a$ vs $Q_{add}$ and hope all $F_p$'s now have the p as subscript; we shifted to Julian day and spelled out the abbreviations. |
| -Figure 3 and 4 refer to discharge using different words – keep consistency | We adopted discharge as the standard term |
| -Figures 6 and 7: Correct subscript with the p in Fp | thanks |
| Part II: | |
| -Figure 2: Axis title and axis are overlapping, clarify time step of values shown | Adjusted |

| | |
|---|---|
| -Figure 4b: impossible to distinguish the six points as shown – could you jitter them, or show them next to each other? If this is the point, that's fine too. | We made the points larger, so they can be distinguished |
| -Figures 5 & 6: define abbreviations in legend or spell out – I couldn't find if they were used elsewhere in a Table or the paper | Done |
| -Figure 8: consider coloring the text for the line fits by the colors corresponding to catchments | Thanks, done |

We have combined figures 8 and 9 of part 1 to ensure they are presented at the same size.

The final set of figures is:

**Figures Part I:**

Figure 1. A. Multiple perspectives on the way flood risk is to be understood, monitored and handled according to different knowledge systems; B. Basic requirements for a 'metric' to be used in public discussions of natural resource management issues that deserve to be resolved and acted upon (modified from van Noordwijk et al., 2016)

**Salience:**

① Exposure human presence  ② Hazard frequency & duration

③ Vulnerability Victim, **damage** and its economic value

*Avoided flood damage as perceived Ecosystem Service*

**Credibility:**

④ Directly observable hydrograph

⑤ **Topography & engineered** river channel, reservoirs, flood plain (and its subsi-dence), dykes, drainage, storage, extractions

Climate variability and change

⑥ $Q = P - E - \Delta S$

⑦ Watershed functions: pathways, water use and flow buffering

⑧

*Avoided degradation, active restoration*

**7A** *Land cover:*
- Natural forest
- Forest-derived
- Plantations
- Tree-based Ag
- Open-field Ag
- Degraded lands
- Settlements

*Spatial configuration*

**7B** *Patch-level*
- Rainfall interception
- Infiltration
- Surface filter effects
- Soil macroporosity (decline & buildup)
- Water storage and use for transpiration

*Spatial configuration*

**7D** *Riparian vegetation*
- Buffer and filter effects

*Hillslope/landscape*
- Drainage vs retention
- Buffer and filter effects
- 'Effective rainfall'

**7C**

***Ecosystem structure  ⇔  Ecosystem function  // watershed management***

[revised manuscript text omitted]

---

## Author Response (AR8)

Dear Meine,

Many thanks for your email.

Please create one Latex file for each part und put them in a .zip file. You should be able to upload the .zip file in the "text" box, where only doc/docx/tex are allowed (normally).

In case this does not work, please upload part 1 in the text box. Part 2 needs to be added to the figure .zip file. This is not the correct way, but it will work.

I will make an internal remark for my colleagues from the production office, that this paper is a split paper.

In case you have any further question, contact me again.

Kind regards,

Anna
* * *
Copernicus Publications
The Innovative Open-Access Publisher

Anna Feist-Polner
Editorial Support

Copernicus GmbH
Bahnhofsallee 1e
37081 Göttingen
Germany

Phone: +49 551 90 03 39 41
Fax: +49 551 90 03 39 90 41

http://www.copernicus.org
@copernicus_org
* * *
Copernicus Gesellschaft mbH
USt-IdNr.: DE216566440
Based in Göttingen, Germany
Registered in HRB 131 298
County Court Göttingen
Managing Director Thies Martin Rasmussen
* * *
**From:** van Noordwijk, Meine (ICRAF) [mailto:M.vanNoordwijk@cgiar.org]
**Sent:** 04 Apr 2017 23:44
**To:** Copernicus Publications Editorial Support
**Cc:** Tanika, Lisa (ICRAF); Lusiana, Betha (ICRAF)
**Subject:** hess-2015-538 (author) - manuscript accepted for final publication

Dear Natascha
We're pleased with the acceptance of our manuscript – as it actually has become two in the process, we seek your advice. We are transferring all the content to Latex – but in doing so it becomes two separate files. How should we upload this for further processing?
Best regards
Meine van Noordwijk